# Metabolism-based targeting of MYC via MPC-SOD2 axis-mediated oxidation promotes cellular differentiation in group 3 medulloblastoma

Emma Martell[1,2], Helgi Kuzmychova [1], Esha Kaul[3], Harshal Senthil[1], Subir Roy Chowdhury[4], Ludivine Coudière Morrison[5], Agnes Fresnoza[6], Jamie Zagozewski[5], Chitra Venugopal[7,8], Chris M. Anderson[9,10], Sheila K. Singh [7,8,11], Versha Banerji[4,5,12,13], Tamra E. Werbowetski-Ogilvie [5,14] & Tanveer Sharif [1,2,4] ✉

Group 3 medulloblastoma (G3 MB) carries the worst prognosis of all MB subgroups. MYC oncoprotein is elevated in G3 MB tumors; however, the mechanisms that support MYC abundance remain unclear. Using metabolic and mechanistic profiling, we pinpoint a role for mitochondrial metabolism in regulating MYC. Complex-I inhibition decreases MYC abundance in G3 MB, attenuates the expression of MYC-downstream targets, induces differentiation, and prolongs male animal survival. Mechanistically, complex-I inhibition increases inactivating acetylation of antioxidant enzyme SOD2 at K68 and K122, triggering the accumulation of mitochondrial reactive oxygen species that promotes MYC oxidation and degradation in a mitochondrial pyruvate carrier (MPC)-dependent manner. MPC inhibition blocks the acetylation of SOD2 and oxidation of MYC, restoring MYC abundance and self-renewal capacity in G3 MB cells following complex-I inhibition. Identification of this MPC-SOD2 signaling axis reveals a role for metabolism in regulating MYC protein abundance that has clinical implications for treating G3 MB.

Brain tumors are the leading cause of cancer death in children under 20[1]. Medulloblastoma (MB) is the most common primary pediatric brain malignancy, representing over 20% of newly diagnosed childhood central nervous system cancers[2]. Over the past decade, proteogenomic profiling of hundreds of MB tumors from several independent groups has classified MB tumors into at least four consensus molecular subgroups: Wingless (WNT) MB, Sonic Hedgehog (SHH) MB, group 3 (G3) MB and group 4 (G4) MB[3,4]. The prognosis for

[1]Department of Pathology, Rady Faculty of Health Sciences, University of Manitoba, Winnipeg, MB, Canada. [2]Department of Human Anatomy and Cell Science, Rady Faculty of Health Sciences, University of Manitoba, Winnipeg, MB, Canada. [3]Faculty of Science, University of Manitoba, Winnipeg, MB, Canada. [4]CancerCare Manitoba, Winnipeg, MB, Canada. [5]Department of Biochemistry and Medical Genetics, Rady Faculty of Health Sciences, University of Manitoba, Winnipeg, MB, Canada. [6]Central Animal Care Services, University of Manitoba, Winnipeg, MB, Canada. [7]McMaster Stem Cell and Cancer Research Institute, McMaster University, Hamilton, ON, Canada. [8]Department of Surgery, Faculty of Health Sciences, McMaster University, Hamilton, ON, Canada. [9]Neuroscience Research Program, Kleysen Institute for Advanced Medicine, Health Sciences Centre, Winnipeg, MB, Canada. [10]Department of Pharmacology and Therapeutics, Rady Faculty of Health Sciences, University of Manitoba, Winnipeg, MB, Canada. [11]Department of Biochemistry and Biomedical Sciences, McMaster University, Hamilton, ON, Canada. [12]Department of Internal Medicine, Rady Faculty of Health Sciences, University of Manitoba, Winnipeg, MB, Canada. [13]Department of Medical Oncology and Hematology, CancerCare Manitoba, Winnipeg, MB, Canada. [14]Present address: CancerCare Manitoba, Winnipeg, MB, Canada. ✉e-mail: Tanveer.Sharif@umanitoba.ca

MB patients varies greatly depending on the molecular subgroup of the tumor, with the best survival rate being >95% in the WNT subgroup and the deadliest subgroup, G3 MB, harboring a survival rate of <60%[5].

The oncogene c-*MYC* (MYC hereafter) is commonly amplified and overexpressed in G3 MB but not in other subgroups[6]. High levels of MYC are associated with poor outcomes (~40% survival) and unresponsiveness to almost all current therapies[6]. Treatment options for G3 MB patients are limited, where the current standard of care involves surgery followed by radiation therapy and cytotoxic chemotherapy (i.e., cisplatin and vincristine with lomustine or cyclophosphamide)[7]. Moreover, patients who survive often suffer severe long-term health problems due to the treatments they received as a child[8]. While there are limited targeted therapies under clinical evaluation for MB brain tumors (NCT03434262; NCT04023669; and NCT01878617), further efforts are needed to expand treatment options and to identify effective agents with less toxic effects on the developing nervous system.

Although various cancer types, including many G3 MB tumors, exhibit aberrantly high MYC abundance, clinical targeting of MYC has remained elusive[9–12]. MYC is a key transcription factor that potentially regulates ~15% of the genome, controlling almost every cellular process[13]. Moreover, MYC regulates different targets depending on cell type. Still, the broad spectrum of MYC-controlled processes means that direct targeting of MYC can cause undesirable side effects and toxicity to normal cells that depend on MYC function[14]. Moreover, MYC is an intrinsically disordered protein that lacks defined targetable structures for small molecule inhibitors[10]. Hence, we posit efforts can be re-directed to exploit indirect means of targeting MYC within hyperactive MB cells. Unfortunately, the regulatory mechanisms that support MYC abundance remain unclear. Hence, a better understanding of the processes that help maintain high MYC levels in G3 MB cells is much needed to identify targets that could be exploited for therapeutic purposes.

One of the significant functions of MYC is to regulate cellular bioenergetics[15]. MYC controls multiple metabolic pathways, including glycolysis, glutaminolysis, fatty acid oxidation, and oxidative phosphorylation (OXPHOS)[15,16]. Cancer cells can leverage these metabolic pathways to maintain their increased bioenergetic and biosynthetic demands. Hence, *MYC* amplification is critical for supporting tumor metabolic reprogramming[15]. On the other hand, emerging evidence demonstrates that metabolism is far more essential in maintaining cellular expression profiles than previously appreciated[17]. Metabolism can directly influence gene and protein expression by modulating the supply of precursors for epigenetic and post-translational modifications[17,18]. Although targeting metabolism to block tumor energy production has been a sought-after therapeutic strategy for several years, manipulating metabolism to target oncogenic factors represents a unique approach. While the role of MYC in regulating metabolism has been extensively studied over three decades, it is unclear whether metabolism may reciprocally regulate MYC to support its enhanced abundance in cancer. Such reciprocal regulatory relationships have been demonstrated for other major transcription factors such as TP53, which has been shown to regulate the transcription of NAD$^+$ synthesizing enzymes[19]. These NAD + synthesizing enzymes can in turn modulate TP53 stability by mediating the activity of NAD$^+$-dependent deacetylases[20]. Yet, the role of metabolism for regulating MYC remains relatively unexplored.

Here, we investigate the concept of leveraging metabolism-targeting interventions as an indirect means of modulating MYC abundance and activity in G3 MB. We identify a targetable metabolic vulnerability in which G3 MB cells demonstrate exquisite sensitivity towards inhibitors of complex-I in the electron transport chain (ETC) but not to other metabolism-targeting agents, including glycolytic inhibitors. Moreover, we uncover a mechanism whereby targeting

OXPHOS via inhibition of complex-I leads to inactivating acetylation of antioxidant enzyme SOD2, inducing rapid and specific oxidation of MYC followed by proteasomal degradation in a mitochondrial pyruvate carrier (MPC)-dependent manner. The therapeutic implications of these findings are demonstrated by the observations that OXPHOS inhibition decreases MYC levels and significantly suppresses the growth and self-renewal capacity of various well-characterized G3 MB cell lines, and other *MYC*-amplified cancer cells from multiple tumor types, including ovarian, colorectal and breast carcinomas. Ultimately, treatment with an orally bioavailable and blood-brain barrier (BBB) permeable complex-I inhibitor impairs the growth of intracerebellar orthotopic G3 MB tumor xenografts, induces differentiation, and significantly prolongs animal survival. These findings reveal a targetable role for OXPHOS metabolism that contributes to MYC abundance via the MPC-SOD2 axis in G3 MB.

## Results
### Complex I inhibition suppresses MYC abundance
*MYC* amplification is common in many aggressive tumors including, ovarian cancer, breast cancer, pancreatic ductal adenocarcinoma, colorectal carcinoma, and others (Fig. 1A)[21]. *MYC* is amplified in approximately 11% of G3 MB tumors[5,22], making it one of the top 5 tumor types with the highest incidence of *MYC* amplification (Fig. 1A). *MYC* amplification universally corresponds with a worse prognosis across cancer types (Supplementary Fig. 1A). In G3 MB tumors specifically, high *MYC* levels correspond with an overall 10-year survival rate of <50%, compared to >70% survival in patients with *MYC*$^{LOW}$ tumors (Fig. 1B). Because of this, MYC is an attractive therapeutic target for cancer therapy. Despite the challenges associated with designing small-molecule inhibitors of MYC activity, progress has been made in developing agents that interfere with MYC interactions and block MYC transcriptional activity[11]. For example, 10058-F4 targets the interaction between MYC and its essential binding partner MAX[23]. However, these inhibitors have yet to be translated into clinical settings due to concerns associated with toxicity[11]. Indeed, we observed that although 10058-F4 effectively reduced the cell number of the well-established *MYC*-amplified G3 MB cell line HD-MB03 at various doses, it had a similar effect on normal human astrocytes, which constitute a major cellular component of the brain (Supplementary Fig. 1B). We found that even a single low dose (10 μM) treatment of 10058-F4, which did not impair the cell number of HDMB03 cells or astrocytes after 24 hours, significantly inhibited the proliferation of both G3 MB and normal astrocytes after treatment for several days (Fig. 1C). Therefore, alternative strategies are required to target MYC selectively in G3 MB while mitigating the harm to normal cell populations.

To identify potential processes that could be targeted to modulate MYC expression or activity in G3 MB cells, we analyzed differentially enriched cellular processes in MYC-driven *versus* non-MYC activated G3 MB tumors. Proteomics analyses performed on 14 different G3 MB patient tumors previously clustered the samples into MYC-activated or non-MYC-activated tumors[4]. Functional enrichment analysis of the proteins that were significantly differentially expressed (fold-change >2.0 and $p < 0.05$) in MYC-activated *versus* non-MYC activated G3 MB tumors revealed that one of the processes most significantly altered in MYC-activated G3 MB tumors is metabolic processes ($p = 1.0 \times 10^{-10}$) (Supplementary Fig. 1C). Further gene-set enrichment analysis (GSEA) using normalized enrichment score (NES) confirmed that MYC-activated tumors are enriched in Hallmark MYC targets (V1−NES: 2.37; $p < 0.0001$; V2−NES: 2.75; $p < 0.0001$) and demonstrated that MYC-activated tumors have enriched abundance of enzymes involved in glycolysis (NES: 1.25; $p = 0.069$), glutamine metabolism (NES 1.90; $p = 0.002$), fatty acid metabolism (NES: 1.38; $p = 0.018$), and oxidative phosphorylation (NES: 2.47; $p < 0.0001$) (Fig. 1D). Analysis of a larger RNA-sequencing dataset of 144 G3 MB tumors demonstrated similar trends of increased enrichment in these

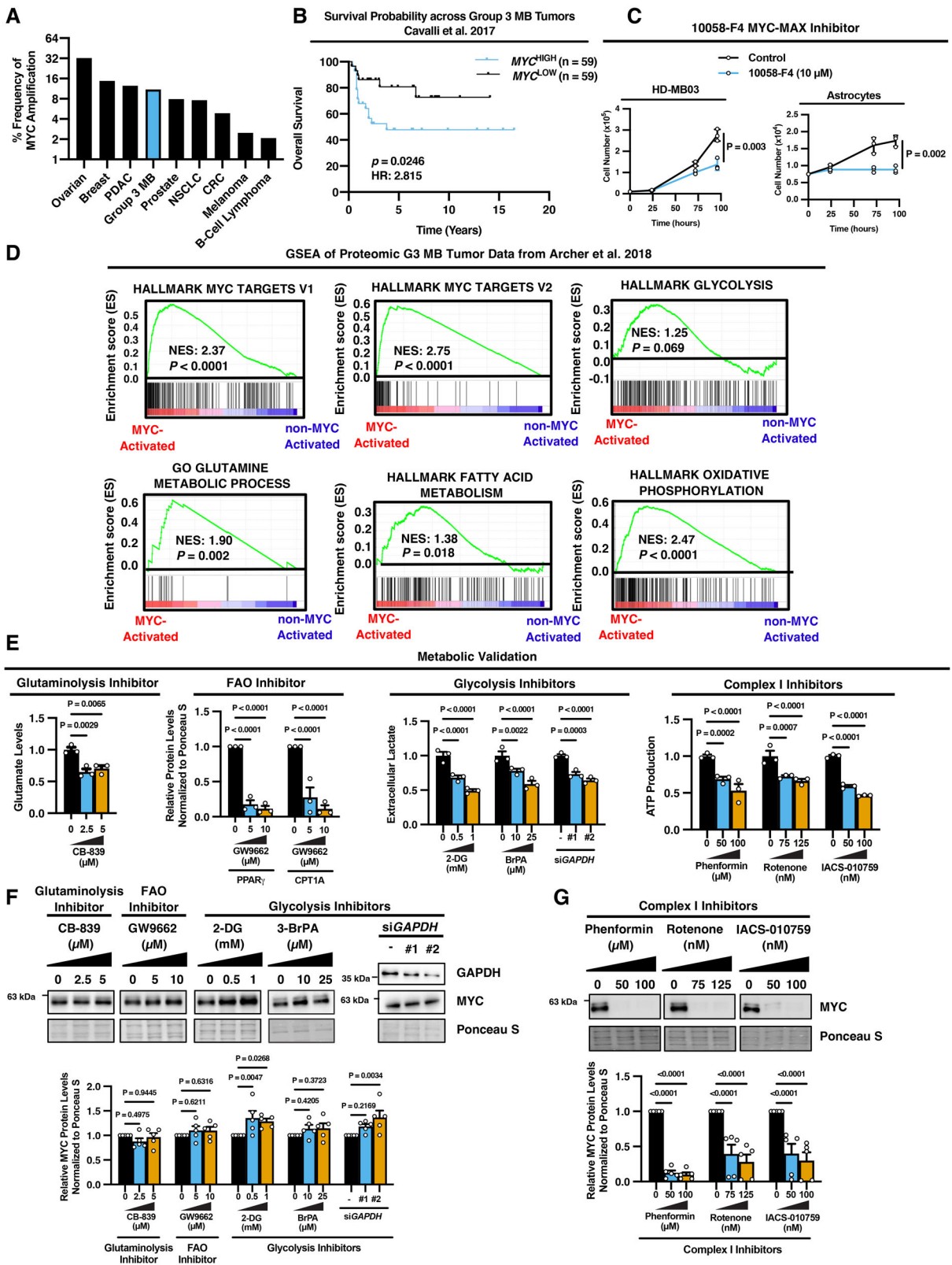

metabolic gene expression signatures in MYC-amplified versus non-MYC-amplified tumors, although the enrichment scores were not significant (Supplementary Fig. 1D). This discrepancy could be explained by the fact that transcriptional signatures do not always correspond with differences in protein levels, and there could be post-translational mechanisms accounting for the correlations observed between MYC and metabolic proteins in patient tumors.

While MYC has been reported to regulate these various metabolic processes[15,16], the concept that metabolism may play a role in reciprocally regulating MYC levels or activity is relatively unexplored. To investigate this notion, we utilized a well-established primary human MYC-amplified G3 MB line, HD-MB03[24]. Using agents targeting the various metabolic processes that were enriched in MYC-activated G3 MB tumors, we treated HD-MB03 G3 MB cells with doses within their

**Fig. 1 | Metabolism reciprocally regulates MYC levels in G3 MB. A** % Frequency of MYC amplifications across different tumor types (ovarian, *N* = 584 cases; breast, *N* = 1084 cases; pancreatic ductal adenocarcinoma, PDAC, *N* = 184 cases; Group 3 medulloblastoma; G3 MB, *N* = 168 patients; prostate, *N* = 494 cases; non-small cell lung carcinoma, NSCLC, *N* = 487 cases; colorectal carcinoma, CRC, *N* = 594 cases; melanoma, *N* = 444 cases; and B-cell lymphoma, *N* = 48 cases)[21,22,59,60]. **B** Kaplan–Meier analysis of overall survival probability of G3 MB patients from Cavalli et al.[5] *N* = 118 patients. *P* values and hazard ratio were determined using the logrank method. **C** HD-MB03 cells and normal human astrocytes were treated with 10058-F4 and cells were counted at indicated time points. Data are presented as mean values of *N* = 3 experimental replicates ±SEM. *P* values were calculated using two-sided student's *t* test. **D** Analysis of proteomics data of G3 MB tumors (*N* = 14) and gene set enrichment analysis (GSEA) was performed with Hallmark and gene ontology (GO) gene sets (enrichment scores and *P* values are calculated using a weighted two-sided Kolmogorov–Smirnov-like statistic and normalized based on the size of the gene set to yield the normalized enrichment score, NES)[4,61,62]. **E** HD-MB03 cells were treated for 24 hours with indicated doses of metabolic inhibitors and the indicated metabolic validation was performed including measurement of: glutamate levels, protein levels of fatty acid oxidation (FAO) enzymes PPARγ and CPT1A, lactate secretion, and ATP production. Data are presented as mean values of *N* = 3 experimental replicates ± SEM. *P* values are calculated using two-way ANOVA with Tukey's multiple comparisons. **F**, **G** HD-MB03 cells were treated for 24 hours with indicated doses of **F** glutaminolysis, FAO, glycolysis, and **G** complex-I inhibitors, and MYC protein levels were monitored by immunoblotting. Graph represents mean values of densitometry quantification of blots from *N* = 5 experimental replicates ± SEM. *P* values are calculated using two-way ANOVA with Tukey's multiple comparisons. Source data provided in Source Data File.

reported therapeutic ranges for 24 hours and we validated that these doses significantly impaired their target metabolic pathways where; the glutaminolysis inhibitor CB-839 reduced glutamate levels, the fatty acid oxidation (FAO) inhibitor GW-9662 reduced the levels of the FAO transcription factor PPARγ and its target CPT1A (mitochondrial fatty acid transporter), inhibition of glycolysis using 2-DG, 3-BrPA, or small-interfering RNA targeting *GAPDH* (si*GAPDH)* impaired lactate production, and the electron transport chain (ETC) inhibitors phenformin, rotenone, and IACS-010759 reduced ATP levels (Fig. 1E). We then monitored the effect of these metabolic agents on MYC protein abundance. We found that inhibitors of glutaminolysis, FAO, or glycolysis did not suppress the levels of MYC protein or its critical MYC binding partner, MAX (Fig. 1F and Supplementary Fig. 1E). In stark contrast, multiple inhibitors of complex-I in the ETC, which block OXPHOS, led to a drastic depletion of MYC and MAX abundance in HD-MB03 G3 MB cells (Fig. 1G and Supplementary Fig. 1F). This includes the orally bioavailable and BBB penetrable complex-I targeting agent, IACS-010759 (Fig. 1G and Supplementary Fig. 1F)[25]. IACS-010759 is currently under clinical evaluation to treat acute myeloid leukemia (AML), pancreatic cancer, and breast cancer (NCT02882321 and NCT03291938), but has never been tested against MB brain tumors. These results unveil a targetable metabolic vulnerability where complex-I inhibitors suppress MYC abundance in G3 MB cells.

### IACS-010759 treatment promotes differentiation and suppresses the growth and stemness of G3 MB cells

As we observed that complex-I inhibitors, but not other metabolism-targeting agents, suppressed MYC levels in HD-MB03 G3 MB cells, we evaluated how this regulation of MYC corresponds with cell growth and viability. In line with our findings that MYC levels remain intact following FAO and glycolysis inhibition, we similarly observed no significant impact on cell numbers in HD-MB03 cells 24 hours after treatment with the same doses (Supplementary Fig. 2A). However, we found that while inhibition of glutaminolysis had no effect on MYC levels, CB-839 treatment significantly decreased the cell number of HD-MB03 cells after 24 hours, possibly through a MYC-independent mechanism (Supplementary Fig. 2A). In contrast, we found that suppressing OXPHOS using complex-I inhibitors (phenformin, rotenone, and IACS-010759), which significantly decreased MYC abundance, also impaired the cell number of HD-MB03 cells after 24 hours of treatment (Supplementary Fig. 2A).

To further characterize the safety and efficacy of OXPHOS inhibitors for the treatment of G3 MB tumors, we tested the BBB-permeable complex-I targeting agent IACS-010759 against four different MYC-amplified G3 MB cell lines (HD-MB03, SU_MB002, MB3W1, and D283)[24,26–28] as well as normal human astrocytes and neural stem cells (NSCs). We found that a single 100 nM dose of IACS-010759 significantly impaired the proliferation (average of 60% decrease in cell numbers) of all four distinct G3 MB cell lines (HD-MB03, SU_MB002,

MB3W1, and D283) for up to four days (Fig. 2A). Notably, the proliferation of normal human brain astrocytes or neural stem cells (NSCs) was unaffected by IACS-010759 treatment (Fig. 2B), which is critical as damage to NSC populations is mainly responsible for the neurotoxicity caused by cancer treatments in children. These findings reveal a potential therapeutic opportunity where complex-I inhibition significantly suppresses the growth of G3 MB cells at low doses and is well-tolerated by normal brain cell populations. Furthermore, we confirmed that IACS-010759 acts similarly on MYC levels in other G3 MB cells. We found that MYC abundance markedly decreased after 24 hours of treatment with IACS-010759 in HD-MB03, SU_MB002, MB3W1, and, D283 cells (Fig. 2C and Supplementary Fig. 2B). However, the same IACS-010759 treatment did not affect basal levels of MYC in normal human brain astrocytes or NSCs (Fig. 2D and Supplementary Fig. 2B), highlighting that complex-I inhibition selectively suppresses MYC in cancer cells. We further validated the correlation between mitochondrial oxidative metabolism and MYC abundance that was observed in patient tumors within our G3 MB cell models. We found that D283, MB3W1, SU_MB002, and HD-MB03 cells display varying basal abundance of MYC, with D283 and MB3W1 cells harboring the lowest MYC levels of the four G3 MB cells, and HD-MB03 displaying the highest levels (Supplementary Fig. 2C). We found that the levels of critical mitochondrial metabolic enzymes involved in the tricarboxylic acid cycle (TCA); citrate synthase (CS), succinate dehydrogenase A (SDHA), and fumarate hydratase (FH), as well as the ETC (SDHA), correlated with increasing levels of MYC in G3 MB cell lines (Supplementary Fig. 2C). Moreover, the maximal oxygen consumption rate (OCR), which is a readout of mitochondrial OXPHOS metabolic capacity, similarly correlated with MYC abundance in G3 MB cells (*R* = 0.9060; Supplementary Fig. 2D). Altogether, these findings highlight the important relationship between MYC abundance and OXPHOS metabolism in G3 MB cells.

Our findings suggest that complex-I inhibitors such as IACS-010759 may be an attractive therapeutic strategy for treating various types of MYC-activated cancers. In further support of this notion, we found that 100 nM IACS-010759 treatment significantly decreased the abundance of MYC in various types of cancer cells from MYC-activated ovarian (A2780 and HEYA8), colorectal (HCT116 and SW480), and breast (MDA-MB-468) cancer cell lines (Supplementary Fig. 3A). This loss in MYC abundance was accompanied by a significant decrease in the cell number of all five cancer cell lines after 24 hours of IACS-010759 treatment (Supplementary Fig. 3B). These results suggest that complex-I inhibitors may provide an effective therapeutic strategy for targeting MYC across the cancer spectrum.

We further determined that the loss of MYC abundance and cell number after 24 hours of IACS-010759 treatment in G3 MB cells is not the result of apoptotic cell death. We could not detect the presence of the active cleaved form of the executioner caspase-3 or cleavage of its downstream target PARP1 following 24 hours of IACS-010759 treatment in HD-MB03 or SU_MB002 G3 MB cells (Fig. 2E and

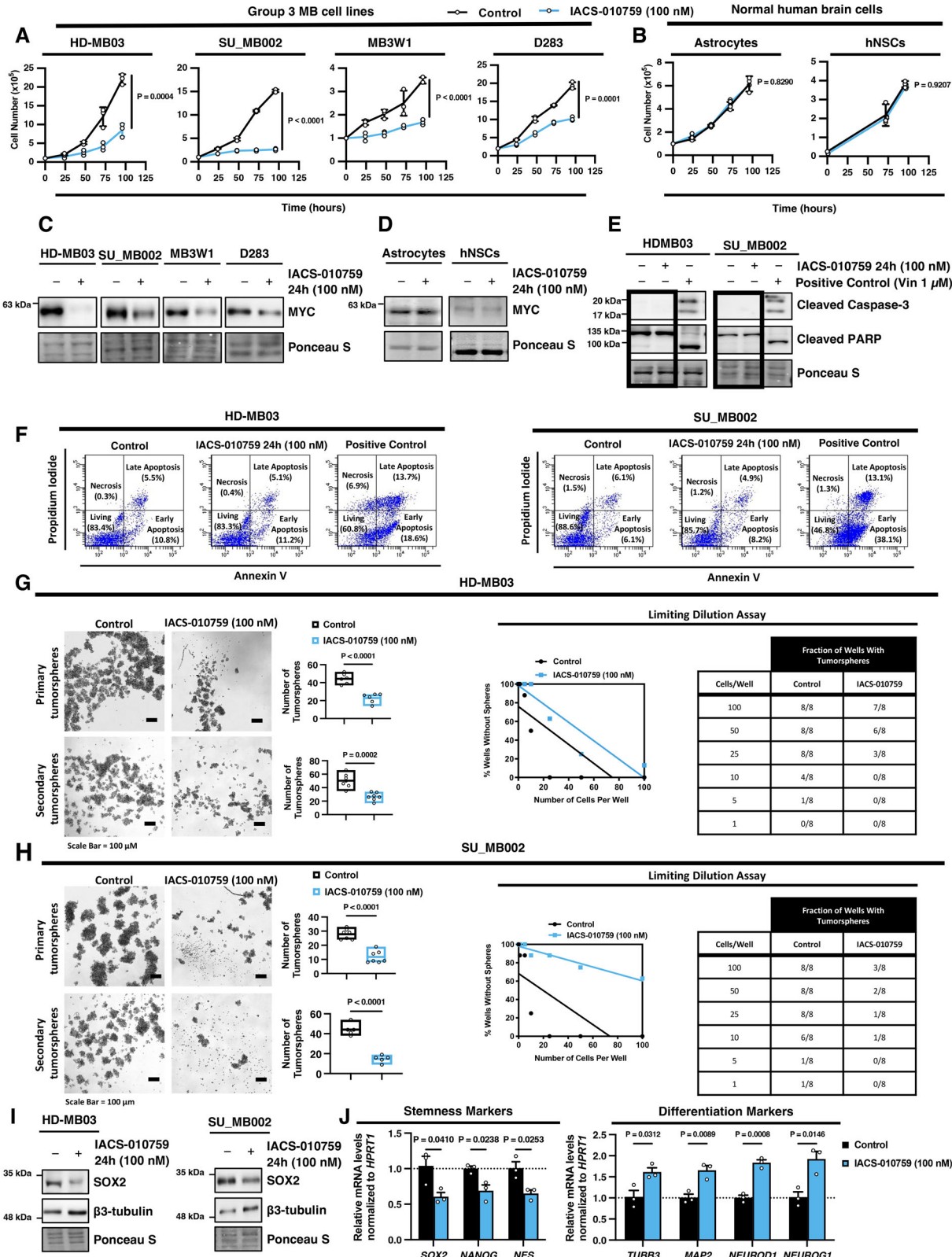

Supplementary Fig. 3C). In contrast, treatment with the cytotoxic chemotherapy Vincristine (1 μM) strongly induced activation of cleaved caspase-3, which promoted cleavage of PARP1 (Fig. 2E and Supplementary Fig. 3C). Furthermore, annexin-V/propidium iodide (PI) staining demonstrated no significant increase in apoptotic cells with exposed phosphatidylserine following 24 hours of IACS-010759 treatment (Fig. 2F and Supplementary Fig. 3D).

G3 MB cells are thought to originate from neural progenitor populations that arise during rhombic lip development, and these malignant counterparts maintain stem-cell properties such as the ability to self-renew and undergo multi-lineage differentiation, which contributes to their aggressive nature[29,30]. Moreover, MYC is known to support stem-cell maintenance and inhibit differentiation[31]. Therefore, loss of MYC levels may correspond with a decrease in stemness

**Fig. 2 | Complex I inhibition impairs stemness and promotes differentiation in G3 MB cells. A** Various G3 MB cells (HD-MB03, SU_MB002, MB3W1, D283) and **B** normal human brain cells (astrocytes and human neural stem cells; hNSCs) were treated with IACS-010759, and cells were counted at indicated time points. Data are presented as mean values of $N = 3$ experimental replicates ± SEM. *P* values were calculated using two-sided student's *t* test. **C, D** Representative western blots of MYC from $N = 3$ experimental replicates of **C** HD-MB03, SU_MB002, MB3W1, D283, and **D** normal human brain cells (astrocytes and human neural stem cells; hNSCs) treated with IACS-010759 for 24 hours. **E, F** Representative **E** western blots of cleaved caspase-3 and cleaved PARP and **F** dot plots of Annexin V/PI positive populations from $N = 3$ experimental replicates of HD-MB03 and SU_MB002 cells treated with IACS-010759 for 24 hours or positive control cytotoxic chemotherapy vincristine (Vin; 1 μM). **G, H** Representative primary (HD-MB03: $N = 6$ and SU_MB002: $N = 7$ experimental replicates) and secondary (HD-MB03: $N = 7$ and

SU_MB002: $N = 5$ experimental replicates*)* tumorsphere images (Scale Bar = 100 μm) and quantification of total sphere number (>50 μm) presented as box-plot with the box limits at minima and maxima and center line at mean, graph and table showing the proportion of wells (HD-MB03 & SU_MB002: $N = 8$ experimental replicates) with tumorspheres formed in limiting dilution assay from **G** HD-MB03 and **H** SU_MB002 cells treated with IACS-010759. *P* values were calculated using two-sided student's *t* test. **I** Representative western blots of SOX2 and β3-tubulin from $N = 3$ experimental replicates of HD-MB03 and SU_MB002 cells treated with IACS-010759 for 24 hours. **J** HD-MB03 cells were treated with IACS-010759 for 24 hours and the mRNA expression of stemness (*SOX2, NANOG,* and *NES*) and differentiation (*TUBB3, MAP2, NEUROD1,* and *NEUROG1*) markers was quantified by qRT-PCR analysis presented as mean ± SEM from $N = 3$ experimental replicates. *P* values were calculated using two-sided student's *t* test. Source data provided in Source Data File.

capacity and a shift towards a more differentiated and non-proliferative phenotype that may contribute to the decrease in cell numbers following IACS-010759 treatment. Indeed, IACS-010759 treatment decreased the number of primary and secondary tumorspheres formed by HD-MB03 and SU_MB002 cells, indicating an impairment in the proliferation and self-renewal of stem cell populations (Fig. 2G–H). Additionally, limiting dilution assay (LDA) demonstrated that IACS-010759 treatment decreases the proportion of self-renewing stem cell populations, which coincided with a decrease in the levels of the stemness transcription factor SOX2 (Fig. 2G–I; Supplementary Fig. 3E). Similarly, IACS-010759 treatment also inhibited tumorsphere formation in ovarian cancer cells (A2780 and HEYA8; Supplementary Fig. 3F). This loss in stemness capacity triggered by IACS-010759 treatment corresponded with induction of differentiation in G3 MB cells, as evidenced by an increase in the neuronal lineage marker β3-tubulin/TUBB3 (Fig. 2I and Supplementary Fig. 3E). Additional markers of stemness and differentiation were validated in HD-MB03 G3 MB cells by qRT-PCR analysis, and we confirmed that the expression of several stemness factors (*SOX2, NANOG*, and *NES*) decreased following IACS-010759 treatment while markers of differentiation (*TUBB3, MAP2, NEUROD1,* and *NEUROG1*) were significantly increased (Fig. 2J). Critically, the same IACS-010759 treatment did not impact the tumorsphere formation capacity or alter the levels of SOX2 or β3-tubulin in normal human NSCs (Supplementary Fig. 3G–H). Altogether, these findings demonstrate that IACS-010759 treatment diminishes the stem cell population of G3 MB cells through a combination of impaired proliferation and hampering stemness properties while promoting a transition towards a more differentiated, less proliferative state. In contrast, normal human NSCs maintain their stemness capacity under IACS-010759 treatment. These findings highlight the ability of IACS-010759 to selectively suppress malignant tumor cells without harming normal cell populations necessary for proper development.

### Downregulation of MYC is important for the response of G3 MB cells to IACS-010759 treatment

To determine the significance of MYC downregulation in the response of G3 MB cells to IACS-010759 treatment, we performed comprehensive gene expression profiling of Hallmark MYC target genes curated from The Molecular Signatures Database (MSigDB)[32], as well as additional well-characterized downstream targets of MYC transcriptional activation (Fig. 3A). Following IACS-010759 treatment in HD-MB03 G3 MB cells, we observed a significant decrease in the enrichment of many downstream gene targets that are positively regulated by MYC including: *NIP7* (log2 fold-change: −2.48; $p = 0.000538$), *GLS1* (log2 fold-change: −2.20; $p < 0.000001$), *AIMP2* (log2 fold-change: −2.19, $p = 0.000136$), *NOP56* (log2 fold-change: −2.15; $p = 0.000276$), *NOP16* (log2 fold-change: −1.73; $p = 0.000225$), *MRTO4* (log2 fold-change: −1.63; $p = 0.000919$), and more, corroborating a decline in MYC-dependent transcriptional activity (Fig. 3A). We validated the

downregulation of the MYC-target GLS1 following IACS-010759 treatment by immunoblotting in HD-MB03 cells (Fig. 3B). In addition to positively promoting the transcription of many genes, MYC can also modulate the levels of certain proteins through the regulation of non-coding RNAs that influence post-translational protein stability. It has been demonstrated that suppression of the wild-type (WT) form of tumor suppressor TP53 is mediated by the c-Myc-Inducible Long noncoding RNA Inactivating P53 (*MILIP*), which promotes TP53 polyubiquitination and degradation[33]. G3 MB tumors commonly maintain WT TP53, and we found that upon IACS-010759-mediated downregulation of MYC, *MILIP* transcript levels were also suppressed and this corresponded with an increase in WT TP53 protein levels, which in turn may offer an additional desirable therapeutic benefit for suppressing tumor growth (Fig. 3A, B). To conclusively demonstrate the importance of MYC downregulation in mediating the response of G3 MB cells to complex-I inhibition, we restored MYC expression in IACS-010759-treated HD-MB03 G3 MB cells using exogenous overexpression. Re-establishing MYC levels in IACS-010759-treated cells rescued the expression of multiple downstream MYC target genes, including *MILIP*, and the protein abundance of GLS1 (Fig. 3C, D). In contrast, the re-introduction of MYC repressed IACS-010759-mediated upregulation of TP53 levels (Fig. 3D). Re-instating MYC levels renewed the growth and restored the stemness capacity of HD-MB03 G3 MB cells following IACS-010759 treatment while suppressing differentiation, highlighting the importance of MYC regulation for the fate of G3 MB cells in response to complex-I inhibition (Fig. 3E–G). These unique findings offer a fresh perspective regarding the significance of MYC oncoprotein regulation in mediating responses to metabolic agents in G3 MB tumor cells.

### IACS-010759-mediated mitochondrial ROS production is responsible for decreasing MYC abundance in G3 MB cells

To further understand the impact of IACS-010759-mediated complex-I inhibition on G3 MB cells, MYC regulation and downstream MYC signaling, we confirmed the efficacy of IACS-010759 for targeting complex-I activity and overall mitochondrial respiratory capacity in G3 MB cells. Using Oroboros respirometry, we measured the maximal oxygen consumption rate (OCR) of intact G3 MB cells as well as substrate-specific OCR in permeabilized cells as a read-out of complex-specific ETC activity. Indeed, we found that a single 100 nM treatment of IACS-010759 effectively suppressed the respiratory capacity of G3 MB cells, as we observed a decrease in the basal and maximal OCR (Fig. 4A, B; Supplementary Fig. 4A, B). Furthermore, we observed that pyruvate and malate-dependent oxygen consumption was severely impaired, indicating inhibition of complex-I specific activity, which corresponded with a decrease in ATP production (Fig. 4A–C; Supplementary Fig. 4A–C). These findings confirm the efficacy of IACS-010759 treatment in impairing mitochondrial respiration in G3 MB cells.

To further understand the overall impact of IACS-010759 treatment on mitochondrial function, we monitored mitochondrial reactive

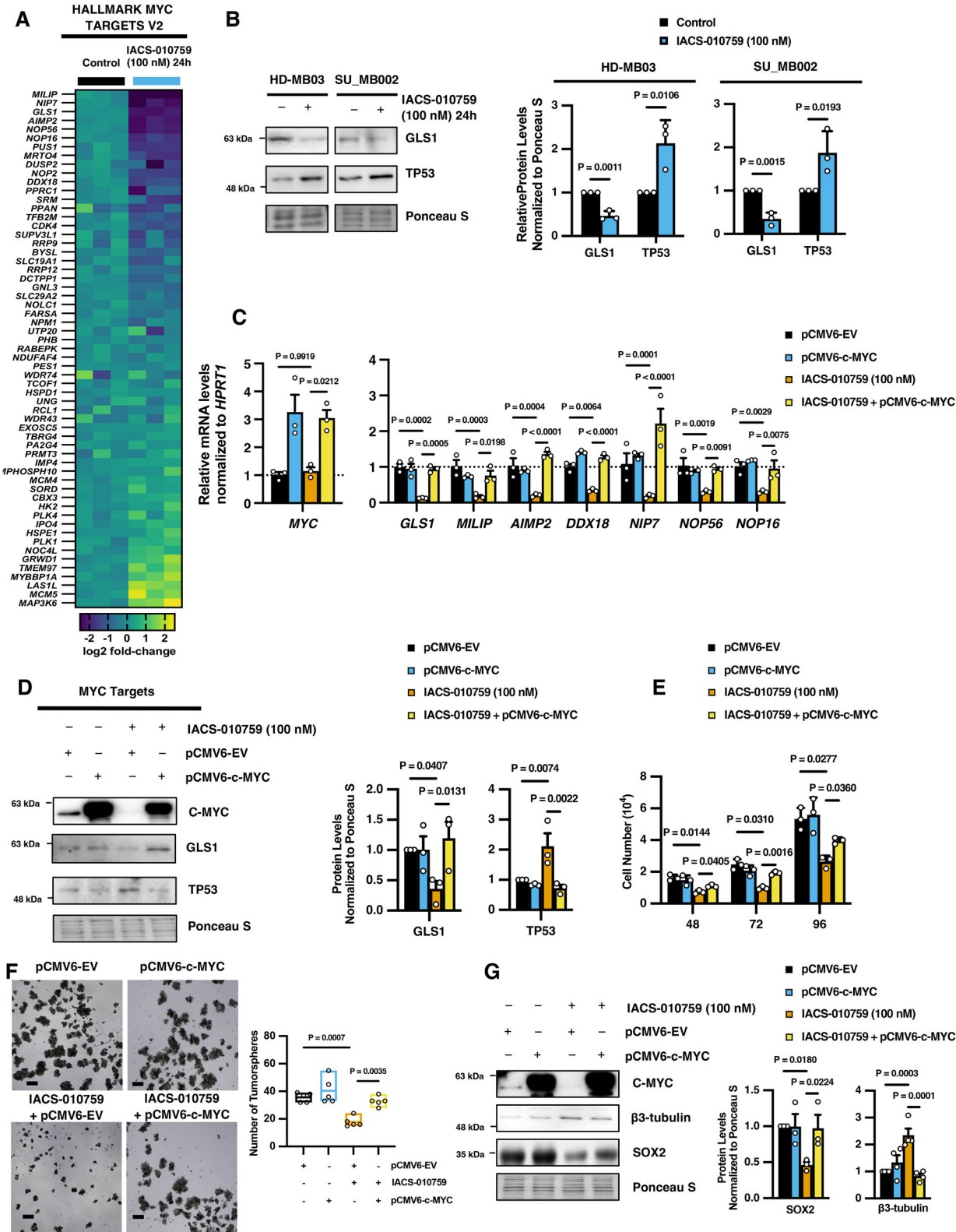

oxygen species (mROS) production, which can serve as a readout of mitochondrial health. Using the MitoSOX stain to detect mitochondrial superoxide anion species, we found that IACS-010759-treatment significantly elevated mROS production in HD-MB03 and SU_MB002 G3 MB cells, which could be suppressed by using the mitochondrial specific superoxide dismutase (SOD) mimetic, MitoTEMPO (Fig. 4D, E). ROS play a pivotal role in cellular signaling mechanisms, and mROS can influence extra-mitochondrial pathways by entering the cytosol

through the transient opening of the mitochondrial permeability transition pore (MPTP)[34]. Total cytosolic superoxide was measured using dihydroethidium (DHE), which becomes oxidized in the cytosol and translocates to the nucleus to generate red fluorescence. We confirmed that the mROS produced following IACS-010759 treatment contributes to overall cellular ROS accumulation as demonstrated by increased DHE oxidation, which was restored by blocking mitochondrial-specific ROS generation with MitoTEMPO (Fig. 4F). This

**Fig. 3 | IACS-010759-mediated suppression of MYC is important for the regulation of G3 MB tumor properties. A** Log2 fold-change in mRNA levels of MYC target genes in HD-MB03 cells treated with IACS-010759 for 24 hours. $N = 3$ experimental replicates. **B** Representative western blots and densitometry quantification of GLS1 and TP53 from HD-MB03 and SU_MB002 G3 MB cells treated with IACS-010759 for 24 hours. Graph represents mean values of densitometry quantification of blots from $N = 3$ experimental replicates ± SEM. $P$ values were calculated using two-sided student's $t$ test. **C, D** HD-MB03 cells were treated with IACS-010759 followed by exogenous overexpression of MYC for 48 hours and subjected to: **C** qRT-PCR analysis of selected MYC target genes, $P$ values were calculated using two-way ANOVA with Tukey's multiple comparisons. Data are presented as mean values of $N = 3$ experimental replicates ± SEM; **D** immunoblot analysis of TP53 and GLS1, graphs represent mean values of densitometry quantification of blots from $N = 3$ experimental replicates ± SEM, $P$ values were calculated using one-way ANOVA with Fisher's LSD test. **E** HD-MB03 cells were treated with IACS-010759 followed by exogenous overexpression of MYC for 48, 72, and 96 hours and subjected to cell count analysis. Data are presented as mean values of $N = 3$ experimental replicates ± SEM. $P$ values were calculated using two-way ANOVA with Tukey's multiple comparisons. **F, G** HD-MB03 cells were treated with 100 nM of IACS-010759 followed by exogenous overexpression of MYC and subjected to **F** tumorsphere formation analysis; representative images (Scale Bar = 100 μm) and total sphere number (>50 μm) presented as box-plot with bounds from minima to maxima and center line at mean from $N = 5$ experimental replicates where $P$ values were calculated using one-way ANOVA with Tukey's multiple comparisons, and **G** immunoblot analysis of SOX2 ($N = 3$ experimental replicates) and β3-tubulin ($N = 4$ experimental replicates), graphs represent mean values of densitometry quantification of blots ± SEM where $P$ values were calculated using one-way ANOVA with Fisher's LSD test. Source data provided in Source Data File.

release of mROS into the cytosol could be attributed to increased MPTP opening. To measure changes in mitochondrial membrane potential (MMP), we utilized the JC-1 stain. When MMP is high, JC-1 is sequestered in the mitochondria and forms red fluorescent J-aggregates. Under conditions of stress and increased mitochondrial permeability, MMP decreases and leads to dispersion of JC-1 in its green monomeric form. IACS-010759 treatment led to a significant decline in the red/green JC-1 ratio, indicating impaired MMP and increased mitochondrial permeability (Fig. 4G). MPTP opening was confirmed using the calcein stain, where cells are loaded with green fluorescent calcein that is normally sequestered in the mitochondria. Under conditions of MPTP opening, calcein can enter the cytosol, where $CoCl_2$ quenches the fluorescence. We found that IACS-010759 treatment decreased the green fluorescent calcein signal in HD-MB03 G3 MB cells, indicating increased opening of the MPTP (Fig. 4H).

Notably, scavenging mitochondrial superoxide by using two different mitochondrial antioxidant mimetics, MitoTEMPO and MnTmPyP, restored MYC protein levels and reversed the modulation of MYC targets GLS1 and TP53 following IACS-010759 treatment in HD-MB03 and SU_MB002 G3 MB cells (Fig. 4I-J; Supplementary Fig. 4D, E). Similarly, another antioxidant, glutathione (GSH), effectively restored MYC abundance in IACS-010759-treated G3 MB cells (Supplementary Fig. 4F). Scavenging mROS using MitoTEMPO similarly rescued MYC levels following IACS-010759 treatment in several cancer cell lines from ovarian, colorectal, and breast cancer (Supplementary Fig. 4G). These findings highlight a mechanism whereby ROS signaling can influence MYC levels in G3 MB and other cancers.

## IACS-010759 promotes MYC oxidation and proteasomal degradation in G3 MB cells

ROS signaling can influence cellular expression profiles through multiple mechanisms, including transcriptional regulation or post-translational modifications[35]. To understand how ROS may be regulating MYC levels in G3 MB cells, we began by monitoring the changes in MYC abundance over time during IACS-010759 treatment to determine if MYC downregulation is an early or late-stage response. Time-course analysis revealed that MYC protein levels decline as early as 3 hours and are almost entirely ablated following 24 hours of IACS-010759 treatment in HD-MB03 G3 MB cells (Fig. 5A; Supplementary Fig. 5A). In contrast, the mRNA transcript levels of *MYC* increased at early time points following IACS-010759 treatment before returning to baseline levels after 24 hours (Fig. 5B). The early depletion of MYC protein abundance corresponded with modulation of MYC-targets where *GLS1*/GLS1 mRNA and protein expression steadily declined while TP53 levels increased, indicating that the time-response of MYC protein downregulation correlates with a loss in MYC activity (Fig. 5A, B; Supplementary Fig. 5A).

We confirmed that IACS-010759 treatment functionally blocks complex I-dependent oxygen consumption after 3 hours using Oroboros respirometry, corresponding with a decrease in ATP production

(Supplementary Fig. 5B, C). Moreover, mitochondrial superoxide accumulation occurs as early as 3 hours following IACS-010759 treatment in HD-MB03 G3 MB cells (Fig. 5C). This corresponds with an increase in cytoplasmic ROS signal that can be restored by treatment with the mitochondrial antioxidant MitoTEMPO (Supplementary Fig. 5D). To further implicate the role of mROS production as an upstream molecular event influencing MYC stability and not just a consequence of MYC downregulation and ATP depletion, we found that restoration of MYC levels by exogenous overexpression was unable to restore the decrease in ATP caused by IACS-010759 treatment or the accumulation of mitochondrial superoxide (Supplementary Fig. 5E).

These findings suggest that IACS-010759-mediated ROS production may be decreasing the post-translational stability of MYC protein. Indeed, we found that IACS-010759 treatment significantly diminished MYC protein half-life as determined using the translation inhibitor cycloheximide (CHX) (Fig. 5D). We observed that MYC protein levels in IACS-010759 treated cells decreased by ~50% after only 30 minutes of translation inhibition, whereas the levels of MYC remained relatively stable in vehicle-treated control cells for up to 2 hours following CHX exposure (Fig. 5D). Similarly, we found that IACS-010759 treatment significantly reduced the half-life of MYC in SU_MB002 cells, where MYC levels remained stable for up to 30 minutes, and IACS-010759 treatment reduced the half-life of MYC to ~20 minutes (Supplementary Fig. 5F). Altogether these findings indicate that IACS-010759 treatment increases MYC protein turnover.

The ubiquitin-proteasome system (UPS) is the primary mechanism responsible for the post-translational degradation of proteins. Therefore, we postulated that IACS-010759 treatment might be increasing MYC protein turnover via ubiquitin-targeted proteasomal degradation. Immunoprecipitation of MYC followed by immunoblotting for polyubiquitin modifications confirmed that IACS-010759 treatment increased the specific ubiquitination of MYC (Fig. 5E). To further decipher the role of the UPS in IACS-010759-mediated MYC degradation, we utilized two different proteasome inhibitors, MG-132 and bortezomib. MG-132 is an inhibitor of the classic 26 S proteasome, whereas bortezomib explicitly inhibits the 20 S proteasome core, which can function as an independent unit or part of the 26 S proteasome. We found that treatment with either MG-132 or bortezomib led to the accumulation of MYC protein in control cells, in line with basal rates of MYC protein turnover (Fig. 5F, G). Inhibition of the proteasome using both MG-132 and bortezomib completely restored MYC levels at 3 hours post-IACS-010759 treatment (Fig. 5F, G). However, after prolonged IACS-010759 treatment for 24 hours, inhibition of the 26 S proteasome using MG-132 could not rescue the accumulation of MYC protein in IACS-010759 treated cells to the same extent as MG-132-treated control cells (Supplementary Fig. 5G). Whereas 20 S proteasome inhibition using bortezomib treatment was sufficient to completely restore MYC levels in IACS-010759 treated cells to the same extent as bortezomib-treated control cells after 24 hours

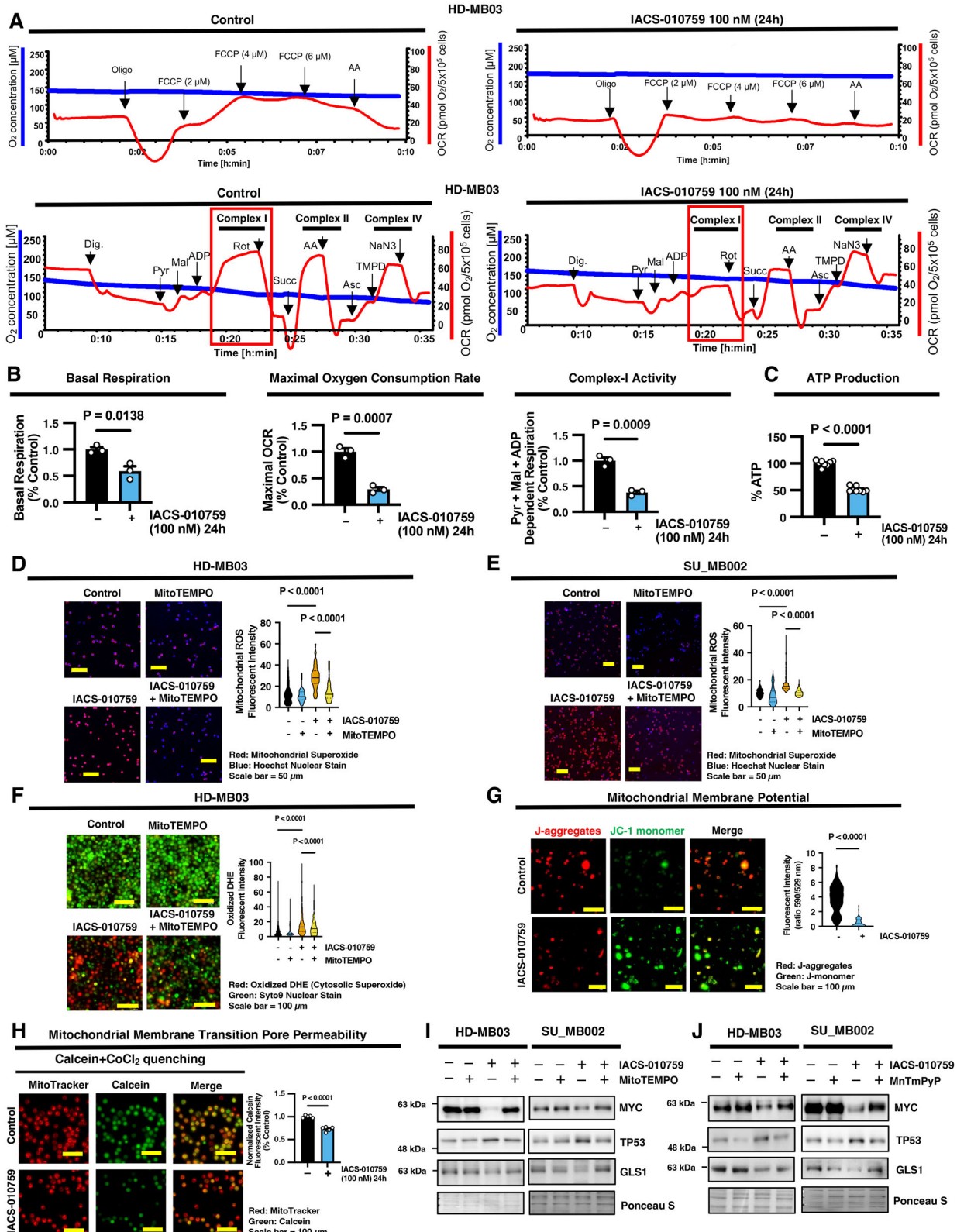

(Supplementary Fig. 5H). Under conditions of prolonged oxidative stress or energy depletion, the 26 S proteasome can dissociate into the 20 S proteasome core and mediate the degradation of oxidized and ubiquitinated proteins[36]. Therefore, we questioned whether IACS-010759-mediated ROS production might be promoting the oxidation and subsequent proteasomal degradation of MYC.

Cysteine is one of the amino acid residues most susceptible to oxidation, and MYC harbors ten cysteine residues. When cysteine residues are not oxidized, they are present in their reduced thiol form. To monitor changes in cysteine oxidation of MYC, we used a well-characterized method involving the precipitation of reduced cysteine residues using streptavidin-agarose beads followed by immunoblotting to detect changes in the levels of reduced MYC protein (Fig. 5H)[37]. We found that IACS-010759 treatment significantly decreased the proportion of reduced MYC protein in both HD-MB03 and SU_MB002 G3 MB cells, indicating that IACS-010759 is promoting MYC oxidation

**Fig. 4 | IACS-010759 blocks complex-I and promotes reactive oxygen species production. A**–**C** HD-MB03 cells were treated with IACS-010759 (24 hours) and subjected to **A** Oroboros respirometry with a representative tracing of maximal and substrate-specific oxygen consumption rate (OCR), **B** quantifications of basal, maximal, and pyruvate-malate dependent OCRs ($N = 3$ experimental replicates) and **C** ATP levels ($N = 7$ experimental replicates) presented as mean ± SEM. $P$ values were calculated using two-sided student's $t$ test. **D**, **E** Representative images (Scale Bar = 50 μm) of **D** HD-MB03 and **E** SU_MB002 cells treated with IACS-010759 and/or MitoTEMPO for 24 hours and labeled with MitoSOX stain (red) and Hoechst (blue). Violin plot represents the quantification of fluorescent intensity per cell of $N > 40$ cells from $N = 3$ experimental replicates with solid line at mean and dashed lines at quartiles. $P$ values were calculated using two-way ANOVA with Tukey's multiple comparisons. **F** Representative images (Scale Bar = 100 μm) of HD-MB03 cells treated with IACS-010759 and/or MitoTEMPO for 24 hours and labeled with DHE (red) and Syto9 stain (green). Violin plot represents the quantification of

fluorescent intensity per cell of $N > 100$ cells from $N = 3$ experimental replicates with solid line at mean and dashed lines at quartiles. $P$ values were calculated using two-way ANOVA with Tukey's multiple comparisons. **G** Representative images (Scale Bar = 100 μm) of HD-MB03 cells treated with IACS-010759 and loaded with JC-1 dye (red = J-aggregates; green = J-monomer). Violin plot represents the quantification of red/green fluorescent ratio per cell of $N > 100$ cells from $N = 3$ experimental replicates with solid line at mean and dashed lines at quartiles. $P$ values were calculated using two-sided student's $t$ test. **H** Representative images (Scale Bar = 100 μm) of HD-MB03 cells treated with IACS-010759 for 24 hours and loaded with calcien dye (green), CoCl₂, and MitoTracker (red). Graph represents the mean ± SEM fluorescent intensity from $N = 5$ experimental replicates. $P$ values were calculated using two-sided student's $t$ test. **I**, **J** Representative western blots of MYC, TP53, and GLS1 from $N = 3$ experimental replicates of HD-MB03 and SU_B002 G3 MB cells treated with IACS-010759 along with either **I** MitoTEMPO or **J** MnTmPyP for 24 hours. Source data provided in Source Data File.

(Fig. 5I, J). In contrast, the oxidation of another MB oncogene, OTX2, did not change following IACS-010759 treatment, highlighting the potential specificity of this response for MYC regulation (Supplementary Fig. 5I). Furthermore, this regulation of MYC oxidation by IACS-010759 treatment appears to be conserved across different cancer types, as we observed a similar response in A2780 ovarian carcinoma cells (Supplementary Fig. 5J). These findings identify an oxidative post-translational modification of the MYC protein that may have important physiological and pathological roles in regulating MYC stability.

### Selective cysteine residues are responsible for MYC protein oxidation and degradation following IACS-010759 treatment

To firmly establish that ROS induction is responsible for MYC oxidation and degradation following IACS-010759 treatment, we found that scavenging mitochondrial ROS with MitoTEMPO restored the levels of reduced MYC protein, indicating suppression of MYC oxidation, which corresponded with ablation in the specific ubiquitination of MYC (Fig. 6A, B). The functional importance of this ROS-mediated regulation of MYC is exemplified by the finding that MitoTEMPO treatment restored the stemness capacity and suppressed differentiation in HD-MB03 G3 MB cells following IACS-010759 treatment (Fig. 6C; Supplementary Fig. 6A).

To further characterize this oxidative post-translational modification of MYC, we generated c-terminal GFP-tagged MYC-expressing plasmid constructs with individual point mutations in all 10 cysteine residues to determine the sites which are responsible for MYC oxidation and degradation and confirmed similar overexpression efficiency of all constructs (Fig. 6D; Supplementary Fig. 6B). Cysteine residues were substituted to glycine, and we leveraged these mutant constructs to determine the susceptibility of individual cysteine residues towards oxidation using the well-characterized biotin-switch assay (Fig. 6E, F)[38–41]. In this assay, reduced cysteine thiols are blocked with a non-labeled alkylating agent (N-ethylmaleimide, NEM) followed by reduction of oxidized residues using a strong reducing agent (Tris(2-carboxyethyl)phosphine hydrochloride; TCEP; Fig. 6F). The newly reduced thiols, which were originally subjected to oxidation, were labeled with a biotin-conjugated alkylating agent (Maleimide-PEG2-Biotin; Fig. 6F). The GFP-tagged exogenous mutant MYC constructs can be immunoprecipitated and changes in their oxidation status can be monitored by blotting and detection by chemiluminescence using streptavidin-HRP (Fig. 6F). If a particular cysteine residue is normally susceptible to oxidation following IACS-010759 treatment, then we should observe less biotin labeling and decreased chemiluminescent signal when that residue is mutated to glycine as compared to the WT MYC control construct. Using this assay, we confirmed that WT MYC is undergoing oxidation following IACS-010759 treatment as we observed a ~2-fold increase in chemiluminescent biotin-labeling signal (Supplementary Fig. 6C). Moreover, we found that the majority of MYC

cysteine residues were responsible for a proportion of MYC oxidation following IACS-010759 treatment, as eight of the cysteine mutant constructs (C85G, C132G, C148G, C186 G, C203G, C315G, C357G, and C453G) displayed significantly decreased biotin-labeling potential as compared to the WT-control (Fig. 6G(i, ii)).

To determine which cysteine residues play a role in mediating MYC degradation, we overexpressed GFP-tagged MYC constructs in HD-MB03 cells containing either wild-type MYC or individual cysteine mutants, and then subjected cells to IACS-010759 treatment. We then performed immunoblotting for GFP to detect only exogenous MYC protein to determine how individual cysteine residues impact MYC stability and degradation. We confirmed that exogenous WT MYC protein is efficiently degraded following IACS-010759 treatment in HD-MB03 cells. We found that a total 5 mutant cysteine MYC constructs (C148G, C203G, C315G, C357G and C453G) significantly impaired degradation potential following IACS-010759 treatment, whereas the other 5 residues had no significant impact on MYC degradation (C40G, C85G, C132G, C186G, and C223G) (Fig. 6H(i, ii)). Additionally, findings from global cysteine oxidation proteomics analysis performed by two separate groups confirmed that MYC protein is susceptible to cysteine oxidation under a variety of different oxidative stressors[42,43]. These studies identified C85, C315, and C357 as potential cysteine oxidation sites (Fig. 6I)[42,43]. The difference in oxidative stress inducing agents used in these studies combined with the lower sensitivity of proteomics analysis to identify only the most abundant peptides along with the fact that certain digestion protocols may not allow all MYC cysteine residues to be covered by mass spectrometry, could explain why only a small fraction of MYC cysteine residues were identified in these analyses. Taken together, our findings using the biotin-switch assay along with independent proteomics analysis confirms the susceptibility of MYC cysteine residues to undergo oxidative post-translational modifications. Moreover, our mutational analysis pinpoints the exact cysteine residues that are responsible for oxidation-mediated degradation of MYC following IACS-010759 treatment.

### IACS-010759 treatment leads to the acetylation and inactivation of SOD2 through modulation of pyruvate metabolism dynamics

To understand how IACS-010759 treatment promotes mROS accumulation, we focused on assessing whether the mitochondrial antioxidant system is functioning properly. In the mitochondria, superoxide anion is converted into the less reactive hydrogen peroxide ($H_2O_2$) by the mitochondrial manganese superoxide dismutase (MnSOD, also known as SOD2). Therefore, we monitored the effect of IACS-010759 treatment on SOD2 levels and activity. We found that IACS-010759 treatment did not alter SOD2 levels in HD-MB03 and SU_MB002 G3 MB cells (Fig. 7A). However, SOD2 activity is strongly influenced by post-translational modifications, mainly acetylation. The acetylation of SOD2 within its catalytic domain at lysines 68 and 122 inhibits its antioxidant activity[44,45]. Despite SOD2 levels remaining

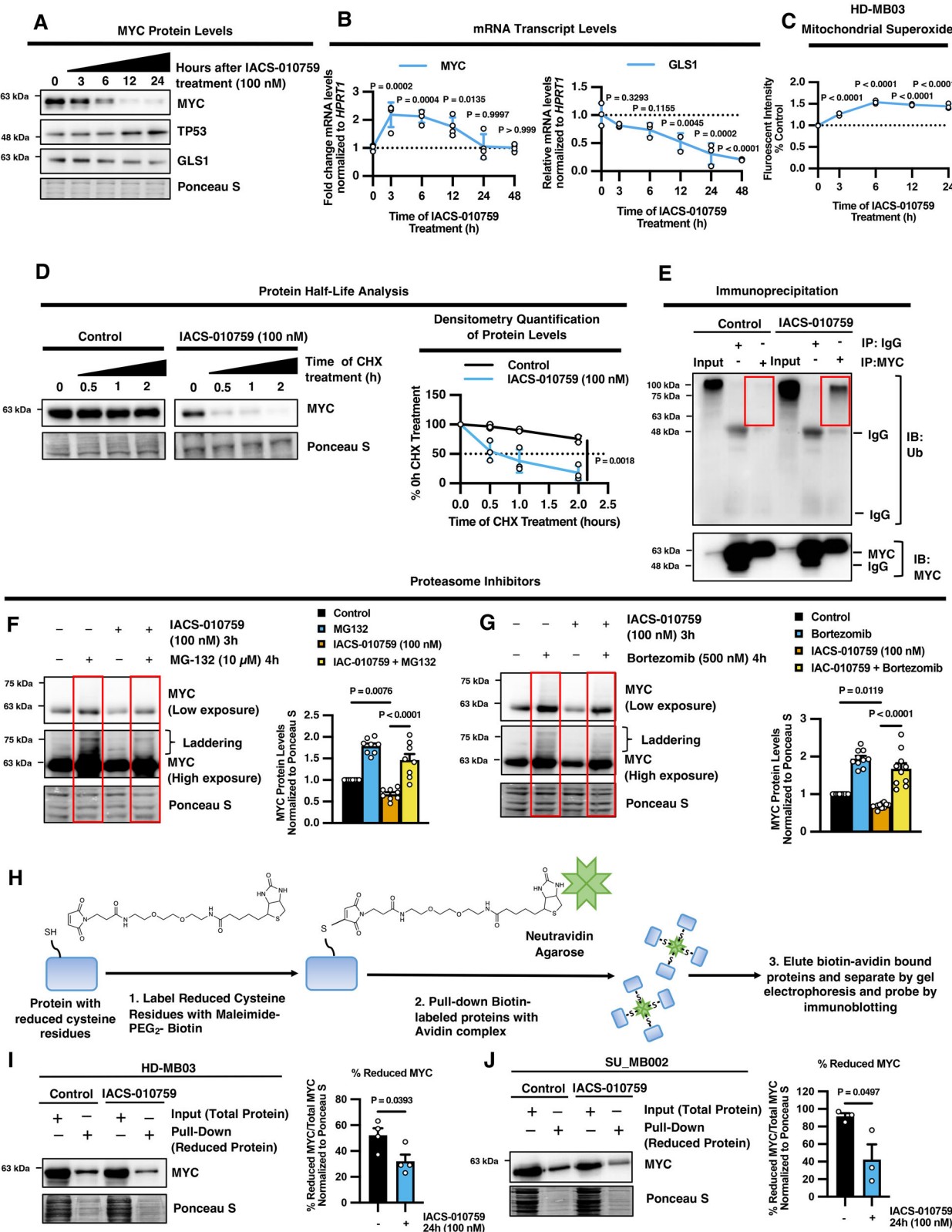

stable, we found that the acetylation of SOD2 at K68 and K122 was markedly enhanced following IACS-010759 treatment in both HD-MB03 and SU_MB002 G3 MB cells, indicating an impairment in SOD2 antioxidant enzymatic activity (Fig. 7A). To further implicate the importance of SOD2 activity in the regulation of mitochondrial superoxide levels and MYC protein abundance, we transiently silenced SOD2 using siRNA. We found that impairment of SOD2 significantly

promoted mROS accumulation and consequently decreased MYC levels while also modulating MYC targets GLS1 and TP53 (Supplementary Fig. 7A, B).

Protein acetylation can be influenced by metabolism through the regulation of pyruvate dynamics. Pyruvate can be transported into the mitochondria via the mitochondrial pyruvate carrier (MPC), where it is converted to acetyl-CoA by pyruvate dehydrogenase (PDH). PDH

**Fig. 5 | MYC undergoes protein oxidation and proteasomal degradation following complex I inhibition in G3 MB cells. A**–**C** HD-MB03 cells were treated with IACS-010759 for indicated time points and subjected to: **A** immunoblot analysis of MYC, TP53 and GLS1, representative blots from $N = 3$ experimental replicates; **B** qRT-PCR analysis for *MYC* and *GLS1* mRNA levels presented as mean ± SEM from $N = 3$ experimental replicates, and **C** MitoSOX stain, graph represents quantification of fluorescent intensity using a plate reader and normalized to cell number, presented as mean values ± SEM from $N = 3$ experimental replicates where *P* values were calculated using one-way ANOVA with Dunnett's multiple comparisons test relative to control. **D** Representative western blots of MYC in HD-MB03 cells treated with IACS-010759 for 24 hours followed by the translation inhibitor cycloheximide (CHX) for 0.5, 1, and 2 hours. Graphs represents densitometry quantification of blots presented as mean ± SEM from $N = 3$ experimental replicates where *P* values were calculated at final time point using two-sided student's *t* test. **E** Representative western blot of polyubiquitin modifications following immunoprecipitation of

MYC or IgG negative control isotope antibody in HD-MB03 lysates treated with IACS-010759 compared to input lysate (no immunoprecipitation) from $N = 3$ experimental replicates. **F**, **G** HD-MB03 cells were treated with IACS-010759 for 3 hours followed by the proteasome inhibitors **F** MG-132 ($N = 8$ experimental replicates) or **G** Bortezomib ($N = 10$ experimental replicates) for 4 hours and subjected to immunoblot analysis for MYC. Graphs represents densitometry quantification of blots presented as mean ± SEM. *P* values were calculated using one-way ANOVA with Fisher's LSD test. **H** Schematic diagram depicting the protocol for oxidation assay. **I** HD-MB03 ($N = 4$ experimental replicates) and **J** SU_B002 ($N = 3$ experimental replicates) cells were treated with IACS-010759 and subjected to western blot of maleimide-PEG2 biotin-labeled reduced MYC levels precipitated with streptavidin-agarose relative to input lysates (no precipitation). Graphs represent densitometry quantification presented as mean ± SEM. *P* values were calculated using two-sided student's *t* test relative to vehicle control. Source data provided in Source Data File.

activity is in turn regulated by phosphorylation, where phosphorylation of Ser293 by pyruvate dehydrogenase kinases (PDKs) inhibits its catalytic activity. To understand whether IACS-010759 treatment may be enhancing SOD2 acetylation through the modulation of pyruvate dynamics, we probed the key enzymes involved in this process. We observed that the MPC components, MPC1 and MPC2, were elevated following IACS-010759 treatment in HD-MB03 and SU_MB002 G3 MB cells, indicating increased import of pyruvate into the mitochondria (Fig. 7B). In addition, we found that there was less activation of the PDH inhibitor PDK1 (as demonstrated by a decrease in the activating autophosphorylation site Ser241) which corresponded with less inhibitory phosphorylation of PDH at Ser293 decreased following IACS-010759 treatment, which is established to promote the conversion of pyruvate into acetyl-CoA (Fig. 7B; Supplementary Fig. 7C). To conclusively determine whether these changes in enzymes related to pyruvate metabolism truly enhance the shuttling and processing of pyruvate towards acetyl-CoA production, we performed heavy-labeled isotopologue tracing of $U^{13}C_3$-pyruvate by mass spectrometry. We observed decreased $^{13}C$ isotopologue labeling of lactate derived from pyruvate which corresponded with an increase in the abundance of $^{13}C$-labeled acetyl-CoA in IACS-010759 treated G3 MB cells as compared to controls, indicating increased shuttling of pyruvate to Acetyl-CoA via enhanced PDH activity (Fig. 7C). Furthermore, we observed enhanced $^{13}C$-labeled isotopologues of several TCA cycle intermediates including citrate, fumarate, and malate (Fig. 7D). Importantly, we observed that these changes in pyruvate metabolism dynamics correspond with an increase in total cellular acetyl-CoA pools following IACS-010759 treatment (Fig. 7E). These findings implicate an important role for the regulation of pyruvate dynamics following IACS-010759 treatment in influencing acetyl-CoA production and protein acetylation (Fig. 7F). To determine the sequence of these events in relation to MYC downregulation, we monitored the timeline of changes in pyruvate metabolic enzymes following IACS-010759 treatment. We observed that MPC1 and MPC2 levels increase while the phosphorylation of PDH decreases at early time points following IACS-010759 treatment, in line with the downregulation of MYC abundance (Fig. 7G and Supplementary Fig. 7D; Fig. 5A). Altogether, these findings demonstrate that IACS-010759 treatment leads to alterations in pyruvate metabolism dynamics and increases inhibitory acetylation of the mitochondrial antioxidant SOD2.

### Inhibition of mitochondrial pyruvate import blocks SOD2 acetylation and mitochondrial ROS production, restoring MYC expression in IACS-010759 treated G3 MB cells

To place this alteration in pyruvate dynamics as an upstream molecular event that is not just an outcome of MYC downregulation, we found that the levels of MPC1 and MPC2 remained enhanced and PDH remained activated in IACS-010759-treated HD-MB03 G3 MB cells even after the restoration of MYC by exogenous overexpression

(Supplementary Fig. 8A). In contrast, blocking mitochondrial pyruvate import using the MPC inhibitor UK-5099 restored MYC levels and downstream MYC-targets GLS1 and TP53 (Fig. 8A). To highlight the importance of this shift in pyruvate metabolic dynamics for influencing protein acetylation and SOD2 activity, we found that inhibition of MPC using UK-5099 repressed the activation of PDH, suppressed the accumulation of acetyl-CoA, and ultimately blocked the specific acetylation of SOD2 in IACS-010759-treated G3 MB cells (Fig. 8A–C; Supplementary Fig. 8B, C). This restoration of SOD2 activity following MPC inhibition in IACS-010759-treated cells alleviated the total oxidative stress and mROS production in HD-MB03 G3 MB cells (Fig. 8D). Furthermore, this suppression of oxidative stress ultimately restored the reduced levels of MYC, indicating a decrease in MYC oxidation, and decreased the specific ubiquitination of MYC, which corresponds with the rescue in MYC protein abundance observed following MPC inhibition in IACS-010759-treated cells (Fig. 8A, E, F). The significance of this regulation in pyruvate dynamics for the response to IACS-010759 treatment is underscored by the observation that UK-5099 restored the tumorsphere formation ability of HD-MB03 G3 MB cells and rescued the levels of the stemness factor SOX2 while suppressing the differentiation marker β3-tubulin/TUBB3 following IACS-010759 treatment (Fig. 8G–H). Taken together, these findings implicate a role for the MPC in modulating MYC stability.

### Oral administration of IACS-010759 impairs tumor growth and prolongs survival in a pre-clinical orthotopic G3 MB brain tumor xenograft model

To assess the clinical implications of utilizing IACS-010759 to treat G3 MB tumors, we implemented a pre-clinical orthotopic intracerebellar xenograft mouse model. The HD-MB03 G3 MB line is a well-established xenograft model for studying highly aggressive G3 MB tumors in vivo[24,46,47]. When transplanted into the brains of NOD-SCID gamma (NSG) mice, HD-MB03 cells develop tumors that resemble the large-cell/anaplastic (LCA) histomorphology, with numerous mitoses and apoptotic bodies, commonly exhibited in many G3 MB patient tumors (Supplementary Fig. 9A). HD-MB03 xenograft tumors also maintain high MYC-positive staining similar to patient G3 MB tumors with amplified MYC (Supplementary Fig. 9B)[48]. We first evaluated the safety of our IACS-010759 treatment regimen in non-tumor-bearing NSG mice, and we found that oral gavage administration of either 5 or 7.5 mg/kg of IACS-01759 in 0.5% methylcellulose for five consecutive days followed by a two-day treatment holiday for 6 weeks was well-tolerated by animals, as indicated by no significant change in their weight over the treatment course (Fig. 9A). To test the efficacy of IACS-010759 for treating G3 MB tumors, $1 \times 10^5$ HD-MB03 cells were implanted into the cerebellum of NSG mice by stereotactic injection (Fig. 9B). We observed small tumors visible by MRI at 5-days post-surgery which were even more clear by H&E staining of tumor tissues collected at this time point (Fig. 9B). Tumors continue to grow and are

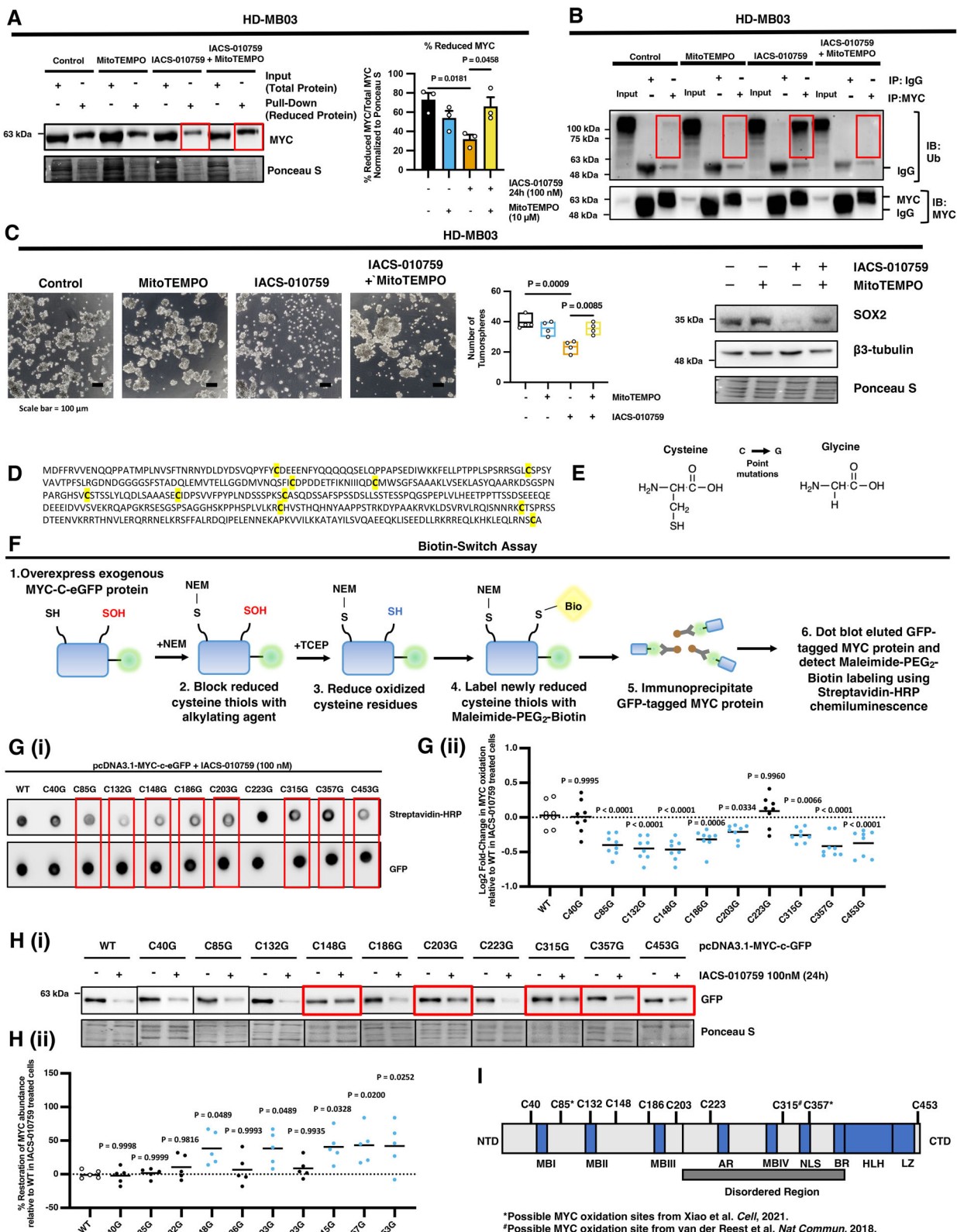

even more well-formed at 10 days post-surgery, therefore we divided animals into two cohorts to evaluate the efficacy of IACS-010759 treatment as an early intervention starting treatment at 5-days post-op or as a treatment in well-established tumors starting at 10 days post-op (Fig. 9B). Tumor-bearing mice were treated with either placebo control or 7.5 mg/kg of IACS-010759 by oral gavage on a schedule of 5 days on, 2 days off for up to 6 weeks (Fig. 9B). We found that animal survival was

significantly prolonged in mice treated with IACS-010759 starting at either 5 or 10 days post-op in this highly aggressive tumor model (HR: 4.373; $p = 0.002$ and HR: 4.351; $p = 0.0015$; Fig. 9C).

Using T2 MRI imaging At 18 days post-op, we found that all control animals had developed large tumor masses in the cerebellum (average 19.36 mm³), whereas tumors from IACS-010759-treated animals were markedly smaller (average 1.852 mm³) (Fig. 9D(i, ii); Supplementary

**Fig. 6 | Specific cysteine residues mediate MYC oxidation and degradation in IACS-010759 treated G3 MB cells. A−C** HD-MB03 cells were treated with IACS-010759 and/or MitoTEMPO for 24 hours and subjected to: **A** western blot of maleimide-PEG2 biotin-labeled reduced MYC levels precipitated with streptavidin-agarose relative to input lysates (no precipitation) with densitometry quantification from $N = 3$ experimental replicates presented as mean ± SEM, where $P$ values were calculated using two-way ANOVA with Tukey's multiple comparisons; **B** polyubiquitin modifications following immunoprecipitation of MYC or IgG negative control from $N = 3$ experimental replicates; **C** tumorsphere formation analysis with representative images (Scale Bar = 100 μm) and total sphere number (>50 μm) presented as box-plot with the box limits at minima and maxima and center line at mean from $N = 4$ experimental replicates where $P$ values were calculated using one-way ANOVA with Tukey's multiple comparisons; and representative western blots of SOX2 and β3-tubulin from $N = 3$ experimental replicates. **D** MYC protein sequence with cysteine residues highlighted. **E** Schematic of the structural changes with a substitution of cysteine to glycine mutation. **F** Schematic diagram depicting the biotin-switch assay protocol. **G** (i) Dot blot analysis of oxidized immunoprecipitated exogenous pcDNA3.1-MYC-c-eGFP MYC protein containing either wild-type MYC or point mutations in one of the 10 individual cysteine residues following 24 hours of IACS-010759 (100 nM) treatment. (ii) Densitometry quantification of $N = 8$ experimental replications presented as Log2 fold-change relative to wild-type controls with line at mean. $P$ values were calculated using one-way ANOVA with Dunnett's multiple comparisons. **H** (i) Western blot analysis of exogenous pcDNA3.1-MYC-c-eGFP MYC protein containing either wild-type MYC or point mutations in one of the 10 individual cysteine residues following 24 hours of IACS-010759 (100 nM) treatment. (ii) Densitometry quantification of $N = 5$ experimental replications presented as % restoration of MYC protein levels relative to IACS-010759-treated wild-type controls with line at mean. $P$ values were calculated using one-way ANOVA with Dunnett's multiple comparisons. **I** Schematic diagram of MYC protein regions and cysteine sites and possible MYC oxidation sites identified by Xiao et al.[42] and van der Reest et al.[43]. Source data provided in Source Data File.

Fig. 9C). Moreover, Hematoxylin and eosin (H&E) staining of brain tumor tissues collected at endpoint demonstrated that control mice had a higher tumor burden than IACS-010759-treated mice (Fig. 9D(i, ii)). Importantly, immunohistochemistry (IHC) analysis confirmed that IACS-010759-treated tumors harbored lower MYC levels than placebo controls (Fig. 9E). IACS-010759-treated tumor cells were also less proliferative, as indicated by lower expression of Ki67 compared to control tumors (Fig. 9E). In line with our in vitro tumorsphere formation assays, levels of the stemness marker SOX2 were abolished in IACS-010759-treated tumors which corresponded with an increase the differentiation marker β3-tubulin (Fig. 9E). These findings highlight the therapeutic potential of this agent as a treatment option to improve outcomes for MB patients.

### IACS-010759 treatment modulates the MPC-SOD2-MYC signaling axis in group 3 MB brain tumors in vivo

To validate the effect of IACS-010759 treatment on MYC-regulated signaling in G3 MB tumors in vivo, we assessed the levels of downstream MYC-targets, GLS1 and TP53, using IHC. Indeed, in line with the lowered abundance of MYC in IACS-010759-treated HD-MB03 tumors (Fig. 9E) we also observed that GLS1 protein expression was strongly depleted while TP53 levels were significantly elevated in IACS-010759-treated tumors compared to placebo controls (Fig. 10A), indicating and impairment in MYC activity.

Importantly, IACS-010759-treated HD-MB03 tumors also exhibited increased oxidative stress as demonstrated using two distinct markers that detect DNA oxidation (8-hydroxy-2-deoxyguanosine; 8-oxo-DG) and lipid peroxidation (4-hydroxy-2-noneal; 4-HNE) (Fig. 10B). Moreover, we confirmed that IACS-010759 treatment enhanced MPC1 and MPC2 levels, while the abundance of phosphorylated PDH was decreased in IACS-010759-treated HD-MB03 xenograft tumors (Fig. 10C). These changes in pyruvate metabolism signaling corresponded with increased levels of acetylated SOD2 K68 (there are no commercially available acetylated SOD2 K122 antibodies suitable for IHC) (Fig. 10C), indicating impaired SOD2 antioxidant activity which complements the increase in oxidative stress observed in IACS-010759-treated HD-MB03 tumors (Fig. 10B). These findings highlight the importance of this MPC-SOD2-MYC signaling axis in the response of group 3 MB tumors towards IACS-010759 treatment.

Taken together, this study unveils a metabolic vulnerability underpinning the regulation of the post-translational stability of MYC that can be therapeutically exploited for treating aggressive pediatric G3 MB brain tumors. A schematic diagram summarizing the key mechanisms uncovered in this study can be found in Fig. 10D.

### Discussion

MB brain tumors remain a leading cause of cancer-related mortality in children[1,2]. Moreover, over 2/3 of childhood MB brain tumor survivors suffer from life-long neurological sequelae due to the cytotoxic chemotherapy and radiation treatments they received[8]. To develop safer and more effective treatments for childhood cancers, there is a need to identify therapies that can target molecules which are unique to the cancer cells and do not harm heathy tissues. MYC is one of the most frequently dysregulated oncogenes across all human cancers and is well-known to drive poor patient outcomes[11]. In childhood MB brain tumors, patients in the G3 subgroup commonly exhibit MYC amplification/activation, and these patients experience the worst survival prognosis of <40%[6]. MYC's ubiquitous functions and disordered structure have limited our ability to effectively target this oncogene for cancer therapy, which has led to it being branded as 'undruggable'[11]. Despite this, substantial efforts have been exhausted to adopt innovative strategies for overcoming these challenges to developing MYC inhibitors. Some of these strategies include blocking *MYC* mRNA expression using antisense oligonucleotides, but cancer cells can overcome these mechanisms by stabilizing MYC protein[11]. In recent years, the notion of targeting MYC by promoting its proteasomal degradation has gained traction, although Proteolysis-targeting chimeras (PROTACs) directly targeting MYC have yet to be developed[11]. The Omomyc mini-mutant protein has been shown to promote MYC proteasomal degradation and has demonstrated promising results in phase-I clinical trials[11]. However, most efforts for designing clinically applicable small molecule inhibitors of MYC aim to block the interaction between MYC and its obligate binding partner MAX to inhibit MYC-dependent transcriptional activity[11]. Still, many of these compounds are ineffective due to low in vivo potency, poor pharmacokinetic properties, or intolerable off-target effects. Here, we evaluated one of these agents, 10058-F4, against G3 MB cells and normal brain cell populations, and we also found that it displays significant toxicity towards normal human astrocytes, making such a drug unsuitable for treating childhood brain tumors. Therefore, we pursued an unconventional concept of inhibiting MYC indirectly by targeting cellular metabolism in G3 MB.

Given that MYC is a global regulator of cellular metabolic networks, we postulated that metabolism might also play a reciprocal role in helping maintain enhanced MYC abundance in MYC-activated G3 MB tumors. We demonstrate that complex-I inhibitors deplete MYC levels in several G3 MB cell models without affecting the proliferation or MYC abundance in normal human astrocytes or NSCs. We tested an orally bioavailable, BBB penetrable complex-I targeting agent, IACS-010759, in an intracerebellar orthotopic G3 MB xenograft model and found that this treatment modality is well-tolerated and significantly effective at prolonging animal survival by suppressing tumor growth and burden. IACS-010759-treated G3 MB xenografts harbor lower MYC levels, corresponding with a decrease in markers of stemness and proliferation and increased differentiation. Our in-depth mechanistic characterization revealed that IACS-010759 treatment regulates MYC

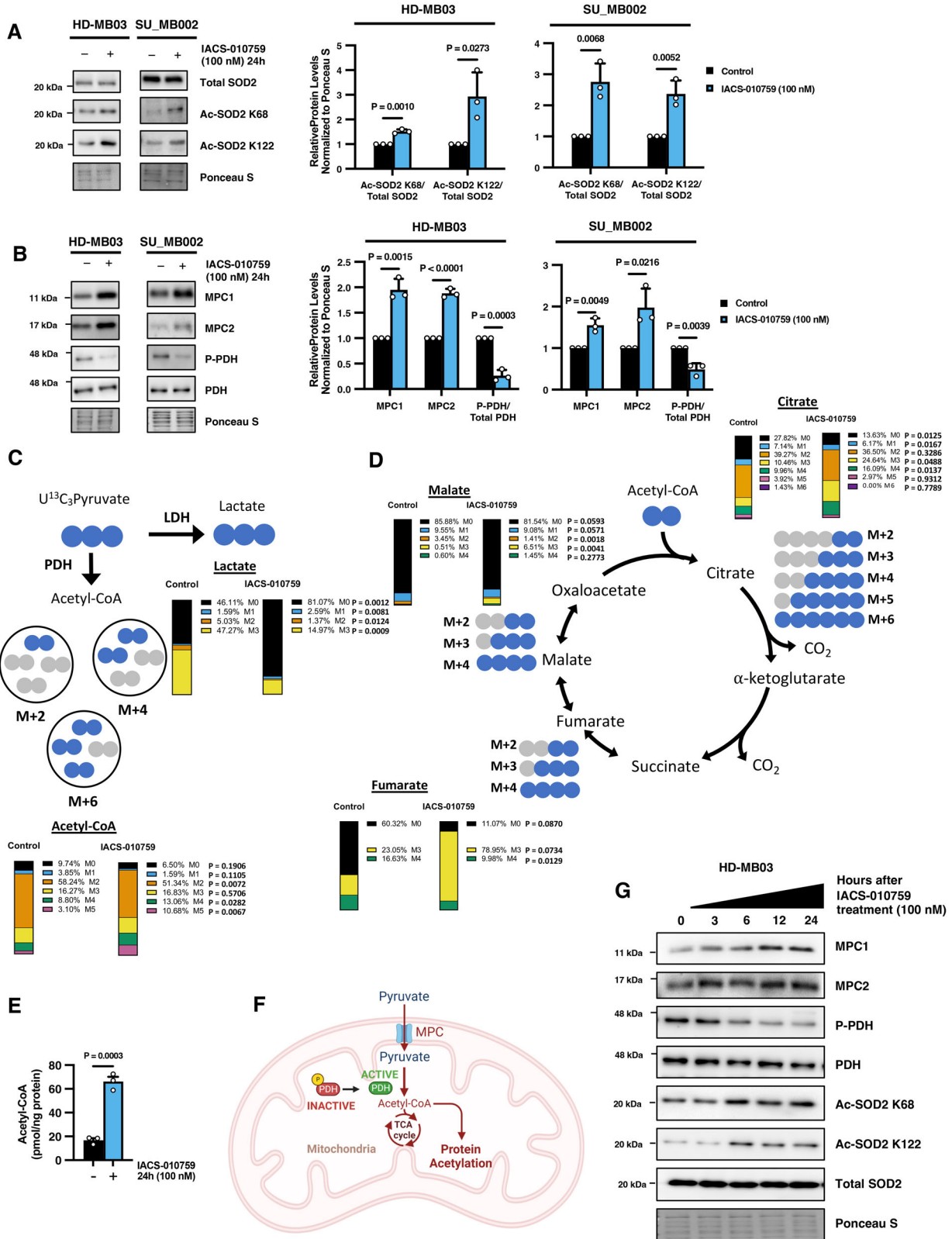

at the protein level by promoting MYC oxidation at cysteine residues C85, C132, C148, C186, C203, C315, C357, and C453 where the sites C148, C203, C315, C357, and C453 are ultimately responsible for enhancing MYC proteasomal degradation. Cysteine oxidation has been shown to be an important signaling mechanism mediated by ROS[35]. It has previously been demonstrated that many metabolic enzymes are susceptible to cysteine oxidation under conditions of oxidative stress[43]. The availability of multiple cysteine residues and the conformation of MYC may play a role in its specific susceptibility towards oxidation following complex-I inhibition as compared to other transcription factors such as OTX2. In a therapeutic context, our findings highlight that IACS-010759-mediated oxidation and degradation of MYC could be beneficial for inhibiting MYC levels in virtually all MYC-activated tumors, including those with genomic amplification,

**Fig. 7 | IACS-010759 treatment influences pyruvate metabolic dynamics in G3 MB cells. A** Representative western blots and densitometry quantifications presented as mean ± SEM of SOD2, acetylated SOD2 lysine 68 (Ac-SOD2 K68), and acetylated SOD2 lysine 122 (Ac-SOD2 K122) from $N = 3$ experimental replicates of HD-MB03 and SU_MB002 G3 MB cells treated with IACS-010759. $P$ values were calculated using two-sided student's $t$ test relative to vehicle control.
**B** Representative western blots and densitometry quantifications presented as mean ± SEM of mitochondrial pyruvate carriers 1 and 2 (MPC1 and MPC2), phosphorylated pyruvate dehydrogenase (P-PDH) Ser293, and total PDH from $N = 3$ experimental replicates of HD-MB03 and SU_MB002 G3 MB cells treated with IACS-010759. $P$ values were calculated using two-sided student's $t$ test relative to vehicle control. **C, D** Model depicts the possible fates of pyruvate to be converted into **C** lactate or acetyl-CoA and **D** subsequent incorporation of acetyl-CoA into TCA cycle intermediates. Pyruvate incorporation into metabolites along the proposed pathway was examined by pre-treating HD-MB03 cells (vehicle or 100 nM IACS-010759) for 24 hours before sample lysis and the addition of U$^{13}$C$_3$Pyruvate. Isotopomeric distribution for indicated metabolites **C** lactate, acetyl-CoA, **D** citrate, fumarate, and malate was measured after 1 h using mass spectroscopy. $P$ values were calculated using two-sided student's $t$ test relative to vehicle control. Raw data of individual values from $N = 3$ experimental replicates is present in Source Data File. **E** Total acetyl-CoA levels were measured by fluorometric assay following 24 hours of 100 nM IACS-010759 treatment in HD-MB03 cells presented as mean ± SEM from $N = 3$ experimental replicates with $P$ value calculated using two-sided student's $t$ test. **F** Schematic diagram depicting the regulation of acetylation via pyruvate metabolism. Created with BioRender.com. **G** Representative western blots from $N = 3$ experimental replicates of MPC1, MPC2, P-PDH (Ser293), PDH, Ac-SOD2 K68, Ac-SOD2 K122, and SOD2 from HD-MB03 cells treated with 100 nM IACS-010759 for indicated time points. Source data provided in Source Data File.

increased *MYC* transcription, and enhanced MYC protein activation/stabilization.

This work also fits into the broader context of targeting metabolism as a cancer therapy. The use of metabolism-targeting agents for treating various cancers has been of therapeutic interest for several decades[17,49]. However, most focus has centered on inhibiting enhanced glycolytic metabolism or the 'Warburg effect' in tumors with little clinical success. Interestingly, we found that several glycolysis inhibitors are ineffective at suppressing MYC abundance or the growth of G3 MB cells. Recently, more attention has turned to targeting other metabolic processes in the context of cancer therapy, with considerable interest in manipulating mitochondrial metabolism[50]. For example, the popular anti-diabetic drug metformin, which partly functions by inhibiting mitochondrial complex-I activity, has gained significant traction as a potential anti-cancer agent[51]. Unfortunately, metformin is not a very specific or potent OXPHOS inhibitor[52]. Therefore, much effort is being directed toward developing better OXPHOS-targeting agents. IACS-010759 is a relatively new drug with exquisite potency and specificity towards inhibition of complex-I[25]. Interestingly, IACS-010759 appears to bind in a mechanism distinct from other complex-I targeting agents, which may contribute to its low toxicity profile[53]. Here, we reveal an additional benefit of IACS-010759 treatment that could be explicitly leveraged in MYC-activated tumors. We found that the effect of IACS-010759 treatment on MYC levels is conserved across cells from different tumor types including, colorectal, breast, and ovarian carcinoma. While the inhibition of complex-I activity may play a role in the anti-tumor effects of IACS-010759 treatment, our findings demonstrate that the downregulation of MYC is a significant event that is important for determining the cell-fate of G3 MB cells in response to IACS-010759 treatment. We found that IACS-010759 treatment significantly suppressed MYC-associated gene expression signatures in G3 MB cells. Re-instating MYC expression restores the growth and stemness capacity of IACS-010759-treated G3 MB cells while suppressing differentiation. Moreover, we uncovered an oxidative post-translational modification of MYC induced by IACS-010759 treatment in an MPC-dependent manner. The role of the MPC in cancer is controversial, with both tumor-suppressing and pro-tumorigenic functions identified[54–56]. However, we demonstrate that the MPC can influence the stability of the MYC oncoprotein through the regulation of SOD2-acetylation and mROS production. This in-depth characterization of the MPC-SOD2-MYC axis sheds light on metabolism-mediated mechanisms regulating oncogenic signaling programs.

Two phase-I clinical trials testing the efficacy of IACS-010759 treatment in advanced leukemias, lymphomas, and some solid tumors (i.e., colorectal, breast, pancreatic, and prostate cancer) (NCT02882321 and NCT03291938) have indicated promising preliminary findings for the clinical efficacy of this compound. For the past two decades, there have been no new therapeutic agents introduced into standard clinical practice for treating MB patients. Our study provides a strong rationale for the use of complex-I inhibitors such as IACS-010759 as a treatment paradigm for patients with MYC-activated G3 MB tumors. Further preclinical studies incorporating IACS-010759 with standard-of-care therapies for MB are warranted to guide future clinical investigations for this disease.

Insights into the unique molecular processes regulating oncoprotein abundance and activity will continue to guide the identification of targeted therapies that can selectively suppress the growth of MB tumor cells while sparing the normal cells required for proper brain development. Our findings demonstrate a therapeutically targetable and metabolism-dependent mechanism that modulates MYC abundance in G3 MB cells (without harming normal brain cell populations) well-positioned for quick-to-clinic translation. These findings also hold promising mechanistic molecular insights that can be adapted toward other MYC-activated cancers to which clinical investigations are already underway. Altogether, this study provides a clear, mechanism-based solution to address the problem of targeting the oncoprotein MYC with the potential for near-term clinical translation.

## Methods

### Ethics statement

The study protocol of mouse care and experiments was approved by the University of Manitoba's Animal Care Committee (protocol #21-021). All animal studies complied with relevant ethical regulations for animal testing and research. Human neural stem cells were isolated from embryonic brain tissue specimens from a gestational age of 8–11 weeks. Embryonic material was donated voluntarily for research purposes by women who have had a termination of pregnancy, as approved by the Hamilton Health Sciences/McMaster Health Sciences Research Ethics Board (REB; Project # 08-005). Samples were collected with informed written consent from the mothers whose pregnancy was terminated in strict accordance with institutional and legal ethical guidelines.

### Human G3 MB cell cultures, patient demographics, and treatments

Primary human pediatric MYC-amplified MB cell cultures were obtained from collaborators as kind gifts. Dr. Till Milde provided HD-MB03, a treatment-naïve large cell/anaplastic G3 MB cell model isolated from a male patient ≤3-years old during surgical intervention. SU_MB002 cells were provided by Dr. Yoon-Jae Cho and were derived from an autopsy specimen of the leptomeningeal compartment from a child with treatment-refractory, metastatic G3 MB after receiving only cyclophosphamide treatment. MB3W1 anaplastic G3 MB cells were derived from the malignant cells found in the pleural effusions of a male patient ≤3-years old and kindly provided by Dr. Matthias Wölfl. The D283 G3 MB cell line was established from malignant ascites cells and a peritoneal metastasis from a male patient G3 MB patient between 6 and 10 years old and was purchased from the American Type Culture Collection (ATCC, Rockville, MD, USA). MB cells were maintained in a humidified incubator at 37 °C with 5% $CO_2$ and cultured

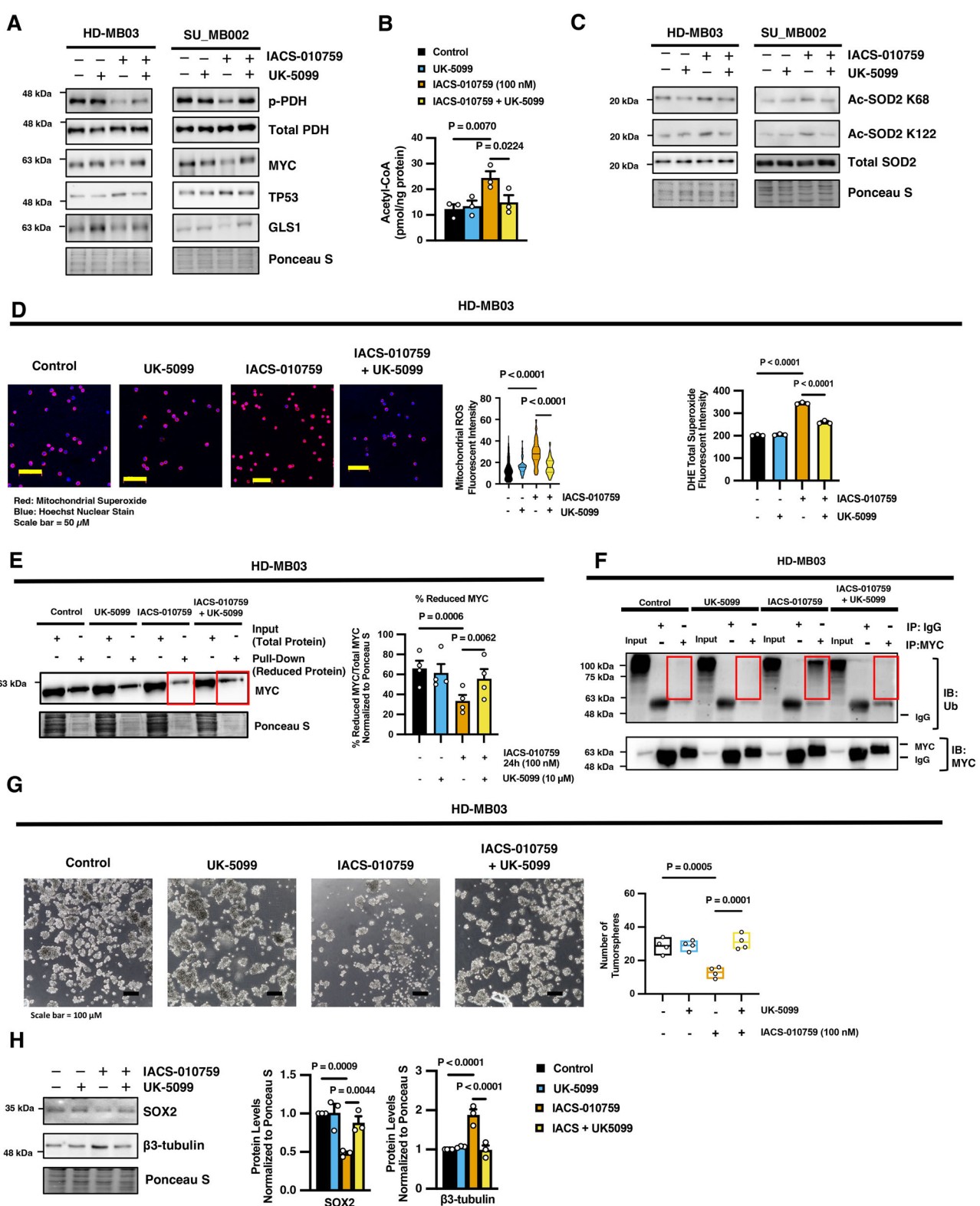

as tumorspheres in serum-free Knockout DMEM/F12 media (Life Technologies, Burlington, ON, Canada) with 1× GlutaMAX (Life Technologies), 20 ng/mL of EGF (STEMCELL Technologies, Vancouver, BC, Canada), 10 ng/mL of bFGF (STEMCELL Technologies), 1× NeuroCult SM1 Supplement (STEMCELL Technologies), 1× N2 Supplement (STEMCELL Technologies), 2 μg/mL Heparin (STEMCELL Technologies), and 1× antibiotic-antimycotic (Life Technologies). Human astrocytes were purchased from ScienCell Research Laboratories,

Carlsbad, CA, USA, and were maintained in complete Astrocyte Medium (ScienCell Research Laboratories). Human neural stem cells were isolated from embryonic brain tissue specimens from a gestational age of 8–11 weeks. Samples were collected with informed written consent from patients in strict accordance with institutional and legal ethical guidelines. Embryonic material was donated voluntarily for research purposes by women who have had a termination of pregnancy, as approved by the Hamilton Health Sciences/McMaster Health Sciences

**Fig. 8 | Mitochondrial pyruvate carrier activity dictates MYC regulation following IACS-010759 treatment in G3 MB cells. A–C** HD-MB03 and SU_MB002 G3 MB cells were treated with IACS-010759 and/or UK-5099 for 24 hours and **A** subjected to immunoblot analysis for p-PDH (Ser293), PDH, MYC, TP53, and GLS1 (Representative of $N = 3$ experimental replicates), **B** Acetyl-CoA measurement in HD-MB03 cells presented as mean ± SEM from $N = 3$ experimental replicates where $P$ values were calculated using one-way ANOVA with Fisher's LSD test, and **C** immunoblot analysis for Ac-SOD2 K68, Ac-SOD2 K122, and SOD2 (Representative of $N = 3$ experimental replicates). **D–H** HD-MB03 cells were treated with IACS-010759 and/or UK-5099 for 24 hours and subjected to: **D** MitoSOX stain (red) and Hoechst (blue) (Scale bar = 50 μm) where violin plot represents the quantification of red fluorescent intensity per cell of $N > 100$ cells from $N = 3$ experimental replicates with solid line at mean and dashed lines at quartiles, and cells were labeled with DHE where graphs represent quantification of fluorescent intensity using a plate reader normalized to cell number presented as mean ± SEM from $N = 3$ experimental replicates, $P$ values were calculated using one-way ANOVA with Tukey's multiple comparisons; **E** western blot of maleimide-PEG2 biotin-labeled reduced MYC levels precipitated with streptavidin-agarose relative to input lysates (no precipitation) and densitometry quantification presented as mean ± SEM from $N = 4$ experimental replicates where $P$ values were calculated using one-way ANOVA with Tukey's multiple comparisons; **F** representative blot of polyubiquitin modifications following immunoprecipitation of MYC or IgG negative control isotope antibody from $N = 3$ experimental replicates; **G** tumorsphere formation analysis with representative images (Scale Bar = 100 μm) and total sphere number (>50 μm) presented as box-plot with the box limits at minima and maxima and center line at mean from $N = 4$ experimental replicates where $P$ values were calculated using one-way ANOVA with Tukey's multiple comparisons, **H** representative western blot of SOX2 and β3-tubulin and densitometry quantification presented as mean ± SEM from $N = 3$ experimental replicates where $P$ values were calculated using one-way ANOVA with Fisher's LSD test. Source data provided in Source Data File.

Research Ethics Board (REB; Project # 08-005)[57], these cells were provided to the Sharif lab by Dr. Sheila Singh (Material transfer agreement (MTA) number: MTO20-120). Samples were dissociated in artificial cerebrospinal fluid containing 0.2 Wunisch U/mL Liberase Blendzyme 3 (Roche), and incubated at 37 °C in a shaker for 15 min. The dissociated tissue was then filtered through a 70 μm cell strainer and collected by centrifugation at 1500 × rpm for 3 min as previously described[58] and cultured in serum-free DMEM/F12 media (Life Technologies) with 20 ng/mL of EGF, 10 ng/mL of bFGF, 1× NeuroCult SM1 Supplement, 1× N2 Supplement, 2 μg/mL Heparin, and 1× antibiotic-antimycotic. A2780 (female derived), HEYA8 (female derived) cells were provided by Dr. Mark Nachtigal, HCT116 (male derived) and SW480 (male derived) cells were provided by Dr. Kirk McManus, and MDA-MB-468 (female derived) cells were provided by Dr. Yvonne Myal. A2780 and HEYA8 cells were maintained in RPMI-1640 media (Life Technologies) with 10% FBS (Life Technologies). SW480 and MDA-MB-468 cells were maintained in DMEM media (Life Technologies) with 10% FBS. HCT116 cells were maintained in McCoy's 5 A media (Life Technologies) with 10% FBS. No commonly misidentified cell lines were used in this study and the cell lines were authenticated by STR profiling (ATCC). All cell lines tested negative for Mycoplasma contamination using the MycoAlert® Mycoplasma Detection Kit (Lonza). For genetic manipulations (i.e., exogenous overexpression of small-interfering RNA knockdown), cells were transfected with respective plasmid vectors or oligonucleotides using Lipofectamine 3000 transfection reagent (ThermoFisher) according to manufacturer's protocol. The vendors and catalog numbers for all drugs, plasmids, and oligonucleotides are provided in Supplementary Table 1. Specific drug doses and treatment duration are indicated in corresponding figure legends.

### Cell count
To monitor cell growth and viability, the trypan blue exclusion assay was used to determine the viability of cells in suspension. Trypan blue is a dye which stains dead cells due to their breached membrane integrity, allowing to discriminate non-viable cells. $2 \times 10^4$ cells were seeded in 12-well plates. 24 hours after seeding, cells were treated with designated drug or vehicle control. At indicated time points, cells were collected and stained 1:1 with trypan blue and viable cells were counted using a Bio-Rad TC20 Cell Counter.

### Bioinformatics analysis
Analysis of TCGA pan-cancer studies was performed using the cBioPortal online bioinformatics platform for cancer genomics[59,60]. The TCGA pan-cancer atlas (a combination of studies from 32 cancers with a total of 10,967 samples)[21] was queried for alterations of MYC and the alteration frequency was reported and summarized based on cancer type. The amplification frequency of selected cancer types was downloaded and plotted using Prism GraphPad 9.0 along with findings

from Northcott et al.[22] regarding the amplification frequency of MYC specifically in group 3 medulloblastoma brain tumors. For survival analysis, the TCGA pan-cancer atlas was queried for alterations of MYC, and the 5-year progression-free survival proportion for 10612 patients from the MYC altered versus MYC unaltered groups was downloaded and plotted using Prism GraphPad 9.0 where logrank hazard ratio and $P$ value were calculated. For survival analysis of group 3 medulloblastoma patients, analysis and data extraction for Cavalli et al.[5] was performed using the R2 genomics and visualization online platform (http://r2.amc.nl) available through the gene expression omnibus (GEO) via accession number GSE85217. MYC mRNA expression and overall survival data from 144 group 3 medulloblastoma tumors was downloaded. Samples were categorized into upper >75% percentile (MYC^HIGH) and lower <25% percentile (MYC^LOW) expression quartiles and plotted using GraphPad 9.0 where logrank hazard ratio and P value were calculated.

Patient MB tumor proteomics data was accessed from Archer et al.[4] (https://doi.org/10.1016/j.ccell.2018.08.004). Data was previously clustered and subtyped into MYC-activated (G3a) and non-MYC-activated (G3b) group 3 MB tumors. A list of significantly differentially expressed genes (both positive and negative, >2.0 fold-change, and $p < 0.05$ based on student's $t$ test) between MYC-activated and non-MYC-activated G3 MB tumors were entered into DAVID for functional annotation of processes based on gene ontology (GO). Statistics were reported by DAVID software. Patient MB tumor RNA-sequencing data was accessed from Cavalli et al.[5] using the R2 genomics and visualization online platform (http://r2.amc.nl; GEO accession number GSE85217). RNA expression from 144 group 3 medulloblastoma tumors was downloaded which were previously subtyped into MYC-activated (G3gamma) and non-MYC-activated (G3alpha and G3beta) group 3 MB tumors. Gene set enrichment analysis (GSEA) was performed by running proteomics data from MYC-activated and non-MYC-activated G3 MB through gene sets from the Molecular Signatures Database (mSigDB) using GSEA software (Broad Institute)[61,62] and metabolic gene signatures that were enriched in MYC-activated Group 3 MB tumors were identified. Statistics were reported by GSEA software. Cysteine oxidation proteomics datasets were accessed from the supplementary data files from van der Reest et al.[43] (https://doi.org/10.1038/s41467-018-04003-3) and Xiao et al.[42] (https://doi.org/10.1016/j.cell.2020.02.012) and queried for peptide hits corresponding to the c-MYC protein.

### Protein extraction and western immunoblotting
Suspension cells were collected and centrifuged at $500 \times g$ for 5 min at 4 °C. Adherent cells were scraped in cold 1× PBS at pH 7.4 and centrifuged at $500 \times g$ for 5 min at 4 °C. Pellets were resuspended in RIPA lysis buffer (25 mM Tris pH 7.6, 150 mM NaCl, 1% NP-40, 1% sodium deoxycholate, 1% SDS) containing 1X protease and phosphatase inhibitor cocktail. Whole cell lysates were incubated on ice for 45 min and

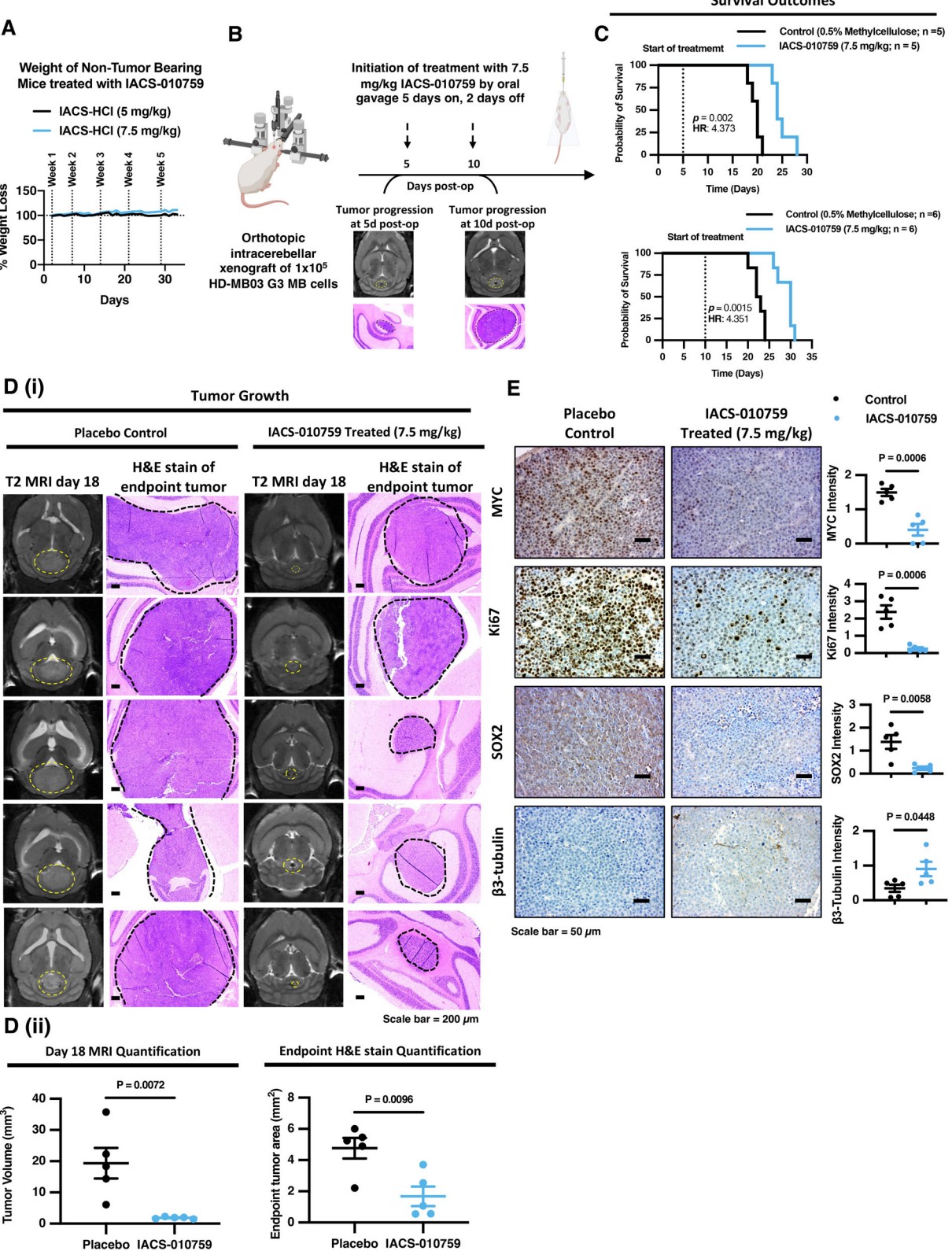

then sonicated for 1 min. The samples were centrifuged at $20,000 \times g$ for 15 min at 4 °C and the supernatants containing the proteins were collected. Protein concentrations were determined using the colorimetric Micro BCA assay kit (Life Technologies) according to manufacturer's instructions. Equal amounts of protein were boiled in Laemmli sample buffer (Bio-Rad) containing 5% β-mercaptoethanol for 5 min and then resolved by SDS-PAGE. Protein was transferred onto nitrocellulose membranes (BioRad). Membranes were coated in Ponceau S dye to detect total protein concentration for normalization. Membranes were then washed in PBST (PBS, 0.05% Tween 20) to remove Ponceau S before blocking. Membranes were blocked in 5% non-fat milk in PBST for 45 min, then washed in PBST and incubated in the appropriate primary antibody overnight at 4 °C with shaking. The primary antibodies were prepared at a 1:1000 dilution in 1% BSA in

**Fig. 9 | IACS-010759 treatment prolongs animal survival in a pre-clinical orthotopic G3 MB xenograft model. A** Weight monitoring of non-tumor-bearing NOD-SCID gamma mice treated with either 5 ($N = 1$) or 7.5 mg/kg ($N = 1$) of IACS-010759 for 5 consecutive days followed by a 2-day treatment holiday (Q2Dx5). **B** Schematic diagram created with BioRender.com depicting the experimental setup of the animal tumor study and the relative tumor development by T2 MRI imaging and H&E stain at 5-days and 10 days post-tumor cell implantation. **C** Kaplan–Meier survival analysis of tumor-bearing animals that received either placebo control (0.5% methylcellulose) or 7.5 mg/kg Q2Dx5 IACS-010759 treatment starting at either 5 days ($N = 5$ per cohort) or 10 days ($N = 6$ per cohort) post-tumor cell implantation. $P$ values and hazard ratio were determined using the logrank method. **D** (i) T2 MRI images of tumor masses from matching planes at 18 days post-surgery and H&E stained brain slices (Scale Bar = 200 μm) from mice at endpoint that received either placebo control (0.5% methylcellulose; $N = 5$ animals) or 7.5 mg/kg Q2Dx5 IACS-010759 treatment ($N = 5$ animals) starting at 5-days post-surgery. (ii) Quantification of tumor volume from 18-day MRI images and area of endpoint tumors from H&E stained slides presented as mean ± SEM from $N = 5$ animals. $P$ values were calculated using two-sided student's $t$ test relative to vehicle control. **E** Representative IHC images (Scale Bar = 50 μm) of MYC, Ki67, SOX2, and β3-tubulin levels in tumors from mice at endpoint that received either placebo control (0.5% methylcellulose; $N = 5$ animals) or 7.5 mg/kg Q2Dx5 IACS-010759 treatment ($N = 5$ animals) starting at 5-days post-surgery and quantification of DAB signal intensity presented as mean ± SEM. $P$ values were calculated using two-sided student's $t$ test relative to vehicle control. Source data provided in Source Data File.

PBST. The following day, membranes were washed in PBS before incubating in appropriate horseradish peroxidase (HRP)-conjugated secondary antibodies. Secondary antibodies were prepared at a 1:10,000 dilution in 5% non-fat milk in PBS and added to the membranes for 1.5 h at room temperature. Following secondary antibody incubation, membranes were washed and proteins were detected using Clarity ECL Western substrate (Bio-Rad) and visualized using the ChemiDoc MP imaging system (Bio-Rad) chemiluminescent setting. Details of specific primary and secondary antibodies used in this study and their dilutions can be found in Supplementary Table 2.

### Glutamate assay
Glutamate measurements were performed using the Fluorometric Glutamate Assay Kit from Abcam (ab138883). Samples and assay buffers were prepared in correspondence with manufacturer's instructions. Briefly, cells were seeded in 6-well plates and treated with indicated doses of CB-839 or vehicle control for 24 hours prior to the experiment. Following treatment, cells were collected and counted for normalization. Cells were lysed in 100 μL of 1× mammalian lysis buffer (Abcam) and incubated for 20 minutes. 50 μL of sample lysate was added to 96-well black-walled plates along with 50 μL of Glutamic acid Reaction Mix prepared according to manufacturers instructions. The plate was incubated at room temperature for 30 minutes and fluorescent intensity of Ex/Em = 540/590 nm was measured using the Synergy 2 microplate reader (BioTek).

### Lactate assay
Lactate measurements were performed using the Lactate Assay Kit from Sigma-Aldrich (MAK064) according to manufacturer's instructions. Samples were treated with indicated agents for 24 hours prior to the experiment. Serum-free growth media was collected from and frozen at -80 °C until processing. On the day of experiment, media samples were diluted 10× in lactate assay buffer prepared according to manufacturer's instructions. 10 μL of diluted media was added to a clear 96-well plate and adjusted to a final volume of 50 μL with lactate assay buffer for a total 50× dilution. 50 μL of the Master Reaction Mix containing 46 μL of lactate assay buffer, 2 μL of lactate enzyme mix and 2 μL of lactate probe was added to each well. Colorimetric absorbance was measured at 570 nm using a SpectraMax Plus Plate Reader.

### ATP assay
ATP measurements were performed using ATP detection assay kit from Cayman Chemicals. Samples and assay buffers were prepared in correspondence with manufacturer's instruction. Briefly, cells were seeded in 24-well plates and treated with either 100 nM of IACS-010759 or vehicle control for the indicated amount of time prior to the experiment. Following treatment, cells were collected and counted for normalization. Cells were homogenized in 0.5 mL of sample buffer. 10 μL of sample lysate was added to 96-well white-walled plates along with 20 μL 5× assay buffer, 8 μL ddH₂O, 1.7 μL DDT, 0.5 μL of ᴅ-luciferin, and 0.1 μL of luciferase per well. Plate was incubated for 20 min and

luminescence was measured using the Synergy 2 microplate reader (BioTek).

### Annexin V apoptosis assay
To monitor apoptosis, cells were stained with the FITC Annexin V Apoptosis Detection Kit I from BD Biosciences according to manufacturers instructions. Live cells were suspended in 1× Annexin V binding buffer (0.1 M Hepes/NaOH (pH 7.4), 1.4 M NaCl, 25 mM CaCl₂) with 5 μL of both Annexin.V and propidium iodide (PI) for 10 minutes and analyzed by flow cytometry using a BD FACS Canto-II and FlowJo™ v10 Software.

### Tumorsphere formation assay
Cells were dissociated into single cell suspensions through gentle trituration by pipetting and cells were seeded at a low density of $2.5 \times 10^3$ cells/well on ultralow-attachment 24-well plates and treated with 100 nM of IACS-010759 after 24 hours. Images of tumorspheres were taken 5 days after initial seeding using a light microscope from multiple fields of view. For secondary sphere formation, primary tumorspheres were dissociated, counted and re-seeded at a density of $2.5 \times 10^3$ cells/well on ultralow-attachment 24-well plates and imaged after 5 days. Using ImageJ software (National Institutes of Health), spheres with a diameter equal to or larger than 50 μm were deemed tumorspheres and the average number was determined from at least 3 independent experiments.

For the limiting dilution assay, cells were seeded at varying densities of 1, 5, 10, 25, 50, and 100 cells/well in multiple wells and treated with 100 nM of IACS-010759 after 24 hours. After 5 days, wells were examined and the number of wells with tumorspheres formed were counted.

### Animal studies
All experiments involving animals were approved by the University of Manitoba's Animal Care Committee (protocol #21-021). Non-obese diabetic (NOD) severe combined immunodeficient (SCID) IL2R gamma null (NSG) mice (NOD.Cg-*Prkdc*^scid^ *Il2rg*^tm1Wjl^/SzJ) purchased from Jackson Laboratories (Strain #:005557) were used for all procedures and animals were housed in IVC caging and held according to the Guidelines of the Canadian Council on Animal Care and the Animal Care and Use Policy of the University of Manitoba. Irradiated feed (5P76 - Prolab® IsoPro® RMH 3000) was used and caging and bedding were sterilized by steam autoclave. Animals had continuous access to food and water for the study duration. Room ambient temperature was 21–23 °C with a relative humidity target of 50%, but within a range of 30–60%. Light cycle was 12 h on/12 h off beginning with lights on at 6:00 a.m.

For safety studies in non-tumor-bearing animals, two male NSG mice between 7-9 weeks old were allocated to receive either 5 or 7.5 mg/kg of IACS-01759 in 0.5% methylcellulose, 5 days a week, for 6 weeks. General animal health was monitored, and weight was recorded daily during the duration of the study with no evident adverse side effects observed.

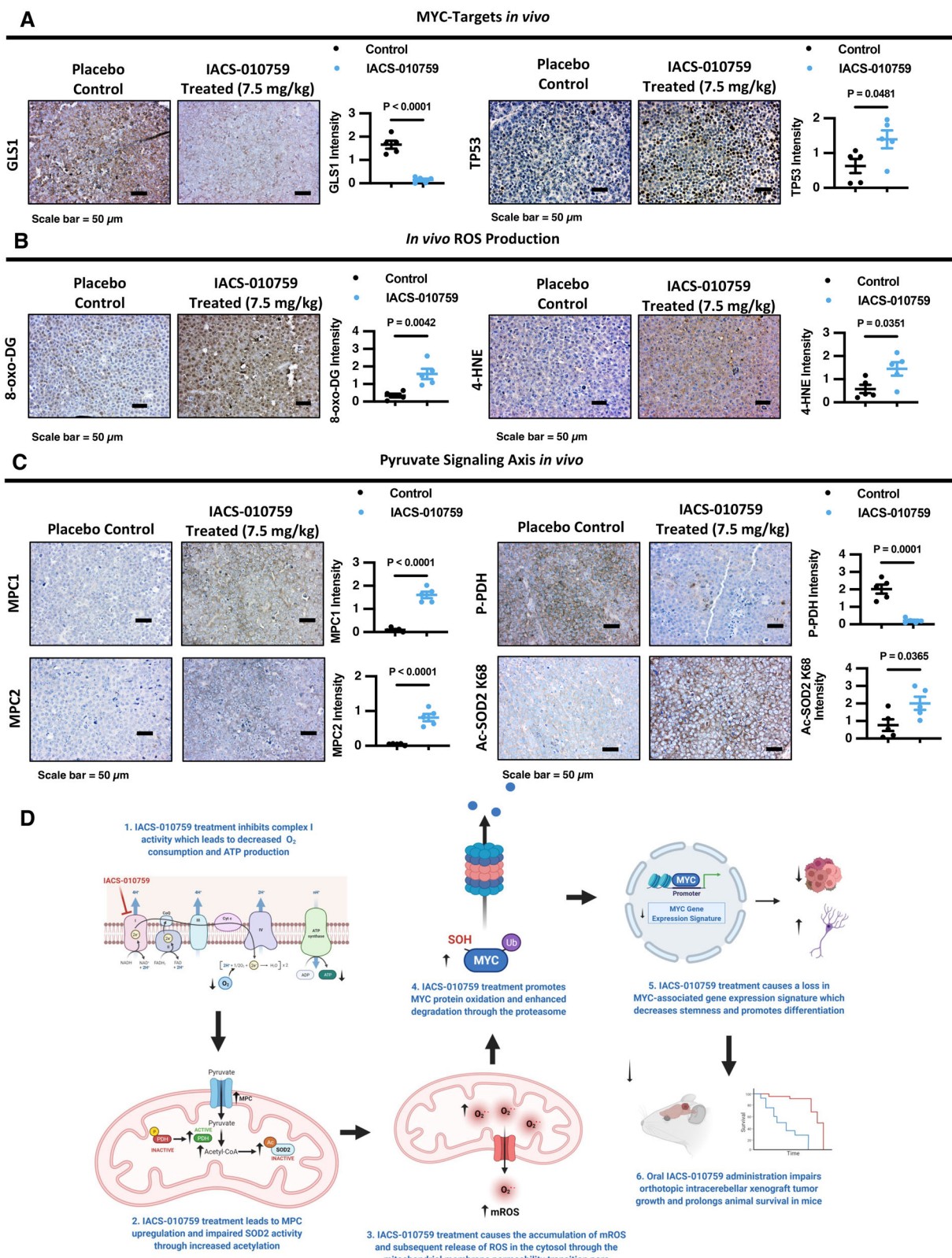

**Fig. 10 | IACS-010759 treatment regulates MYC downstream targets and the MPC-SOD2-ROS signaling axis in vivo. A–C** Representative IHC images (Scale Bar = 50 μm) of **A** GLS1, TP53, **B** 8-Hydroxy-2′-deoxyguanosine (8-oxo-DG), 4-Hydroxynonenal (4-HNE), **C** MPC1, MPC2, P-PDH (Ser293), and Ac-SOD2 K68 levels in tumors from mice at endpoint that received either placebo control (0.5% methylcellulose; *N* = 5 animals) or 7.5 mg/kg Q2Dx5 IACS-010759 treatment (*N* = 5 animals) starting at 5-days post-surgery and quantification of DAB signal intensity presented as mean ± SEM. *P* values were calculated using two-sided student's *t* test relative to vehicle control. **D** Schematic diagram representing the complete proposed mechanism of MYC regulation by IACS-010759 treatment in G3 MB cells. Created with BioRender.com. Source data provided in Source Data File.

To monitor the timeline of tumor development, two male NSG mice between 7-9 weeks old were anesthetized using isoflurane gas (5% induction and 2.5% maintenance), and $1 \times 10^5$ HD-MB03 G3 MB cells suspended in 5 μL of PBS were injected into the cerebellum in a nonrandomized, nonblinded fashion. Tumors were monitored using T2 MRI imaging was performed at either 5 or 10 days post-op using an MR Solutions cryogen free FlexiScan 7 T system (MR Solutions, Guildford, Surrey, UK). After imaging, animals were sacrificed and perfused with formalin and brains were harvested and preserved in formalin for at least 5-7 days prior to histopathology. Formalin-fixed brains were sliced, paraffin-embedded, and cut to prepare tissue slides for H&E.

For tumor studies, 7–9-week-old male NSG mice were anesthetized using isoflurane gas (5% induction and 2.5% maintenance), and $1 \times 10^5$ HD-MB03 G3 MB cells suspended in 5 μL of PBS were injected into the cerebellum in a nonrandomized, nonblinded fashion. At either day 5 or 10 post tumor cell implantation, animals were randomized to receive either 7.5 mg/kg of IACS-010759 or 0.5% methylcellulose as the vehicle control administered by oral gavage 5 days a week. T2 MRI imaging was performed at 18 days post-op using an MR Solutions cryogen free FlexiScan 7 T system (MR Solutions, Guildford, Surrey, UK) to monitor tumor development. It was noted that IACS-010759 treatment increased the sensitivity of animals to isoflurane anesthesia. To mitigate this, animals were imaged by MRI immediately after the two-day treatment holiday prior to initiating the next treatment cycle. For survival studies, animals were monitored daily for general health and weights were recorded, and a 20% reduction from peak body weight was the defined humane endpoint. University of Manitoba's Animal Care Committee has no maximal tumor size/burden restrictions for orthotopic brain tumor studies. Due to the sensitive location of these tumors, humane endpoints are defined by 20% reduction from peak body weight or significant clinical deterioration (i.e., evidence of pain, neurological symptoms, paralysis, etc.) as decided in consultation with the veterinarian, regardless of tumor size/burden. Once endpoint was reached, animals were perfused with formalin and brains were harvested and preserved in formalin for at least 5-7 days prior to histopathology. Formalin-fixed brains were sliced, paraffin-embedded, and cut to prepare tissue slides for H&E staining and immunohistochemistry.

## Immunohistochemistry

Following deparaffinization of xenograft tumor samples, antigen retrieval was performed in citrate buffer at 95–100 °C for 20 min. Slides were washed and treated for 10 min for endogenous peroxidase, and again washed in 1× PBS. The samples were blocked with 3% sheep serum, then incubated with primary antibodies overnight at 4 °C. Slides were washed and incubated with Biotin-conjugated secondary antibody for 2 h at room temperature. Details of specific primary and secondary antibodies used in this study and their dilutions can be found in Supplementary Table 3. Streptavidin/HRP (Life Technologies) was then added for 30 min followed by development with DAB and counterstaining with hematoxylin. Finally, coverslips were mounted with Permount (Fisher Scientific). Images were captured using a Zeiss Axio Imager and quantified using ImageJ software (National Institutes of Health).

Publicly available representative group 3 MB patient tumor histology images (H&E and MYC IHC of Med-411FH) were accessed from the pediatric PDX (Olson) portal on the r2genomics platform.

## Quantitative real-time PCR

RNA was extracted from cells using the Aurum total RNA Mini Kit (Bio-Rad) according to manufacturer's protocol and cDNA was synthesized using iScript (Bio-Rad). Each sample of cDNA was quantitated and diluted to an equal concentration of 10 ng/mL. The Applied Biosystems 7300 real-time PCR machine was used for the quantitative real-time PCR (qRT-PCR), using SYBR Green Supermix (Bio-Rad). All primers, as described in Supplementary Table 4, were purchased from Invitrogen. *HPRT1* was used for normalization of the genes of interest. The results were analyzed using $2^{-\Delta\Delta CT}$ method and expressed as fold change to respective vehicle-treated controls.

## Oroboros respirometry

Mitochondrial respiration was evaluated based on oxygen consumption using high resolution Oroboros oxygraphy (Oroboros Instruments GmbH, Innsbruck, Austria). In brief, an Oroboros oxygraph is a Clarke-type oxygen electrode that has two chambers (0.5 mL volume) equipped with oxygen sensors. Air calibration of these oxygen sensors is performed routinely on any day before starting a respirometric experiment. Cells were treated with either 100 nM of IACS-010759 or vehicle control for the indicated amount of time prior to the experiment.

To measure maximal oxygen consumption, G3 MB cells were collected, counted, and $1 \times 10^5$ cells were resuspended in serum-free Knockout DMEM/F12 media (Life Technologies, Burlington, ON, Canada) with 1× GlutaMAX (Life Technologies), 20 ng/mL of EGF (STEMCELL Technologies, Vancouver, BC, Canada), 10 ng/mL of bFGF (STEMCELL Technologies), 1× NeuroCult SM1 Supplement (STEMCELL Technologies), 1× N2 Supplement (STEMCELL Technologies), 2 μg/mL Heparin (STEMCELL Technologies), and 1× antibiotic-antimycotic (Life Technologies) and added to the Oroboros oxygraphy chambers. Oxygen consumption rate (OCR) served as a surrogate for mitochondrial electron transport chain function. OCR was measured at baseline and following sequential treatments with the ATP synthase inhibitor oligomycin, uncoupler carbonyl cyanide-p-(trifluoromethoxy) phenylhydrazone (FCCP) to remove the pH gradient and enable maximal rates of electron transport to occur, and antimycin A to block respiratory electron flux at Complex III. After measurement of basal respiration rates, the following chemicals were added: oligomycin (2 μM), FCCP (2–6 μM), and antimycin A (2 μM). The mitochondrial respiration parameters are defined as: Baseline respiration is termed basal respiration and maximal respiration is achieved by the addition of the uncoupler FCCP.

To measure substrate-specific oxygen consumption, G3 MB cells were collected, counted, and $1 \times 10^5$ cells were resuspended in K-media (80 mM KCl, 10 mM Tris-HCl, 3 mM $MgCl_2$, 1 mM EDTA, 5 mM potassium phosphate, pH 7.4) and added to the Oroboros oxygraphy chambers. Cells were permeabilized with digitonin (8 μM) and treated with substrates pyruvate (10 mM), malate (2 mM), and ADP (2 mM) as a surrogate for Complex-dependent oxygen consumption followed by rotenone (2 μM) to block Complex I activity. The substrate succinate (10 mM) was added as a surrogate for Complex II-dependent oxygen consumption followed by antimycin A (2 μM) to block respiratory electron flux at mitochondrial complex III. Ascorbate (500 μM) and TMPD (125 μM) were added as a surrogate for Complex IV activity which was blocked with $NaN_3$ (80 mM).

Oroboros DatLab software was used to calculate the OCR and for the graphic presentation of experimental data. Statistical analysis was performed on three independent experiments.

## Mitochondrial superoxide stain

To monitor mitochondrial superoxide production, G3 MB cells were stained with the fluorescent dye MitoSOX Red mitochondrial superoxide indicator (Life Technologies) according to manufacturer's instructions. 50 μg of MitoSOX reagent was dissolved in DMSO to prepare a 5 mM stock solution. Cells were suspended in Hanks Balanced Salt Solution (HBSS) with calcium and magnesium (Life Technologies) containing a final concentration of 5 μM of MitoSOX reagent and 1 μM of Hoechst 33342 nuclear counterstain. Cells were incubated in staining solution for 10 minutes at 37 °C. Following incubation, cells were washed and imaged using a Zeiss Axio Imager with Apotome 2.

### JC-1 stain

To measure mitochondrial membrane potential, G3 MB cells were loaded with the JC-1 dye (Cayman Chemicals) according to manufacturer's instructions. Cells were suspended in culture medium containing a final concentration of 2 μM JC-1 dye and incubated at 37 °C for 15 minutes protected from light. Following incubation cells were washed and imaged in a 96-well black-walled plate using an EVOS Floid (ThermoFisher) fluorescent microscope. JC-1 aggregates were imaged at ex/em of 535/595 nm and JC-1 monomers were imaged at ex/em of 485/535 nm. Fluorescent intensity was quantified using Image J software and the ratio of the fluorescent intensity of JC-1 aggregates to monomers was calculated.

### DHE stain

To monitor overall cellular superoxide production, G3 MB cells were stained with the fluorescent probe Dihydroethidium (DHE) (Life Technologies). Cells were suspended in PBS containing a final concentration of 10 μM of DHE and 5 μM of SYTO™ 9 Green Fluorescent Nucleic Acid Stain (Invitrogen). Cells were incubated in staining solution for 30 minutes at 37 °C. Following incubation, cells were washed and imaged using a Zeiss Axio Imager with Apoptome 2.

### Mitochondrial membrane permeability transition pore opening stain

The Image-iT™ LIVE Mitochondrial Transition Pore Assay Kit (Invitrogen) was used to monitor mitochondrial membrane permeability transition pore opening according to manufacturer's protocol. G3 MB cells were suspended in Hanks Balanced Salt Solution (HBSS) with calcium and magnesium (Life Technologies) containing a final concentration of 1.0 μM calcein AM stock solution, 200 nM MitoTracker Red CMXRos stock solution, and 1.0 mM CoCl2. Cells were incubated in staining solution for 15 minutes at 37 °C. Following incubation cells were washed and imaged in a 96-well black-walled plate using an EVOS Floid fluorescent microscope.

### Protein oxidation assay

Cells were lysed in RIPA buffer containing 500 μM of EZ-Link Maleimide-PEG2-Biotin (Life Technologies) and incubated on ice for 4 h to label reduced thiols. Lysates were sonicated for 20 s and centrifuged at $20,000 \times g$ for 15 min at 4 °C and the supernatants containing the proteins were collected. Protein concentrations were determined using the colorimetric Micro BCA assay kit (Life Technologies) according to manufacturer's instructions. 50 μg of protein lysate was incubated with 150 μL of NeurtAvidin agarose slurry overnight at 4 °C with rotation. The following day, agarose beads were washed and proteins were eluted from the resin-bound complex in 1× Laemmli sample buffer and subjected to SDS-PAGE electrophoresis and western blotting for specified proteins.

### Immunoprecipitation and ubiquitination

Cells were lysed in RIPA buffer containing 10 mm N-ethylmaleimide (NEM) and 1 mM EDTA to block deubiquitinase activity. Lysates were sonicated for 20 s and centrifuged at $20,000 \times g$ for 15 min at 4 °C and the supernatants containing the proteins were collected. Protein concentrations were determined using the colorimetric Micro BCA assay kit (Life Technologies) according to manufacturer's instructions. Immunoprecipitation was performed using Dynabeads™ Protein G (Invitrogen) according to manufacturers protocol. Specified c-MYC antibody (Cell Signaling) was conjugated to 0.15 mg of Dynabeads magnetic beads at a 1:20 dilution. 50 μg of protein lysate was incubated with the antibody-beads complex overnight at 4 °C with rotation. The following day, magnetic bead complexes were washed and proteins were eluted in 1× Laemmli sample buffer and subjected to SDS-PAGE electrophoresis and western blotting using anti-ubiquitin antibody. Details of specific

primary and secondary antibodies used in this study can be found in Supplementary Table 2.

### Site-directed mutagenesis

Mutant MYC constructs were commercially generated using GenScript's site-directed mutagenesis service. The wild-type c-MYC gene (NCBI Gene ID: 4609; Transcript: NM_002467.6; Protein: NP_002458.2) was cloned into the pcDNA3.1(+)-C-eGFP vector using NheI/NotI restriction sites and 10 C → G mutants were made from this wild-type sequence.

### Biotin-switch assay

HD-MB03 cells were transfected with pcDNA3.1(+)-MYC-c-eGFP construct containing either WT MYC or cysteine point mutants using Lipofectamine 3000 transfection reagent (ThermoFisher) according to manufacturer's protocol. 24 hours after transfection, overexpression was validated by fluorescent microscopy and cells were treated with IACS-010759 for an additional 24 hours. The biotin-switch assay was performed as previously described[38–41] with modifications. Briefly, cells were lysed in RIPA buffer (pH 6.8-7.0) containing 3 mM of NEM and samples were incubated at room temperature for 30 minutes protected from light. Lysates were sonicated for 20 s and centrifuged at $20,000 \times g$ for 15 min at 4 °C and the supernatants containing the proteins were collected. Excess NEM was quenched by adding a final concentration of 3 mM L-cysteine and samples were incubated for 1 hour at 37 °C. Oxidized cysteines were reduced by adding a final concentration of 4 mM Tris(2-carboxyethyl)phosphine hydrochloride (TCEP) and pH was adjusted to 7.0 using ammonium hydroxide. Samples were incubated for 1 hour at room temperature. Newly reduced thiols were labeled with 1 mM EZ-Link Maleimide-PEG2-Biotin for 2 hours at room temperature protected from light. Protein concentrations were determined using the colorimetric Micro BCA assay kit (Life Technologies) according to manufacturer's instructions. Immunoprecipitation was performed to isolate GFP-tagged exogenous MYC protein using Dynabeads™ Protein G (Invitrogen) according to manufacturers protocol. Specified GFP antibody (Santa Cruz) was conjugated to 0.15 mg of Dynabeads magnetic beads at a 1:20 dilution. 1 mg of protein lysate was incubated with the antibody-beads complex overnight at 4 °C with rotation. The following day, magnetic bead complexes were washed and proteins were eluted in 2X SDS buffer (4% SDS, 0.25 M DTT, 0.25 M Tris pH 6.8) and samples were dot-blotted directly onto nitrocellulose membrane. Once the nitrocellulose membrane dried completely, the membrane was incubated in blocking buffer (0.5% skim milk powder in PBST) for 1 hour at room temperature. Following blocking, membranes were incubated with either a 1:1000 dilution of Streptavidin-HRP (Life Technologies) in 0.1% BSA in PBST or 1:1000 dilution of anti-GFP antibody (Santa Cruz) in 0.1% BSA in PBST for 1 hour at room temperature. Membranes with streptavidin-HRP were washed and imaged immediately by adding ECL substrate and detecting the chemiluminescent signal using a Bio-Rad Chemi-Doc imager. Membranes with anti-GFP were washed incubated in appropriate secondary antibody conjugated to HRP for 1 hour at room temperature and then imaged using the same protocol. Details of specific primary and secondary antibodies used in this study can be found in Supplementary Table 2.

### U-$^{13}$C$_3$-pyruvate stable isotope tracing central carbon metabolomics

$1 \times 10^7$ cells were pelleted by centrifugation at $600 \times g$ for 5 minutes. Supernatants were removed and cells were washed in PBS. Pellets were resuspended in 1× Cystosol Extraction Buffer Mix (Abcam) containing DTT and 1× Protease/Phosphatase inhibitors (Life Technologies) and samples were incubated on ice for 30 min. Once cells were swollen, samples were lysed using a dounce homogenizer on ice (~30–50 passes) with pestle B. Homogenate was centrifuged for at $700 \times g$ for

10 minutes at 4 °C to pellet cell debris. Supernatant was collected and centrifuged at 10,000 × *g* in a microcentrifuge for 30 minutes at 4 °C to isolate mitochondria. Cytosolic fraction (supernatant) was discarded, and intact mitochondria (pellet) were incubated for 1 hour at 37 °C in the dark in 300 μL of incubation buffer containing the following: 50 mM Tris-HCl pH 7.5, 0.25 mM KCl, 5 mM MgCl$_2$, 2 mM DTT, 1× protease and phosphatase inhibitors, 1 mM ADP, 1 mM ATP, 1 mM NAD, 1 mM FAD, and 10 mM [U-$^{13}$C$_3$]-sodium pyruvate (Cambridge isotopes). Polar metabolites were extracted by adding 600 μL of ice-cold methanol and samples were stored at -80 °C until processing.

Mass spectrometry processing of samples was performed by The Metabolomics Innovation Center (TMIC) Victoria Node at the University of Victoria Genome BC Proteomics Center. Samples were thawed, centrifuged, and 100 μL of the supernatant was dried under nitrogen gas. In the sample residues, 50 μL of 100-mM 3-NPH solution and 50 μL of 100-mM EDC-3% pyridine solution were added. The mixtures were reacted at 30 °C for 40 min. After reaction, 200 μL of water was added to each solution. 10 μL aliquots of the resultant solutions were injected into a C18 column (2.1 × 100 mm, 1.8 μm) to quantitate carboxylic acids by UPLC-MRM/MS with (−) ion detection on an Agilent 1290 UHPLC coupled to an Agilent 6495 C QQQ MS instrument, according to the procedure and LC-MS parameters described previously[63]. For other metabolites, 50 μL of 20% methanol was added to dissolve the residue of each sample. 10 μL of each resultant sample solution was injected into a C18 LC column (2.1 × 100 mm, 2.5 μm) for UPLC-MRM/MS runs with (−) ion detection on a Waters Acquity UPLC system coupled to a Sciex 6500 Plus MS instrument, with the use of a tributylamine acetate buffer−acetonitrile as the mobile phase for gradient elution (5% to 50% B over 22 min) at 0.25 mL/min and 60 °C.

### Statistical analysis

Statistical analysis was performed in GraphPad Prism 9. Statistical parameters including the exact value of n and the statistical significance are reported in the figures and figure legends. To assess significant differences between single measurements of two groups of normally distributed data, unpaired two-tailed Student's *t* test was used. To assess significant differences between more than two groups of normally distributed data, we performed one-way or two-way analysis of variance, followed by either a Fishers Least Significant Difference (LSD) test or when all pairs of datasets were compared, Tukey's multiple comparisons test was performed and when every mean was only compared to the control mean, Dunnett's multiple comparisons test was performed.

### Graphics and illustrations

Original graphics and schematics were generated using Microsoft PowerPoint and created with BioRender.com.

### Reporting summary

Further information on research design is available in the Nature Portfolio Reporting Summary linked to this article.

## Data availability

All data generated or analyzed during this study are included in this manuscript (and its supplementary information files). Source data are provided in Source Data File. The TCGA pan-cancer data used in this study are available from BioPortal for Cancer Genomics[21,59,60] https://www.cbioportal.org/. Information on MYC amplification frequency in group 3 medulloblastoma brain tumors was extrapolated from findings reported by Northcott et al.[22] and SNP profiling array data is available from the gene expression omnibus (GEO) under accession number GSE37385. Patient MB tumor RNA-sequencing data from Cavalli et al.[5] used in this study are available from the online platform (http://r2.amc.nl) and accessible from the gene expression omnibus (GEO) under

accession number GSE85217. Patient MB tumor proteomics data used in this study are available in the supplementary data files from Archer et al.[4] (https://doi.org/10.1016/j.ccell.2018.08.004). Gene sets used in GSEA analysis are available from the Molecular Signatures Database (mSigDB, Broad Institute)[61,62] https://www.gsea-msigdb.org/gsea/msigdb/. Cysteine oxidation proteomics datasets are available in the supplementary data files from van der Reest et al.[43] (https://doi.org/10.1038/s41467-018-04003-3) and Xiao et al.[42] (https://doi.org/10.1016/j.cell.2020.02.012). Patient G3 MB specimen images are available from the R2 online platform in the Pediatric PDX (Olson) portal (https://hgserver1.amc.nl/cgi-bin/r2/main.cgi?dscope=PDX_OLSON&option=about_dscope)[48]. Source data are provided with this paper.

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

## Acknowledgements

We thank Vet Services at the University of Manitoba, particularly Rhonda Kelley, Shawn Blum, and Denise Borowski for exceptional technical support and mentorship during the course of our animal studies. We also thank Dr. Mike Jackson, Small Animal Imaging Facility, and Christine Zhang, Flow Cytometry Core Facility at the University of Manitoba for additional technical support and mentoring. We thank Dr. David Schibli, Jun Han, and the Metabolomics Innovation Center (TMICS) Victoria Node at the University of Victoria Genome BC Proteomics Center for performing the mass spectrometry isotometric analysis. This work was funded by a Project Grant from the Canadian Institutes of Health

Research (T.S. & T.E.W.O.), a Discovery Grant from the Natural Sciences and Engineering Research Council (T.S.), an Emerging Scholar Award grant from the Canadian Cancer Society (T.S.), a Cancer Research Society grant (T.S.), a Health Sciences Center Foundation grant (T.S.), and a Manitoba Medical Services Foundation grant (T.S.). E.M. was supported by a Graduate Student Health Research Studentship Award from Research Manitoba. V.B. is supported by CancerCare Manitoba.

## Author contributions

Conceptualization: T.S., E.M. Investigation: E.M., T.S., H.K., E.K., H.S., S.R.C., L.C.M., A.F., J.Z. Formal analysis: E.M. Resources: T.S., T.E.W.O., S.K.S., V.B. Writing—original draft: E.M., T.S. Writing—review & editing: E.M., T.S., H.K., E.K., H.S., C.V., T.E.W.O., V.B., S.K.S. Project administration: T.S., T.E.W.O., V.B., S.K.S., C.M.A. Supervision: T.S., T.E.W.O., V.B., S.K.S., C.M.A. Funding acquisition: T.S., T.E.W.O., V.B.

## Competing interests

The authors declare no competing interests.
