## [Peer Review File · Nature Communications]

Metabolism-based targeting of MYC via MPC-SOD2 axis-mediated oxidation promotes cellular differentiation in group 3 medulloblastomaREVIEWER COMMENTS

Reviewer #1 (Remarks to the Author):

The manuscript by Martell and co-authors investigated whether leveraging metabolism-targeting interventions can modulate MYC abundance and activity in G3 MB with the final aim to explore whether this could be a viable approach to target these tumour. They show a novel and clinically targetable role for OXPHOS metabolism in maintaining MYC abundance via the MPC-SOD2 axis in G3 MB. This is an interesting study with an excellent degree of novelty which provides fundamental pre-clinical knowledge paving the way for further translational exploration aiming at assessing the potential value of applying this approach in the clinic in the future. Having said that, there are a few critical methodological issues which has to be addressed prior to considering this work further for publication.

Major points:

Appropriate protein loading controls must be shown for all western blots. Ponceu S is not an adequate protein loading control, which size of the non-specifically Ponceau-stained protein have been used for assessment? was it always the same for all the blots? Also, how many times were the western blots repeated? There is no quantification presented so in theory it is possible each of them was carried out only once; quantification (with standard error and statistical analysis) of the WBs (n=3 biological replica) is needed. These are fundamental methodological issues which must be addressed experimentally to ensure the conclusion the authors are drawing are robust.

I am not convinced by the timeline of the in vivo experiments. Pre-clinical in vivo data must mimic the clinical situation as best as possible to ensure the results are potentially translatable. No one will ever treat a patient prior to the tumour being big enough to cause symptoms and be diagnosed. What is the rationale for treating the xenografted mice 5 days after implantation of the MB cells? Unless the authors can show that there is already a well formed tumour at this timepoint, this experimental design is not conducive to model tumour treatment.

Other points:

Figure 1F: Was this observation validated in other datasets, ideally those with larger tumours numbers. Also, can the enrichment of these pathways be predicted from RNAseq data?

Figure 1H-I: The authors should show that the pathways are indeed inhibited upon treatment with the different agents? This is to ensure that effect (or no effect) seen on MYC level is not related to a suboptimal activity of the inhibitor used. Is there any effect on MAX level of expression? Is there any non MYC amplified G3 lines which can be used to validate the findings?

Figure 3B: what is the mechanism mediating the increased quantity of TP53? Is MILIP expression reduced in IACS-01075 treated cells?

Figure 3C: is it expected that MYC mRNA level is not reduced upon treatment? What is the authirs interpretation of this finding?

Figure 4A: It would be useful to plot control and treated cells respirometry's profiles in the same graph so the reader can better appreciate the differences between the two conditions.

Figure 4I-J: It would be interesting to know if ROS signaling influences MYC levels also in the other MYC-amplified tumour cell lines used in Figure 1/S1. This could highlight a possible MB-specific mechanism of regulation of MYC.

Figure 5I-J: It is not clear what are the first samples loaded in each of the blot? From the legend above (with + and -) it looks like no input and no pull down was loaded here.

Also, what type of quantification/normalization was done? Ponceau S is not enough.

Can the authors show whether the impairment of MYC-oxidation blocks the mechanism they describe? It would seem a crucial point to support their claim "These findings demonstrate a novel oxidative post-translational modification of the MYC protein that may have important physiological and pathological roles in regulating MYC stability".

Figure 6 E, G and H and Figure S6: Quantification and a proper normalization of WBs is needed to claim any conclusions, particularly if phosphorylation or post-translational events are studied.

Figure 7 A and B: As above. From the only blot presented here the repression or restoration claimed by the authors is not clear.

Minor points

Figure 1G is not really useful

Figure S4C: legend (with + or -) is not present for all the samples

Figure 6F: increase size of fonts used and reduce the surrounding scheme of mitochondria that is not really needed

Page 16: "we found that the inhibitory phosphorylation of PDH at Ser293 decreased following IACS-010759 treatment, which is established to promote the conversion of pyruvate into acetyl-CoA (Fig. 7F & G)." reference to the figure is not correct.

Page 17: "rescued the levels of the stemness factor SOX2 while suppressing the differentiation marker β 3-tubulin/TUBB3 following IACS-010759 treatment (Fig. 8F & G)." reference to the figure is not correct.

Reviewer #2 (Remarks to the Author):

In this work, the authors suggest that MYC-overexpressed medulloblastomas (MB) are hypersensitive to inhibitors of oxidative phosphorylation. Starting from proteomics analyses of several tumor samples, they demonstrated enrichments in metabolic proteins in general and oxphos in particular. When different metabolic inhibitors were tested on MB cell lines, only oxphos inhibitors demonstrated specific toxicity to MYC expressed cell lines. The authors took advantage of a clinically relevant compound IACS-010759 (IACS), which is a relatively specific inhibitor of complex I of the respiratory chain, to further explore the liability of MYC overexpressed tumors on oxphos and the specific therapeutic potential of IACS. Further, they suggested the MYC expression relies on oxphos due to protection from MYC oxidation, and hence IACS mode of action is by oxidation of MYC and targeting it to proteasomal degradation. The authors further suggested that MYC oxidation is caused due to the acetylation and inhibition of SOD2.

This is a robust study which is aimed not only at demonstrating therapeutic potential of IACS in MYC overexpressed tumors in general, but also at mechanistic understanding of the IACS mode of action. However, there are multiple issues that should be addressed.

1. Several metabolic pathways were suggested to be activated in MYC-transformed MB tumors following the proteomics study. But it is not well explained how these observations led the authors to hypothesize that blocking those metabolic pathways would lower MYC levels, particularly as most of these genes are known MYC targets - hence downstream of MYC?
2. The authors used a battery of metabolic blockers to demonstrate which of the upregulated metabolic pathway is also essential for MYC overexpressed MB cells. However, they did not demonstrate that any of the compound used metabolically affected the cells as expected. Metabolic validation should be included. Later indeed they showed the effect of IACS on oxidative phosphorylation, but that was a study on permeabilized cells (practically a biochemical assay on mitochondria).
3. Much of the mechanistic work was done in MYC overexpressed cells. However, the authors did not demonstrate that like in tumors, there is a MYC-dependent (or correlated) increase in metabolic proteins, and they did not directly link increased oxphos in these cells to MYC overexpression.
4. p53 is known to be upregulated due to many reasons. The link between IACS and p53 may or may not be related to MYC expression (downregulation of MYC and MILIP as the authors suggested). IACS may cause DNA damage (via ROS production?), or p53 phosphorylation and stabilization in any other MYC-independent manner. Indeed, the authors later showed that exogenous MYC overexpression prevented p53 induction by IACS, but this again may be related to the ROS/DNA damage or other mode of p53 induction, and not directly to the MYC-MILIP proposed pathway.

5. ROS production in general, and on mitochondria in particular, would have a general effect on mitochondrial TCA cycle and oxphos, hence the effect of IACS may be further expanded. In the Oroboros study indeed it seems that complex II is also significantly affected. Further, as mentioned above, this technology is relying on studying mitochondrial activity under artificial conditions where oxidizable substrates are added stepwise. Also, ADP, which is essential for oxphos activity is only added once through the study, and may be fully converted to ATP which would lead to further decrease in downstream complex analyses. In short, it would be far more informative to show that IACS actually blocks respiration, and potentially induces compensatory glycolytic flux using intact cells.

6. The proposed mechanism of IACS function in the MYC expressed cells is to target MYC for proteasomal degradation. It is therefore unclear why exogenous MYC can rescue the IACS phenotype and how the exogenous MYC is not subjected to this degradation. Indeed, the cells massively overexpressed the protein, but can it stoichiometrically avoid oxidation and degradation? Many labile proteins that are targeted for proteasomal degradation such as p53 and HIF1 α are difficult to be overexpressed exogenously unless the mode of inducing their degradation is also manipulated.

7. According to the authors' proposed mode of action, the inhibition of SOD2 directly (not with IACS) should have a similar effect on MYC levels and on cell death in MYC expressed cells. Can the authors demonstrate this?

8. Blocking pyruvate import to mitochondria would additionally impact oxphos. Therefore, it seems somewhat conflicting that two compounds that block oxphos (IACS and UK5099) would contradict each other's effect on MYC and cell growth. The authors suggest that while IACS increase pyruvate-dependent acetylation, UK5099 block it. However, blocking NADH oxidation by IACS would increase mitochondrial NADH and hence will inhibit PDH activity which is required for pyruvate dependent acetylCoA production. This conflict must be explained. Most importantly, it remained to be seen that IACS actually increases the metabolic flux of pyruvate to acetylCoA?

Reviewer #3 (Remarks to the Author):

Group 3 medulloblastomas (MB) are generally associated with the over-expression of Myc and carry the worst prognosis of all MB subgroups. The work/model presented by Martell et al describes a plausible mechanism by which Myc protein in these tumors is allowed to accumulate. Briefly, the authors demonstrate that inhibition of Complex I of the ETC leads to an accumulation of AcCoA, which then inactivates SOD2 via the inhibitory acetylation of K12/K68. The loss of SOD2 activity leads to an accumulation of ROS and the oxidation-mediated degradation of Myc. The latter occurs in a manner that is dependent on the intactness of the mitochondrial pyruvate carrier (MPC), which supplies the mitochondrial pyruvate from which AcCoA is derived.

Overall, this is a very well-written, concise and easy to follow study that offers a fresh take on the role of Myc in Group 3 MBs specifically and is even potentially applicable to other tumors in which Myc is over-expressed. In this approach, the authors have asked how the abnormal metabolism of a tumor cell might work to stabilize Myc, thus allowing it to accumulate to even higher levels. They describe a novel mechanism whereby inhibiting complex-I of the ETC causes the inhibitory acetylation of SOD2, leading to an aberrant accumulation of ROS and the oxidation of Myc. This in turn is followed by Myc's rapid proteasome-mediated degradation in a manner that is dependent on the mitochondrial pyruvate carrier. Complex I inhibition could thus be viewed as an indirect means of inhibiting Myc that would circumvent the problems associated with its more direct inhibition.

Major points

1. The statement that 10058-F4 inhibited normal astrocytes to the same degree as it did MB cells is flawed for two reasons. First, exceedingly high concentrations (150 μ M) of 10058-F4 were used, which is likely associated with multiple non-specific effects. Most published studies with 10058-F4 have used concentrations of <50 μ M. Depending on the cell line, concentrations as low as 10 μ M

have been shown to be very inhibitory against tumor cell lines without affecting normal cells. Second, the experiment using these lower concentrations should be conducted over several days, not 24 hr. Such short treatment times almost certainly under-estimates the efficacy of 10058-F4.

2. In Fig. 1H&I, the authors claim that only Complex I inhibitors effectively inhibited Myc. No information is provided as to how much these various compounds also inhibited growth. Most Myc inhibitors that have been reported over the years are associated with significant losses of Myc expression (10058-F4 is only one such example). However, the reason is most likely due to the fact that the loss of Myc-Max dimerization leads to growth inhibition and quiescence, which in turn lead to a compensatory turning off of Myc. Thus, the loss of Myc expression is not a direct result of the inhibitor. The results in this figure need to be interpreted in the context of how much cell proliferation was inhibited by each of the above inhibitors....Indeed, the authors go on to show in Fig. 2 that 010759 is an effective suppressor of MB proliferation and is also effective against several other non-brain tumor cancer cells. It seems likely that 101759 leads to a significant impairment of ATP production given that it is a Complex I inhibitor.

3. On p. 9. The authors state: "we found that IACS- 010759 treatment decreased the proportion of self-renewing stem cell populations in HD-MB03 and SU_MB002 G3 MB cells, which coincided with a decrease in the levels of the stemness transcription factor SOX2 (Fig. 2G & H)." As discussed above, it is not clear whether this statement is true. If stem cells are unable to proliferate, then they will be unable to give rise to the cells that comprise tumor spheres. Therefore, it seems possible that the observed results are not due to a quantitatively "decreased proportion of self-renewing stem cells" as much as to an inability of these cells to maintain a state of proliferation. Sox2, Nanog and NES are cell cycle regulated so their loss of expression (Fig. 2H) could be due either to loss of the stem cell population, to their inability to maintain a state of proliferation or to some combination of the two.

4. The inhibition of Complex I likely does much more than simply allow Myc to be degraded due to accumulating oxidative damage. Among the most likely of the consequences is an inhibition of mitochondrial ATP production, which would be consistent with the observed aberrant ROS production. Indeed, they show this to be the case in Fig. 4A. A wide range of Myc inhibitors, regardless of their structure, have been previously shown to inhibit ATP production in association with cell cycle arrest and differentiation of certain tumor cells (Oncotarget. 2015 Jun 30;6(18):15857-70). Moreover, ATP depletion alone, just like Myc inhibition, is sufficient to induce cell cycle arrest and induce differentiation of certain tumor cells, without any effect on Myc levels. This leads one to wonder how much of the anti-tumor effect the authors see in MB cells is due to Myc depletion versus other mechanisms, esp. those related to an overall depletion of ATP. The authors themselves have demonstrated that genes involved in OXPHOS are among the most highly enriched following treatment of MB cells with 101759. The ability to restore the growth of MB cells almost to normal while maintaining them in 101759 (Fig. 3E) is an important experiment as it suggests that the growth inhibition of the cells is truly due the inhibition of Myc rather than Complex I. However, it still strikes me as odd that inhibition of Complex I to a degree that significantly reduces ATP levels allows for normal rates of growth when Myc is restored. This is even more reason for the authors to measure ATP levels in cells (and ROS levels as well) in cells that are treated with 010759 but in which Myc has been restored. I would speculate that the re-expression of Myc may restore normal Complex I activity and normalize both ATP and ROS levels. How this would be done in the face of 010759 is unclear since the precise mechanism of 010759 is unlike. They should also mention how long the cells shown in Fig. 3E were maintained under these conditions and perhaps even show actual growth curves than single points as they do.

5. Fig. 5E purports to show evidence for Myc being ubiquitylated. But in the lower panel (Myc IB), I fail to see any evidence for the "laddering" due to this post-translational modification. Similarly, in panels F & G, although there IS an increase in Myc protein following proteasome inhibition, I do not see an accumulation of higher Mw forms of of Ub-Myc that one would expect.

6. The results in Fig. 5I&J are indirect in that they do not directly measure oxidized cysteines in Myc; rather, they measure only the amount of Myc protein (presumably unoxidized) that remains following 101759 treatment. In addition, the studies do not demonstrate which of the 10 cysteine residues present in Myc are actually being oxidized and are responsible for changing its half-life.

These are really two separate questions as all 10 residues could be highly sensitive to oxidation, with only one or two being responsible for the half-life change. A couple of approaches that provide different answers to these questions could be used to investigate this. First, if possible, one could isolate Myc protein from the cells (assuming there is enough of it) and subject it to mass-spec to identify specific oxidized/unoxidized cysteine residues. Alternatively (and less satisfying), one could purify recombinant Myc protein and then subject it to an oxidative challenge in vitro (for example H₂O₂) and then perform MS. Second, one could express individual cysteine point mutants of Myc to determine which one(s) is (are) the most responsive to destabilizing oxidation. It seems that the most plausible prediction is that there are fewer cysteine residues that are necessary for determining Myc half-life than are actually oxidized.

7. The implication of much of the work presented in Fig. 7 (and consistent with the simplest model) is that a combination of altered Complex I, PDH and MPC activities are responsible for an accumulation of intra-mitochondrial acetylCoA, which in turn is responsible for the acetylation of SOD2 (a mitochondrial protein). However, the authors present an incomplete story of a potentially complicated mechanism. First, they show only changes in the levels of pPDH, which are consistent with an increase in its activity. However, they do not measure PDH activity directly, which can easily be done by examining the conversion of ¹⁴C-pyruvate to ¹⁴C¹⁴O₂. This would potentially allow them to place their MPC and pPDH changes in context. Most importantly, they do not directly measure the levels of acetylCoA in mitochondria. Finally, they fail to account for the possibility of alternate sources of acetylCoA other than pyruvate, most likely from an increase in the activity of FAO, which can be measured directly as well using labeled fatty acids such as palmitate. Defective mitochondria, most notably those resulting from loss of Myc are well known to compensate for their OXPHOS defects by increasing their reliance on FAO. Finally, the inferred change in PDH activity based on its reduced inhibitory phosphorylation might not necessarily correlate with acetylCoA levels; for example, it could be a compensatory mechanism to try to rectify severely compromised levels of acetylCoA as a result of a Complex I defect. This is reasonable since AcCoA regulates PDH activity indirectly via its allosteric inhibition/activation of PDK1 and PDP2.

8. Fig. 8 shows the tumors of 010759-treated mice to be significantly reduced at 18 days. The authors present the difference in tumor size as areas (mm²) rather than as volumes (mm³) and this should be corrected as the differences will be even more impressive than the 7-8-fold they claim based on area. Alternatively, the tumors could be weighed. More importantly, this raises the question of why the treated animals only lived for an additional ~10 days given the large size differences seen at day 18. What were the sizes of the tumors in the treated mice at the time of death? Presumably, they were at least as large as those in the d18 control animals, but if not, then other mechanisms of 010759 action in vivo need to be considered. If they were as large as d18 untreated tumors, then what accounted for the sudden increased rate of tumor growth between d18 and d28? Also, can they demonstrate an increase in either SOD2 acetylation or Myc oxidation as they were able to show in vitro?

Minor points

1. At least two references should be replaced:

-Bottom of p. 3 "...clinical targeting of MYC has remained elusive (9)". There are better and more recent refs/reviews to cite regarding the elusiveness of current therapeutic approaches.

-Bottom of p. 5: Ref. 19 was not the study demonstrating that 10058-F4 directly inhibits Myc-Max heterodimerization by binding directly to Myc. The correct reference for this is: *Mol Cancer Ther.* 2007 Sep;6(9):2399-408.

2. Bottom of p.10: "This activation of WT TP53 offers an additional benefit for IACS-010759 treatment in G3 MB patients, as it is a desirable therapeutic outcome to suppress tumor growth." This statement should be removed or modified as it implies that this effect on p53 is somehow specific for 010759 when in fact the effect is due to the down-regulation of MILIP, which is just one of many Myc target genes.

3. The Myc protein half-life in control MBO3 cells (Fig. 5D) is > 2hr and is convincingly reduced to ~30 min following 101759 addition. However 2 hr is an extremely long half-life for Myc by most standards (actually about 10-times longer). Is the prolonged half-life seen in control MB03 cells

specific for MBO3 or is it seen in other MBs? Perhaps MBO3 cells have a defect other than gene amplification that accounts for their high Myc level expression.

REVIEWER #1 (REMARKS TO THE AUTHOR):

The manuscript by Martell and co-authors investigated whether leveraging metabolism-targeting interventions can modulate MYC abundance and activity in G3 MB with the final aim to explore whether this could be a viable approach to target these tumour. They show a novel and clinically targetable role for OXPHOS metabolism in maintaining MYC abundance via the MPC-SOD2 axis in G3 MB. This is an interesting study with an excellent degree of novelty which provides fundamental pre-clinical knowledge paving the way for further translational exploration aiming at assessing the potential value of applying this approach in the clinic in the future. Having said that, there are a few critical methodological issues which has to be addressed prior to considering this work further for publication.

Major points:

Appropriate protein loading controls must be shown for all western blots. Ponceu S is not an adequate protein loading control, which size of the non-specifically Ponceau-stained protein have been used for assessment? was it always the same for all the blots? Also, how many times were the western blots repeated? There is no quantification presented so in theory it is possible each of them was carried out only once; quantification (with standard error and statistical analysis) of the WBs (n=3 biological replica) is needed. These are fundamental methodological issues which must be addressed experimentally to ensure the conclusion the authors are drawing are robust.

We emphatically agree with the reviewer that proper loading controls are critical for the proper analysis and interpretation of western blots. Therefore, we opted to use a total protein loading control such as Ponceau stain for all protein normalization as opposed to traditional housekeeping proteins such as β -Actin or GAPDH, as the levels of these proteins can sometimes be altered under conditions where there are metabolic alterations or changes in cell morphology during differentiation (Romero-Calvo I, et al. *Anal Biochem.* 2010¹). Due to space constraints, we have presented cropped representative bands of the whole Ponceau stained blots in the main figures, however, our quantifications for **all blots are normalized to the total levels of Ponceau S protein in the whole lane and not just arbitrary bands.** As per the editorial policies of *Nature Communications*, we have now included the total uncropped blot images of all Western blot and Ponceau S images in the Source Data Supplementary File. Moreover, we can confirm that all Western blots have been repeated at least $n=3$ times, and quantifications of all blots with standard error bars and statistical analyses are now present in either the Main or Supplementary Figures of the revised manuscript.

I am not convinced by the timeline of the in vivo experiments. Pre-clinical in vivo data must mimic the clinical situation as best as possible to ensure the results are potentially translatable. No one will ever treat a patient prior to the tumour being big enough to cause symptoms and be diagnosed. What is the rationale for treating the xenografted mice 5 days after implantation of the MB cells? Unless the authors can show that there is already a well formed tumour at this timepoint, this experimental design is not conducive to model tumour treatment.

We agree with the Reviewer that pre-clinical *in vivo* models must try to mimic the clinical scenario as best as possible. Human tumor models in immune-compromised mice are often highly aggressive and progress far quicker than in patients and the animals typically don't present any 'symptoms' until near disease endpoint, as is the case with the HD-MB03 medulloblastoma tumor model. When HD-MB03 cells are transplanted into the cerebellum of immune-compromised mice to create an orthotopic model

of G3 MB, the typical survival time is between 18-25 days without any treatment (Milde T et al., *J Neurooncol.* 2012²; Zagozewski J et al., *Nat Commun.* 2020³). Small tumors can be visible by MRI imaging in this model at 5 days post-surgery, and the tumor development is even more clear by H&E staining of tumor tissues collected at this time point (**Fig 9B**). Tumors continue to grow and are even more well-formed at 10 days post-surgery (**Fig 9B**). With this in mind, we have repeated our animal study and postponed the administration of IACS-010759 treatment until 10 days post-tumor cell implantation, at approximately the halfway point of disease progression. With this new treatment protocol, we found that our treatment was equally as effective at slowing tumor progression and significantly prolonging animal survival (**Fig. 9C & Fig. S9C**). We thank the Reviewer for this suggestion, as this has strengthened our findings and added further confidence in the potential clinical translatability for the use of this treatment in G3 MB patients.

Other points:

Figure 1F: Was this observation validated in other datasets, ideally those with larger tumour numbers. Also, can the enrichment of these pathways be predicted from RNAseq data?

We have now validated our GSEA in an additional larger RNA-sequencing dataset from Cavalli et al., *Cancer Cell*, 2017⁴. This dataset contains samples from a total of 763 medulloblastoma tumors, including 144 G3 MB tumors. These G3 samples have been previously characterized into MYC-amplified subtypes (G3gamma) and non-MYC-amplified (G3alpha and beta). We performed GSEA on G3 subgroup samples (n = 144) and confirmed that the MYC-amplified subtype is enriched in hallmark MYC targets gene signatures (**Fig. S1D**). We also observed a trend towards increased enrichment in MYC-amplified G3 MB tumors of the described metabolic processes including glycolysis, glutaminolysis, fatty acid oxidation, and oxidative phosphorylation. Although the enrichment of these signatures was not statistically significant, this could be reasonably explained due to the fact that RNA and protein levels do not always correlate perfectly, and there are many other factors that influence RNA transcription of these metabolic genes. Moreover, these findings lend further credence that the protein post-translational modifications described in our study may play significant role in the correlation between metabolism and MYC expression.

Figure 1H-I: The authors should show that the pathways are indeed inhibited upon treatment with the different agents? This is to ensure that effect (or no effect) seen on MYC level is not related to a suboptimal activity of the inhibitor used. Is there any effect on MAX level of expression? Is there any non MYC amplified G3 lines which can be used to validate the findings?

We have now included additional metabolic validation in the revised version of this manuscript which confirms that all the inhibitors used in this study are blocking the intended metabolic pathways at the doses tested (**Fig. 1E**). We have also demonstrated that MAX expression follows a similar pattern as MYC following treatment with metabolic inhibitors in G3 MB cells. MAX levels do not decline following treatment with glycolysis, fatty acid oxidation, or glutaminolysis inhibitors but are significantly decreased following treatment with complex-I inhibitors (**Fig. S1E & Fig. S1F**).

Figure 3B: what is the mechanism mediating the increased quantity of TP53? Is MILIP expression reduced in IACS-01075 treated cells?

We have now confirmed that *MILIP* expression is indeed reduced following IACS-010759 treatment in G3 MB cells and that the expression of MILIP can be restored by re-instating MYC levels (**Fig. 3A &**

Fig. 3C). These findings indicated that the MYC-MILIP pathway may play an important role in the regulation of TP53 in IACS-010759 treated cells although we do not rule out the possibility that other mechanisms may also be contributing to TP53 modulation.

Figure 3C: is it expected that MYC mRNA level is not reduced upon treatment? What is the authors interpretation of this finding?

MYC expression can be regulated by a variety of different mechanisms. Many tumor cells carry genomic amplifications in the *MYC* gene, which allows them to produce many copies of *MYC* mRNA transcripts and keep MYC protein levels elevated. It is also common for tumors to leverage mechanisms which prolong MYC protein half-life and prevent the degradation of MYC so that there is less protein turnover. Tumor cells can also use a combination of these mechanisms to keep MYC levels high. In G3 MB, amplifications of *MYC* are common, and the G3 MB models used in this manuscript (HD-MB03 and SU_MB002), carry genomic *MYC* amplifications. Therefore, it is not so surprising that MYC protein levels may be decreased following IACS-010759 treatment due to increased proteasomal degradation, while mRNA levels remain stable due to the constant supply of mRNA transcripts produced by the multiple genomic copies of *MYC*.

Figure 4A: It would be useful to plot control and treated cells respirometry's profiles in the same graph so the reader can better appreciate the differences between the two conditions.

Due to the way in which the Oroboros instrument is designed, it requires each sample to be run in separate chambers with individual sensors. Substrates and inhibitors are added to each individual chamber sequentially and each chamber generates its own independent reading, therefore they cannot be directly plotted by overlaying on same graph. We have included the quantifications with statistical analyses of the peak values corresponding the basal OCRs, maximal OCRs, and substrate-specific OCRs, so that the differences between the two conditions are clearly presented to aid the reader (**Fig. 4A & Fig. 4B, Fig. S4A & Fig. S4B, Fig. S5B & S5C**).

Figure 4I-J: It would be interesting to know if ROS signaling influences MYC levels also in the other MYC-amplified tumour cell lines used in Figure 1/S1. This could highlight a possible MB-specific mechanism of regulation of MYC.

We would like to thank the reviewer for this excellent suggestion. We have now repeated this experiment testing whether the mitochondrial antioxidant mimetic MitoTEMPO can restore MYC abundance following IACS-010759 treatment in other MYC-amplified cancer cell lines including: the ovarian carcinoma cell line A2780, the breast cancer cell line MDA-MB-468 and the colorectal cancer cell line SW480. Consistent with our observations in group 3 MB cells, we found that scavenging mitochondrial ROS restored MYC protein levels in these additional cancer cell lines, highlighting that this mechanism may also be applicable in various cancer types (**Fig. S4G**).

Figure 5I-J: It is not clear what are the first samples loaded in each of the blot? From the legend above (with + and -) it looks like no input and no pull down was loaded here. Also, what type of quantification/normalization was done? Ponceau S is not enough.

We sincerely apologize, this figure was mislabelled and the legend above the blot has now been amended. We thank the review for bringing this error to our attention and we are sorry for any confusion this caused.

In **Figures 5I-J**, reduced cysteine thiols were labeled with a maleimide-PEG2-biotin tag and proteins with reduced cysteines were precipitated using streptavidin agarose beads and run alongside total protein input controls. The same amount of protein that was subjected to streptavidin precipitation (50 ug) was run in the input lane, therefore the percentage of reduced MYC versus total MYC protein could be quantified. Proteins were also normalized to the total Ponceau S stain. Quantification calculations were performed as follows:

$$\left(\frac{\text{Densitometry of reduced MYC protein levels}}{\text{Densitometry of Ponceau S stain}} \right) \div \left(\frac{\text{Densitometry of total MYC protein levels}}{\text{Densitometry of Ponceau S stain}} \right)$$

Can the authors show whether the impairment of MYC-oxidation blocks the mechanism they describe? It would seem a crucial point to support their claim “These findings demonstrate a novel oxidative post-translational modification of the MYC protein that may have important physiological and pathological roles in regulating MYC stability”.

Using various rescue experiments, including blocking the accumulation of ROS using the antioxidant MitoTEMPO and preventing SOD2 acetylation using the MPC inhibitor UK-5099, we have found that these manipulations effectively impair IACS-mediated MYC oxidation and subsequently restore MYC levels and downstream phenotypes (i.e., stemness and differentiation; **Fig. 6A-C & Fig. 8E-G**). Moreover, through a concerted effort we have now generated MYC plasmid constructs with point mutations in all 10 individual MYC cysteine residues and elucidated the cysteine oxidation sites that are important for mediating MYC degradation following IACS treatment. Mutations in cysteines C148, C203, C315, C357 and C453 significantly blocked MYC protein oxidation and impaired MYC protein degradation in IACS treated cells (**Fig. 6E**). Altogether, these findings highlight the importance of MYC cysteine oxidation in regulating its protein stability in group 3 MB.

Figure 6 E, G and H and Figure S6: Quantification and a proper normalization of WBs is needed to claim any conclusions, particularly if phosphorylation or post-translational events are studied.

We agree with the reviewer, and we have provided graphs of densitometry quantifications from $n=3$ replicates of all western blots in the revised version of this manuscript. For the western blots from the original Figures 6E, G, H and Figure S6 (now corresponding to **Fig. 7A, 7B, 7G & Fig. S8A** in the revised version of this manuscript), the post-translational modifications of proteins have been normalized to the total levels of the respective proteins and quantifications are present in **Fig. 7A-B, Fig. S7D & Fig. S8A**) in the revised manuscript.

Figure 7 A and B: As above. From the only blot presented here the repression or restoration claimed by the authors is not clear.

As requested, we have provided graphs of densitometry quantifications from $n=3$ replicates of all western blots in the revised version of this manuscript. The quantifications of western blots from the original Figures 7A and B (now corresponding to **Fig. 8A & Fig. 8C** in the revised version of this manuscript), are present in **Fig. S8B & Fig. S8C** in the revised manuscript.

Minor points

Figure 1G is not really useful

This schematic has now been removed in the revised version of this manuscript.

Figure S4C: legend (with + or -) is not present for all the samples

We apologize for this error, this figure was mislabelled and the legend has now been amended (**Fig. S5D**). We thank the review for bringing this error to our attention and we are sorry for any confusion this caused.

Figure 6F: increase size of fonts used and reduce the surrounding scheme of mitochondria that is not really needed

We have made these adjustments suggested by the reviewer and increased font sizes and reduced the size of the schematic (now presented in **Fig. 7F**).

Page 16: “we found that the inhibitory phosphorylation of PDH at Ser293 decreased following IACS-010759 treatment, which is established to promote the conversion of pyruvate into acetyl-CoA (Fig. 7F & G).” reference to the figure is not correct.

We sincerely apologize for this error and we have corrected the figure reference in the revised version of this manuscript.

Page 17: “rescued the levels of the stemness factor SOX2 while suppressing the differentiation marker β 3-tubulin/TUBB3 following IACS-010759 treatment (Fig. 8F & G).” reference to the figure is not correct.

We sincerely apologize for this error and we have corrected the figure reference in the revised version of this manuscript.

REVIEWER #2 (REMARKS TO THE AUTHOR):

In this work, the authors suggest that MYC-overexpressed medulloblastomas (MB) are hypersensitive to inhibitors of oxidative phosphorylation. Starting from proteomics analyses of several tumor samples, they demonstrated enrichments in metabolic proteins in general and oxphos in particular. When different metabolic inhibitors were tested on MB cell lines, only oxphos inhibitors demonstrated specific toxicity to MYC expressed cell lines. The authors took advantage of a clinically relevant compound IACS-010759 (IACS), which is a relatively specific inhibitor of complex I of the respiratory chain, to further explore the liability of MYC overexpressed tumors on oxphos and the specific therapeutic potential of IACS. Further, they suggested the MYC expression relies on oxphos due to protection from MYC oxidation, and hence IACS mode of action is by oxidation of MYC and targeting it to proteasomal degradation. The authors further suggested that MYC oxidation is caused due to the acetylation and inhibition of SOD2.

This is a robust study which is aimed not only at demonstrating therapeutic potential of IACS in MYC overexpressed tumors in general, but also at mechanistic understanding of the IACS mode of action. However, there are multiple issues that should be addressed.

1. Several metabolic pathways were suggested to be activated in MYC-transformed MB tumors

following the proteomics study. But it is not well explained how these observations led the authors to hypothesize that blocking those metabolic pathways would lower MYC levels, particularly as most of these genes are known MYC targets - hence downstream of MYC?

We apologize for the lack of clarity in the explanation of the rationale behind our hypothesis. As the reviewer mentioned, the role of MYC in regulating metabolism is well-established and we acknowledge that the genes involved in the metabolic pathways which we observed were enriched in **Fig. 1D** are known targets of MYC. Although the many pathways and processes that MYC regulates have been extensively investigated, however, the various mechanisms that in turn modulate MYC expression and stability are less well characterized. Given that MYC is so important for regulating metabolic processes, it is not unreasonable to assume that some of these processes may play a reciprocal role in supporting MYC levels as an intrinsic feedback mechanism. Such reciprocal regulatory relationships have been demonstrated for other major transcription factors such as TP53, which has been shown to regulate the transcription of NAD⁺ synthesizing enzymes and these enzymes in turn modulate TP53 stability by mediating the activity of NAD⁺-dependent deacetylase enzymes (Pan LZ et al., *Cell Cycle*. 2014⁵; Liu J et al., *Elife*. 2021⁶). We have modified our explanation of our hypothesis in the revised version of this manuscript to increase clarity (**pg. 4, lines 17-21**).

2. The authors used a battery of metabolic blockers to demonstrate which of the upregulated metabolic pathway is also essential for MYC overexpressed MB cells. However, they did not demonstrate that any of the compound used metabolically affected the cells as expected. Metabolic validation should be included. Later indeed they showed the effect of IACS on oxidative phosphorylation, but that was a study on permeabilized cells (practically a biochemical assay on mitochondria).

This is an excellent point brought forward by the reviewer. We have now performed extensive metabolic characterization of all the inhibitors used in this study to validate their efficacy at the chosen doses (**Fig. 1E**). We have confirmed that treatment with;

- (i) Treatment with CB-839, which inhibits the activity of GLS1 to block the conversion to glutamine to glutamate to inhibit glutaminolysis, leads to lowered glutamate levels in G3MB cells at 2.5 and 5 μ M.
- (ii) GW9662, an inhibitor of the transcription factor PPAR γ which blocks the transcription of the FAO carrier CPT1A, leads to decreased CPT1A expression in G3 MB cells at 5 and 10 μ M doses.
- (iii) All glycolytic inhibitors tested (2-DG, BrPA, and siGAPDH) decreased lactate production at the indicated doses.
- (iv) Similar to our observations with the complex I inhibitor IACS-010759, we found that other ETC inhibitors (Phenformin and Rotenone) strongly hampered ATP production.

In addition to the previous metabolic characterization that we performed with IACS-010759 which showed that this compound blocks complex-I mediated oxygen consumption and suppresses ATP production, we performed further validation of the effect of IACS-010759 on maximal oxygen consumption rate on non-permeabilized intact cells using oroboros respirometry. In this assay, cells are treated with 100 nM of IACS-010759 for 24 hours and resuspended in their normal growth media then loaded into the oroboros chamber where they are exposed to increasing doses of the ETC uncoupler FCCP to determine the maximal oxygen consumption rate. We found that IACS-010759 drastically decreases the maximal OCR in HDMB03 G3 MB cells (**Fig. 4A & Fig. 4B**).

3. Much of the mechanistic work was done in MYC overexpressed cells. However, the authors did not demonstrate that like in tumors, there is a MYC-dependent (or correlated) increase in metabolic proteins, and they did not directly link increased oxphos in these cells to MYC overexpression.

Indeed, as this study is focused on G3 MB tumor cell models the cell lines available for this type of cancer are all MYC-amplified to some degree. Although we have found that some of these cells contain higher MYC protein abundance as compared to others, with the HDMB03 and SU_MB002 cell lines harboring the highest levels of MYC protein while D283 and MB3W1 contain relatively lower MYC expression (Fig. S2C). As per the reviewer's suggestion we sought to investigate whether these differences in MYC abundance could be correlated with OXPHOS activity. Indeed, we observed a positive correlation between the levels of MYC protein abundance and the maximal OCR of group 3 MB cells (Fig. S2D), as well as a positive correlation between MYC and several mitochondrial metabolic proteins (CS, SDHA, and FH; Fig. S2C). Altogether, these findings are in agreement with those observed in patient tumors where a high mitochondrial metabolic phenotype correlates with enhanced MYC abundance in group 3 MB cells.

4. p53 is known to be upregulated due to many reasons. The link between IACS and p53 may or may not be related to MYC expression (downregulation of MYC and MILIP as the authors suggested). IACS may cause DNA damage (via ROS production?), or p53 phosphorylation and stabilization in any other MYC-independent manner. Indeed, the authors later showed that exogenous MYC overexpression prevented p53 induction by IACS, but this again may be related to the ROS/DNA damage or other mode of p53 induction, and not directly to the MYC-MILIP proposed pathway.

We have performed qPCR analysis to examine the effect of IACS-010759 on *MILIP* expression in G3 MB cells to understand whether this proposed pathway is playing a role in the regulation of P53 levels. Indeed, we found that consistent with the downregulation of MYC levels observed following IACS-010759 treatment, we found that expression of the MYC-target *MILIP* is also significantly decreased (Fig. 3A). This decrease in *MILIP* expression may potentially alleviate the inhibition of TP53 and play a role in the subsequent upregulation of TP53 abundance in G3 MB cells. To further implicate the role of MYC in governing this regulation of *MILIP*, we found that replenishing MYC levels restored *MILIP* expression (Fig. 3C). These findings suggest that the MYC-*MILIP* axis may be playing a role in influencing TP53 levels in G3 MB cells following IACS-010759 treatment. However, we do not rule out that there may be additional MYC-independent mechanisms modulating TP53. As the reviewer mentioned, there are several processes which are known to regulate TP53 including ROS, DNA damage, etc. and it may be an accumulation of these mechanisms along with MYC-downregulation that ultimately activate TP53 following IACS-010759 treatment.

5. ROS production in general, and on mitochondria in particular, would have a general effect on mitochondrial TCA cycle and oxphos, hence the effect of IACS may be further expanded. In the Oroboros study indeed it seems that complex II is also significantly affected. Further, as mentioned above, this technology is relying on studying mitochondrial activity under artificial conditions where oxidizable substrates are added stepwise. Also, ADP, which is essential for oxphos activity is only added once through the study, and may be fully converted to ATP which would lead to further decrease in downstream complex analyses. In short, it would be far more informative to show that IACS actually blocks respiration, and potentially induces compensatory

glycolytic flux using intact cells.

As suggested by the reviewer, we have performed additional analysis regarding the effect of IACS-010759 on the maximal oxygen consumption rate (OCR) of intact, non-permeabilized cells and we have found that IACS-010759 treatment significantly impairs respiratory capacity of HD-MB03 G3 MB cells (**Fig. 4A & Fig. 4B**).

6. The proposed mechanism of IACS function in the MYC expressed cells is to target MYC for proteasomal degradation. It is therefore unclear why exogenous MYC can rescue the IACS phenotype and how the exogenous MYC is not subjected to this degradation. Indeed, the cells massively overexpressed the protein, but can it stoichiometrically avoid oxidation and degradation? Many labile proteins that are targeted for proteasomal degradation such as p53 and HIFa are difficult to be overexpressed exogenously unless the mode of inducing their degradation is also manipulated.

This is an excellent point mentioned by the reviewer. Indeed. We do observe that exogenous MYC is susceptible to IACS-mediated degradation, as highlighted in our newly added **Fig. 6H (i) & (ii)**, where when we introduce GFP-tagged wild-type MYC in HDMB03 cells, we see that IACS-010759 treatment significantly decreases the GFP signal, indicating degradation of exogenous MYC.

However, in our add-back study to investigate the impact of re-introducing MYC in IACS-010769-treated cells on cell growth, stemness, and transcription, we employed several strategies to minimize this effect:

1. Exogenous MYC is added after 12-hours of IACS-010759 treatment. This allows time for the degradation of endogenous MYC to occur whereas exogenous MYC may not be degraded as quickly.
2. Cells are given 1.5 µg/mL of MYC overexpression plasmid. As seen in **Fig. 3C & Fig. 3D** this leads to a very strong overexpression of MYC. Therefore, while some exogenous MYC may be subject to degradation, there is still sufficient MYC to restore levels above normal abundance and this has a functional impact on cellular phenotype as demonstrated in **Fig. 3E-G**).

Since we observed an abundant amount of MYC using this exogenous overexpression technique, this negates the need to additionally manipulate degradation mechanisms.

7. According to the authors' proposed mode of action, the inhibition of SOD2 directly (not with IACS) should have a similar effect on MYC levels and on cell death in MYC expressed cells. Can the authors demonstrate this?

This is an excellent question raised by the reviewer. To the best of our knowledge there are no specific SOD2 inhibitors available on the market. Therefore, to answer this question, we employed small-interfering RNA silencing of the SOD2 gene to transiently inhibit SOD2 expression and ultimately impair SOD2 antioxidant activity. We found that silencing of SOD2 indeed leads to increased accumulation of mitochondrial ROS which corresponds with a downregulation of MYC abundance and downstream MYC targets (**Fig. S7A-B**), consistent with our proposed mode of action. We sincerely thank the reviewer for this suggestion as the addition of this data has greatly strengthened our findings.

8. Blocking pyruvate import to mitochondria would additionally impact oxphos. Therefore, it seems somewhat conflicting that two compounds that block oxphos (IACS and UK5099) would

contradict each other's effect on MYC and cell growth. The authors suggest that while IACS increase pyruvate-dependent acetylation, UK5099 block it. However, blocking NADH oxidation by IACS would increase mitochondrial NADH and hence will inhibit PDH activity which is required for pyruvate dependent acetylCoA production. This conflict must be explained. Most importantly, it remained to be seen that IACS actually increases the metabolic flux of pyruvate to acetylCoA?

This is an excellent point raised by the reviewer. We agree that inhibition of complex I will likely inhibit NADH oxidation and increase mitochondrial NADH levels which is known to inhibit PDH activity under certain circumstances. However, complex I inhibition by IACS-010759 treatment also significantly impairs ATP generation, which itself can stimulate PDH activity via inhibition of PDK. Indeed, we found that the levels of phosphorylation at the auto-phosphorylation site Ser241 which is required for PDK1 activity are decreased following IACS-010759 treatment in HD-MB03 G3 MB cells (**Fig. S7C**). Our findings indicate that PDH activity is increased following IACS inhibition as the inhibitory phosphorylation of PDH is alleviated.

Additionally, as the reviewer suggested, we have performed a metabolic tracing experiment of U¹³C₃-pyruvate using mass spectrometry detection of central carbon metabolomics (CCM) and we observed that the accumulation of ¹³C-acetyl-CoA isotopologues is significantly increased in IACS-010759-treated cells compared to controls (**Fig. 7C**). Moreover, downstream labeling of TCA cycle intermediates with ¹³C was also increased following IACS-010759 treatment, indicating increased metabolic flux of pyruvate to Acetyl-CoA and TCA cycle metabolites (**Fig. 7D**). We also confirmed that total acetyl-coA levels are elevated in IACS-010759-treated group 3 MB cells using a fluorometric assay kit (**Fig. 7E**). Altogether, these findings lend further credence to support our claims that IACS-010759 treatment changes pyruvate metabolic dynamics that enhances SOD2 protein acetylation ROS production that ultimately leads to MYC oxidation and degradation in group 3 MB cells.

REVIEWER #3 (REMARKS TO THE AUTHOR):

Group 3 medulloblastomas (MB) are generally associated with the over-expression of Myc and carry the worst prognosis of all MB subgroups. The work/model presented by Martell et al describes a plausible mechanism by which Myc protein in these tumors is allowed to accumulate. Briefly, the authors demonstrate that inhibition of Complex I of the ETC leads to an accumulation of AcCoA, which then inactivates SOD2 via the inhibitory acetylation of K12/K68. The loss of SOD2 activity leads to an accumulation of ROS and the oxidation-mediated degradation of Myc. The latter occurs in a manner than is dependent on the intactness of the mitochondrial pyruvate carrier (MPC), which supplies the mitochondrial pyruvate from which AcCoA is derived.

Overall, this is a very well-written, concise and easy to follow study that offers a fresh take on the role in Myc in Group 3 MBs specifically and is even potentially applicable to other tumors in which Myc is over-expressed. In this approach, the authors have asked how the abnormal metabolism of a tumor cells might work to stabilize Myc, thus allowing it to accumulate to even higher levels. They describe a novel mechanism whereby inhibiting complex-I of the ETC causes the inhibitory acetylation of SOD2, leading to an aberrant accumulation of ROS and the oxidation of Myc. This in turn is followed by Myc's rapid proteasome-mediated degradation in a manner that is dependent on the mitochondrial pyruvate carrier. Complex I inhibition could thus

be viewed as an indirect means of inhibiting Myc that would circumvent the problems associated with its more direct inhibition.

Major points

1. The statement that 10058-F4 inhibited normal astrocytes to the same degree as it did MB cells is flawed for two reasons. First, exceedingly high concentrations (150 μ M) of 10058-F4 were used, which is likely associated with multiple non-specific effects. Most published studies with 10058-F4 have used concentrations of <50 μ M. Depending on the cell line, concentrations as low as 10 μ M have been shown to be very inhibitory against tumor cell lines without affecting normal cells. Second, the experiment using these lower concentrations should be conducted over several days, not 24 hr. Such short treatment times almost certainly under-estimates the efficacy of 10058-F4.

We thank the reviewer for bringing this point to our attention. Indeed, we initially tested the effect of multiple doses of 10058-F4 (5-150 μ M) on cell numbers after 24 hours and we found that low doses had no significant effect on either HDMB03 G3 MB cells or astrocytes at this time point (**Fig. S1B**). As suggested by the reviewer, we have now performed additional time-response analysis using low dose (10 μ M) of 10058-F4 on HDMB03 G3 MB cells or astrocytes and similarly found that this dose has no impact after 24 hours but significantly impaired the growth of both HDMB03 cells and astrocytes at later time-points (**Fig. 1C**). These findings strengthen our claim that while 10058-F4 is highly effective at suppressing the growth of MYC amplified G3 MB cells, it also displays some toxicity towards normal brain cell populations and may not be ideal for use in children with MB tumors.

2. In Fig. 1H&I, the authors claim that only Complex I inhibitors effectively inhibited Myc. No information is provided as to how much these various compounds also inhibited growth. Most Myc inhibitors that have been reported over the years are associated with significant losses of Myc expression (10058-F4 is only one such example). However, the reason is most likely due to the fact that the loss of Myc-Max dimerization leads to growth inhibition and quiescence, which in turn lead to a compensatory turning off of Myc. Thus, the loss of Myc expression is not a direct result of the inhibitor. The results in this figure need to be interpreted in the context of how much cell proliferation was inhibited by each of the above inhibitors....Indeed, the authors go on to show in Fig. 2 that 010759 is an effective suppressor of MB proliferation and is also effective against several other non-brain tumor cancer cells. It seems likely that 101759 leads to a significant impairment of ATP production given that it is a Complex I inhibitor.

This is an excellent point brought forward by the reviewer. As suggested, we have now performed additional cell count analysis following treatment with the various metabolic agents for the time period described in **Fig. 1E-G**. We found that inhibition of fatty acid oxidation (GW9662) or glycolysis (2-DG, BrPA, siGAPDH) did not impair HD-MB03 cell numbers (**Fig. S2A**). Interestingly, we found that the glutaminolysis inhibitor CB839 decreased cell numbers after 24 hours of treatment although in contrast to complex-I inhibitors, MYC levels remained intact despite this decrease in proliferation (**Fig. 1F-G; Fig. S2A**). These findings suggest that impairment of growth associated with metabolic inhibition does not necessarily correspond with regulation of MYC expression at early time points, and the modulation of MYC abundance appears to be a unique consequence associated with the metabolic perturbations associated with complex-I inhibition. Indeed, we demonstrate that inhibition of complex-I using Phenformin, Rotenone and IACS-010759 leads to impairment of ATP production and inhibition of proliferation (**Fig. 1E & Fig. S2A**). Given that we observed a very early decline in MYC levels after only 3 hours of IACS-010759 treatment (**Fig. 5A**), it is plausible to presume that this loss of MYC

occurs prior to the impairment in cell growth.

3. On p. 9. The authors state: “we found that IACS- 010759 treatment decreased the proportion of self-renewing stem cell populations in HD-MB03 and SU_MB002 G3 MB cells, which coincided with a decrease in the levels of the stemness transcription factor SOX2 (Fig. 2G & H).” As discussed above, it is not clear whether this statement is true. If stem cells are unable to proliferate, then they will be unable to give rise to the cells that comprise tumor spheres. Therefore, it seems possible that the observed results are not due to a quantitatively “decreased proportion of self-renewing stem cells” as much as to an inability of these cells to maintain a state of proliferation. Sox2, Nanog and NES are cell cycle regulated so their loss of expression (Fig. 2H) could be due either to loss of the stem cell population, to their inability to maintain a state of proliferation or to some combination of the two.

We agree with the reviewer that a decrease in stem cell populations can occur as the result of a decrease in the proliferation of stem-cells or a loss in stemness capacity and induction of differentiation. We have demonstrated that IACS-010759 impairs the growth of group 3 MB cells (**Fig. 2A & Fig. S2A**), and promotes differentiation by monitoring the upregulation of differentiation markers TUBB3/B3-tubulin, MAP2, NEUROD1 and NEUROG1 (**Fig. 2I-J**).

We have now performed additional assays characterizing the stemness properties of group 3 MB cells following IACS-010759 treatment including secondary tumorsphere formation and the limiting dilution assay. In the secondary tumorsphere assay, primary tumorspheres are dissociated and re-seeded which allows for enrichment and selection of stem cells with self-renewal capacity. We found that IACS-010759 treatment hampered secondary tumorsphere formation capacity, further supporting the notion that IACS-010759 treatment impairs the stemness capacity of G3 MB cells (**Fig. 2G & Fig. 2H**). In the limiting dilution assay, cells are seeded by serial dilution in decreasing densities from 100 to 1 cell(s) per well. The percentage of wells without tumorspheres are then analyzed to determine the relative proportion of cells with self-renewal capacity within a heterogeneous cell population. We found that IACS-010759 significantly impaired the tumorsphere formation efficiency and proportion of self-renewing stem cells in group 3 MB cells (**Fig. 2G & Fig. 2H**). As the reviewer anticipated, these findings suggest that IACS-010759 impairs the stem cell population through a combination of impaired proliferation and hampering stemness properties while promoting a transition towards a more differentiated, less proliferative state. This description has been updated in the revised version of the text (**pg. 11, lines 7-10**).

4. The inhibition of Complex I likely does much more than simply allow Myc to be degraded due to accumulating oxidative damage. Among the most likely of the consequences is an inhibition of mitochondrial ATP production, which would be consistent with the observed aberrant ROS production. Indeed, they show this to be the case in Fig. 4A. A wide range of Myc inhibitors, regardless of their structure, have been previously shown to inhibit ATP production in association with cell cycle arrest and differentiation of certain tumor cells (Oncotarget. 2015 Jun 30;6(18):15857-70). Moreover, ATP depletion alone, just like Myc inhibition, is sufficient to induce cell cycle arrest and induce differentiation of certain tumor cells, without any effect on Myc levels. This leads one to wonder how much of the anti-tumor effect the authors see in MB cells is due to Myc depletion versus other mechanisms, esp. those related to an overall depletion of ATP. The authors themselves have demonstrated that genes involved in OXPHOS are among the most highly enriched following treatment of MB cells with 101759. The ability to restore the growth of MB cells almost to normal while maintaining them in 101759 (Fig. 3E) is an important

experiment as it suggests that the growth inhibition of the cells is truly due the inhibition of Myc rather than Complex I. However, it still strikes me as odd that inhibition of Complex I to a degree that significantly reduces ATP levels allows for normal rates of growth when Myc is restored. This is even more reason for the authors to measure ATP levels in cells (and ROS levels as well) in cells that are treated with 010759 but in which Myc has been restored. I would speculate that the re-expression of Myc may restore normal Complex I activity and normalize both ATP and ROS levels. How this would be done in the face of 010759 is unclear since the precise mechanism of 010759 is unclear. They should also mention how long the cells shown in Fig. 3E were maintained under these conditions and perhaps even show actual growth curves than single points as they do.

We agree with the reviewer that the effect of complex-I inhibitors on group 3 MB cells is likely not entirely dependent on the downregulation of MYC expression, as metabolic perturbations will no doubt have a plethora of consequences on energy production and nutrient sensing pathways. Indeed, ATP depletion related to complex-I inhibition could also contribute to the observed growth effects in combination with a disruption of MYC expression.

As suggested by the reviewer, to further implicate the importance of MYC downregulation the in response of group 3 MB cells to IACS-010759 treatment, we performed time-response cell count analysis to monitor the effect of MYC restoration on cell proliferation. We found that reinstating MYC levels significantly rescues the growth of G3 MB cells following IACS-010759 treatment, although not 100% to normal levels (Fig. 3E). MYC re-instatement in IACS-010759 treated cells rescued the growth to 68% relative to the control after 48 hours, 78% after 72 hours, and 74% after 96 hours (Fig. 3E). These findings indicate there are additional mechanisms that contribute to the response of group 3 MB cells to IACS-010759 treatment, although MYC downregulation plays a significant role. Furthermore, we found that reintroduction of MYC was unable to restore ATP levels or mitigate ROS accumulation in IACS-010759-treated cells (Fig. S5E), placing the downregulation of MYC abundance as a downstream event occurring after these phenomena.

5. Fig. 5E purports to show evidence for Myc being ubiquitylated. But in the lower panel (Myc IB), I fail to see any evidence for the “laddering” due to this post-translational modification. Similarly, in panels F & G, although there IS an increase in Myc protein following proteasome inhibition, I do not see an accumulation of higher Mw forms of of Ub-Myc that one would expect.

We thank the reviewer for bringing this point to our attention. In the original version of this article, we presented Western blots that were separated on high percentage polyacrylamide gels and images that were taken on low exposure time, which masked the ‘laddering’ effect as these bands were not well separated and were not detected on low exposure. We have repeated these Western blots on low percentage gels and imaged on multiple exposure times and we observe a clear laddering effect of MYC protein in proteasome inhibitor-treated cells, consistent with the accumulation of ubiquitinated MYC protein. We have updated these images in the revised version of this manuscript (Fig. 5F-G, Fig. S5G-H).

6. The results in Fig. 5I&J are indirect in that they do not directly measure oxidized cysteines in Myc; rather, they measure only the amount of Myc protein (presumably unoxidized) that remains following 101759 treatment. In addition, the studies do not demonstrate which of the 10 cysteine residues present in Myc are actually being oxidized and are responsible for changing its half-life. These are really two separate questions as all 10 residues could be highly sensitive to

oxidation, with only one or two being responsible for the half-life change. A couple of approaches that provide different answers to these questions could be used to investigate this. First, if possible, one could isolate Myc protein from the cells (assuming there is enough of it) and subject it to mass-spec to identify specific oxidized/unoxidized cysteine residues. Alternatively (and less satisfying), one could purify recombinant Myc protein and then subject it to an oxidative challenge *in vitro* (for example H₂O₂) and then perform MS. Second, one could express individual cysteine point mutants of Myc to determine which one(s) is (are) the most responsive to destabilizing oxidation. It seems that the most plausible prediction is that there are fewer cysteine residues that are necessary for determining Myc half-life than are actually oxidized.

We agree with the reviewer that these are two very important questions that are critical to answer to help complete the message of this study. For this reason, we have performed a comprehensive series of experiments to address these individual questions.

To identify which of the 10 cysteine residues present in MYC are susceptible to oxidation following IACS-010759 treatment, we performed the well-established biotin-switch assay to measure changes in cysteine oxidation levels (García-Santamarina S et al., *Nat Protoc.* 2014⁷; Burgoyne JR et al., *J Pharmacol Toxicol Methods.* 2013⁸; Li R et al., *Methods Enzymol.* 2017⁹). In this assay, reduced cysteine thiols are blocked with a non-labeled alkylating agent (N-ethylmaleimide, NEM) followed by reduction of oxidized residues using a strong reducing agent (TCEP), and labeling of the newly reduced thiols, which were original subjected to oxidation, with a biotin-labeled alkylating agent (PEG2-maleimide-biotin) (Schematic diagram depicting this procedure is present in **Fig. 6F**). We implemented this powerful and sensitive technique for investigating changes in cysteine oxidation with mutational analysis to determine whether interfering with individual cysteine residues modulates the oxidation levels of MYC protein. To this end, we generated c-terminal GFP-tagged MYC-expressing plasmid constructs with point mutations substituting all 10 individual cysteine residues with glycine and confirmed that all constructs displayed similar overexpression efficiency by monitoring GFP-signal by fluorescence microscopy (**Fig. S6B**). We then leveraged these mutant constructs to determine the susceptibility of individual cysteine residues towards oxidation using the biotin-switch assay, where the GFP-tagged exogenous mutant MYC constructs were immunoprecipitated and changes in their oxidation status were monitored by blotting and detection by chemiluminescence using streptavidin-HRP. In the principal of this assay, if a particular cysteine residue is normally susceptible to oxidation following IACS-010759 treatment, then we would expect to observe less biotin labeling and decreased chemiluminescent signal when that residue is mutated to glycine as compared to the WT MYC control construct. Using this assay, we confirmed that WT MYC is undergoing oxidation following IACS-010759 treatment as we observed a ~2-fold increase in chemiluminescent biotin-labeling signal (**Fig. S6C**). As the reviewer speculated, we found that the majority of MYC cysteine residues were responsible for a proportion of MYC oxidation following IACS-010759 treatment, as eight of the cysteine mutant constructs (C85G, C132G, C148G, C186G, C203G, C315G, C357G, and C453G) displayed significantly decreased biotin-labeling potential as compared to the WT-control (**Fig. 6G (i) & (ii)**). Additionally, findings from global cysteine oxidation proteomics analysis performed by two separate groups confirmed that MYC protein is susceptible to cysteine oxidation under a variety of different oxidative stressors (van der Reest J et al., *Nat Commun.* 2018¹⁰; Xiao H et al., *Cell.* 2020¹¹). These studies identified C85, C315, and C357 as potential cysteine oxidation sites. The difference in oxidative stress inducing agents used in these studies combined with the lower sensitivity of proteomics analysis to identify only the most abundant peptides and the fact that certain digestion protocols may not allow all MYC cysteine residues to be covered by mass spectrometry, could explain why only a small fraction of MYC cysteine residues were identified in these analyses.

Additionally, as the reviewer suggested, to determine which cysteine residues play a role in mediating MYC degradation, we overexpressed GFP-tagged MYC constructs in HD-MB03 cells containing either wild-type MYC or individual cysteine mutants, and then subjected cells to IACS-010759 treatment. We then performed immunoblotting for GFP to detect only exogenous MYC protein to determine how individual cysteine residues impact MYC stability and degradation. We confirmed that exogenous WT-MYC protein is efficiently degraded following IACS-010759 treatment in HD-MB03 cells. As the reviewer anticipated, we observed that there were fewer cysteine residues ultimately responsible for MYC degradation than those which were actually oxidized. We found that a total 5 mutant cysteine MYC constructs (C148G, C203G, C315G, C357G and C453G) significantly impaired degradation potential following IACS-010759 treatment, whereas the other 5 residues had no significant impact on MYC degradation (C40G, C85G, C132G, C186G, and C223G) (**Fig. 6H (i) & (ii)**).

7. The implication of much of the work presented in Fig. 7 (and consistent with the simplest model) is that a combination of altered Complex I, PDH and MPC activities are responsible for an accumulation of intra-mitochondrial acetylCoA, which in turn is responsible for the acetylation of SOD2 (a mitochondrial protein). However, the authors present an incomplete story of a potentially complicated mechanism. First, they show only changes in the levels of pPDH, which are consistent with an increase in its activity. However, they do not measure PDH activity directly, which can easily be done by examining the conversion of ^{14}C -pyruvate to $^{14}\text{C}\text{O}_2$. This would potentially allow them to place their MPC and pPDH changes in context. Most importantly, they do not directly measure the levels of acetylCoA in mitochondria. Finally, they fail to account for the possibility of alternate sources of acetylCoA other than pyruvate, most likely from an increase in the activity of FAO, which can be measured directly as well using labeled fatty acids such as palmitate. Defective mitochondria, most notably those resulting from loss of Myc are well known to compensate for their OXPHOS defects by increasing their reliance on FAO. Finally, the inferred change in PDH activity based on its reduced inhibitory phosphorylation might not necessarily correlate with acetylCoA levels; for example, it could be a compensatory mechanisms to try to rectify severely compromised levels of acetylCoA as a result of a Complex I defect. This is reasonable since AcCoA regulates PDH activity indirectly via its allosteric inhibition/activation of PDK1 and PDP2.

This is an excellent point made by the reviewer. We understand that although decreased phosphorylation suggests an increase in enzymatic activation of PDH, it does not directly measure PDH activity or Acetyl-CoA levels. While we do not have access to the appropriate radioisotope-handling facilities to utilize ^{14}C -pyruvate, we implemented a similar approach to the one that was suggested using U^{13}C_3 -pyruvate isotope tracing by mass spectrometry in collaboration with the metabolomics innovation centre (TMIC), Victoria BC. We observed decreased ^{13}C isotopologue labeling of lactate derived from pyruvate which corresponded with an increase in the abundance of ^{13}C -labeled Acetyl-CoA in IACS-010759 treated G3 MB cells as compared to controls (**Fig. 7C**), indicating increased shuttling of pyruvate to Acetyl-CoA via enhanced PDH activity. Furthermore, we observed enhanced ^{13}C -labeled isotopologues of several TCA-cycle intermediates including citrate, fumarate, and malate (**Fig. 7D**). Altogether, these findings support our observations that decreased phosphorylation of PDH corresponds with enhanced activity following IACS-010759 treatment. To place the role of the MPC and p-PDH changes in the context of regulating total cellular Acetyl-CoA pools, we additionally measured Acetyl-CoA using a quantitative fluorometric-based assay and we observed that IACS-010759 treatment significantly increased Acetyl-CoA levels compared to control G3 MB cells and importantly, this accumulation of total Acetyl-CoA could be blocked using the MPC

inhibitor UK-5099 (Fig. 7E & Fig. 8B). While we don't rule out the possibility of other sources contributing to Acetyl-CoA pools such as fatty acids, our findings indicate that the regulation of pyruvate processing towards Acetyl-CoA mediated by PDH and MPC plays a significant role in regulating Acetyl-CoA production and subsequent protein acetylation.

8. Fig. 8 shows the tumors of 010759-treated mice to be significantly reduced at 18 days. The authors present the difference in tumor size as areas (mm²) rather than as volumes (mm³) and this should be corrected as the differences will be even more impressive than the 7-8-fold they claim based on area. Alternatively, the tumors could be weighed. More importantly, this raises the question of why the treated animals only lived for an additional ~10 days given the large size differences seen at day 18. What were the sizes of the tumors in the treated mice at the time of death? Presumably, they were at least as large as those in the d18 control animals, but if not, then other mechanisms of 010759 action in vivo need to be considered. If they were as large as d18 untreated tumors, then what accounted for the sudden increased rate of tumor growth between d18 and d28? Also, can they demonstrate an increase in either SOD2 acetylation or Myc oxidation as they were able to show in vitro?

This is an excellent suggestion provided by the reviewer. The plot of tumor areas presented in the original version of this manuscript is based on quantification of MRI images taken on day 18 from the same relative imaging plane in all animals. We have now quantified the images from multiple planes to calculate an estimated tumor volume and as the reviewer speculated, we see the difference in tumor size is even more drastic between control (mean volume of 193.6 mm³) and IACS-010759 (mean volume of 18.52 mm³) treated animals (Fig. 9D (i) & (ii)). In this study, the whole mouse brains were dissected upon study endpoint and fixed in formalin and embedded in paraffin to prepare slides for immunohistochemistry. Because tumors tissues were not separated from the brains, we are unable to weigh the tumors from this study.

Given that the HD-MB03 tumor model is highly aggressive, where untreated animals die within 18-25 days following tumor cell implantation (Milde T et al., *J Neurooncol.* 2012²; Zagozewski J et al., *Nat Commun.* 2020³), a 10-day survival extension is very significant and prolongs their expected lifespan by approximately 50%. Based on quantification of H&E-stained brain tissues of endpoint tumors, we see that IACS-010759 treated animals do indeed possess smaller tumors at endpoint as compared to controls (Fig. 9D (i) & (ii)). We are mindful that we are only administering a single treatment agent as opposed to multi-modal aggressive therapeutic regimens consisting of chemotherapy, radiation, and surgery that are used in clinics. Moreover, animals are immune-compromised and do not have an intact immune system that can help to fight off the tumor. Therefore, our treatment strategy is likely not sufficient to completely regress the tumor and indeed, animals still eventually succumb to the disease.

There may be multiple explanations as to why animals ultimately become sick from disease progression despite harboring lower tumor burden. Although these tumors are smaller, they are not completely absent. Even small tumors in the brain, depending on where they are located, can cause symptoms that lead to animal death. Moreover, it is unclear what the effect of IACS-010759 treatment is on tumor-associated processes such as angiogenesis and metastasis which play a major role in disease progression. These processes may still be active even though IACS-010759 treatment is significantly impairing tumor growth.

Finally, as the reviewer suggested, we found that IACS-010759-treated HD-MB03 tumors displayed increased levels of acetylated SOD2 K68 ($p = 0.0365$; unfortunately there are no commercially

available acetylated SOD2 K122 antibodies suitable for IHC) (Fig. 10C). This inhibition of SOD2 activity corresponded with increased oxidative stress in tumors as demonstrated using two distinct markers that detect DNA oxidation (8-hydroxy-2-deoxyguanosine; 8-oxo-DG, $p = 0.0042$) and lipid peroxidation (4-hydroxy-2-noneal; 4-HNE, $p = 0.0351$) (Fig. 10B). Finally, in line with our observed *in vitro* mechanism that IACS-010759-mediated accumulation of ROS promotes MYC oxidation and degradation, we found that IACS-010759 treated tumors displayed depleted MYC levels compared to placebo controls *in vivo* (Fig. 9E; $p = 0.0006$).

Minor points

1. At least two references should be replaced:

-Bottom of p. 3 “..clinical targeting of MYC has remained elusive (9)” . There are better and more recent refs/reviews to cite regarding the elusiveness of current therapeutic approaches.

As suggested by the reviewer we have updated the references to reflect more recent literature pertaining to this statement.

-Bottom of p. 5: Ref. 19 was not the study demonstrating that 10058-F4 directly inhibits Myc-Max heterodimerization by binding directly to Myc. The correct reference for this is: Mol Cancer Ther. 2007 Sep;6(9):2399-408.

We apologize for this error and we sincerely thank the reviewer for bringing this point to our attention. We have updated this reference in the revised version of this manuscript.

2. Bottom of p.10: “This activation of WT TP53 offers an additional benefit for IACS-010759 treatment in G3 MB patients, as it is a desirable therapeutic outcome to suppress tumor growth.” This statement should be removed or modified as it implies that this effect on p53 is somehow specific for 010759 when in fact the effect is due to the down-regulation of MILIP, which is just one of many Myc target genes.

We have modified this statement as follows “G3 MB tumors commonly maintain WT TP53, and we found that upon IACS-010759-mediated downregulation of MYC, *MILIP* transcript levels were also suppressed and this corresponded with an increase in WT TP53 protein levels, which in turn may offer an additional desirable therapeutic benefit for suppressing tumor growth” on **pg 12, lines 7-9** in the revised text.

3. The Myc protein half-life in control MBO3 cells (Fig. 5D) is > 2hr and is convincingly reduced to ~30 min following 101759 addition. However 2 hr is an extremely long half-life for Myc by most standards (actually about 10-times longer). Is the prolonged half-life seen in control MB03 cells specific for MBO3 or is it seen in other MBs? Perhaps MBO3 cells have a defect other than gene amplification that accounts for their high Myc level expression.

Indeed, the reviewer is correct that HD-MB03 cells display a long MYC half-life which is commonly seen in some cancer cells as an additional mechanism to maintain elevated MYC levels along with genomic amplification. We additionally monitored the effect of IACS-010759 treatment on the half-life of MYC in SU_MB002 G3 MB cells. SU_MB002 cells have a much shorter basal half-life of MYC as compared HD-MB03 cells (~30min versus ~2h) although we find that IACS-010759 treatment significantly reduced the half-life of MYC in SU_MB002 cells even further (~20min) (Fig. S5F).

Altogether these findings support IACS-010759 treatment decreases the post-translational stability of MYC within various G3 MB cells that display differing basal levels of MYC half-life.

References

- 1 Romero-Calvo, I. *et al.* Reversible Ponceau staining as a loading control alternative to actin in Western blots. *Anal Biochem* **401**, 318-320, doi:10.1016/j.ab.2010.02.036 (2010).
- 2 Milde, T. *et al.* HD-MB03 is a novel Group 3 medulloblastoma model demonstrating sensitivity to histone deacetylase inhibitor treatment. *J Neurooncol* **110**, 335-348, doi:10.1007/s11060-012-0978-1 (2012).
- 3 Zagozewski, J. *et al.* An OTX2-PAX3 signaling axis regulates Group 3 medulloblastoma cell fate. *Nat Commun* **11**, 3627, doi:10.1038/s41467-020-17357-4 (2020).
- 4 Cavalli, F. M. G. *et al.* Intertumoral Heterogeneity within Medulloblastoma Subgroups. *Cancer Cell* **31**, 737-754 e736, doi:10.1016/j.ccell.2017.05.005 (2017).
- 5 Pan, L. Z. *et al.* The NAD⁺ synthesizing enzyme nicotinamide mononucleotide adenylyltransferase 2 (NMNAT-2) is a p53 downstream target. *Cell Cycle* **13**, 1041-1048, doi:10.4161/cc.28128 (2014).
- 6 Liu, J. *et al.* NMNAT promotes glioma growth through regulating post-translational modifications of P53 to inhibit apoptosis. *Elife* **10**, doi:10.7554/eLife.70046 (2021).
- 7 Garcia-Santamarina, S. *et al.* Monitoring in vivo reversible cysteine oxidation in proteins using ICAT and mass spectrometry. *Nat Protoc* **9**, 1131-1145, doi:10.1038/nprot.2014.065 (2014).
- 8 Burgoyne, J. R., Oviosu, O. & Eaton, P. The PEG-switch assay: a fast semi-quantitative method to determine protein reversible cysteine oxidation. *J Pharmacol Toxicol Methods* **68**, 297-301, doi:10.1016/j.vascn.2013.07.001 (2013).
- 9 Li, R. & Kast, J. Biotin Switch Assays for Quantitation of Reversible Cysteine Oxidation. *Methods Enzymol* **585**, 269-284, doi:10.1016/bs.mie.2016.10.006 (2017).
- 10 van der Reest, J., Lilla, S., Zheng, L., Zanivan, S. & Gottlieb, E. Proteome-wide analysis of cysteine oxidation reveals metabolic sensitivity to redox stress. *Nat Commun* **9**, 1581, doi:10.1038/s41467-018-04003-3 (2018).
- 11 Xiao, H. *et al.* A Quantitative Tissue-Specific Landscape of Protein Redox Regulation during Aging. *Cell* **180**, 968-983 e924, doi:10.1016/j.cell.2020.02.012 (2020).

REVIEWERS' COMMENTS

Reviewer #1 (Remarks to the Author):

The authors have adequately addressed all the points I had raised.

Reviewer #3 (Remarks to the Author):

The authors have responded to all of my original critiques as reviewer #3 and have done an admirable job in doing so. I do not see any additional issues with the manuscript. I therefore recommend acceptance and publication of the revised manuscript.

Reviewer #4 (Remarks to the Author):

In the revised version of the manuscript, the authors have now addressed all my initial concerns. They have provided new data that strengthen their initial findings and allow them to draw robust conclusions.

RESPONSE TO REVIEWERS

Reviewer #1 (Remarks to the Author):

The authors have adequately addressed all the points I had raised.

We appreciate the reviewer's feedback we are pleased that they are satisfied with our response.

Reviewer #3 (Remarks to the Author):

The authors have responded to all of my original critiques as reviewer #3 and have done an admirable job in doing so. I do not see any additional issues with the manuscript. I therefore recommend acceptance and publication of the revised manuscript.

We thank the reviewer for their detailed critiques, and we are pleased that the reviewer appreciates our efforts to address all their original concerns and finds the revised manuscript suitable for publication.

Reviewer #4 (Remarks to the Author):

In the revised version of the manuscript, the authors have now addressed all my initial concerns. They have provided new data that strengthen their initial findings and allow them to draw robust conclusions.

We are glad that the reviewer feels we have addressed all their concerns and we agree that their suggestions have helped us strengthen our findings and have greatly improved our manuscript.